# Lieb-Schultz-Mattis theorem in long-range interacting systems and generalizations

Ruizhi Liu,[1,2] Jinmin Yi,[2,3] Shiyu Zhou,[2] and Liujun Zou[2]

[1]*Department of Mathematics and Statistics, Dalhousie University, Halifax, Nova Scotia, Canada, B3H 4R2*
[2]*Perimeter Institute for Theoretical Physics, Waterloo, Ontario, Canada N2L 2Y5*
[3]*Department of Physics and Astronomy, University of Waterloo, Waterloo, Ontario, Canada N2L 3G1*

In a unified fashion, we establish Lieb-Schultz-Mattis theorem in long-range interacting systems and its generalizations. We show that, for a quantum spin chain, if the multi-spin interactions decay fast enough as their ranges increase and the Hamiltonian has an anomalous symmetry, the Hamiltonian cannot have a unique gapped symmetric ground state. If the Hamiltonian contains only 2-spin interactions, this theorem holds when the interactions decay faster than $1/r^2$, with $r$ the distance between the two interacting spins. Moreover, any pure state with an anomalous symmetry, which may not be a ground state of any natural Hamiltonian, must be long-range entangled. The symmetries we consider include on-site internal symmetries combined with lattice translation symmetries, and they can also extend to purely internal but non-on-site symmetries. Moreover, these internal symmetries can be discrete or continuous. We explore the applications of the theorems through various examples.

## I. Introduction

Understanding and realizing interesting quantum phases of matter is a central goal of condensed matter physics. In this regard, Lieb-Schultz-Mattis-type (LSM) constraints are extremely powerful, which, as initially stated, rule out a unique gapped symmetric ground state based on some basic symmetry-related properties of the system's Hamiltonian, without referring to any other detail of the Hamiltonian [1–3]. Recently, LSM constraints have been interpreted from various perspectives and generalized to different contexts [4–26]. Furthermore, these constraints are identified as a key ingredient to study the classification of quantum phases of matter in a lattice system [18, 27, 28].

Previous studies of LSM constraints often focus on systems with local interactions. However, many systems feature long-range interactions, which usually take the form of a 2-body interaction that decays as $1/r^{\mathfrak{a}}$, with $r$ the distance between the two interacting objects and $\mathfrak{a}$ an exponent. As examples, electronic systems have Coulomb interaction with $\mathfrak{a} = 1$, Rydberg atoms have dipolar or van der Waals interactions with $\mathfrak{a} = 3$ or $\mathfrak{a} = 6$, and for trapped ions $\mathfrak{a}$ can be tuned between 0 and 3 [29]. So an important question is: Are LSM constraints applicable to long-range interacting systems?

In this paper, we prove and generalize LSM theorems in quantum spin chains with long-range interactions, detailed in theorems III.1 and III.3 below. In essence, we show that if 1) the long-range interactions decay fast enough as their ranges increase and 2) the system has an anomalous symmetry [25], then the system cannot have a unique gapped symmetric ground state, and all symmetric pure states must be long-range entangled, regardless whether the states are ground states or not. We remark that the Hamiltonians we consider can contain generic $k$-body interactions with $k > 2$. For 2-body interactions decaying as $1/r^{\mathfrak{a}}$, our theorems hold when $\mathfrak{a} > 2$ (for $k$-body interactions with $k > 2$, the condition under which our theorems hold is stated in Eq. (III.1)). The type of symmetries under consideration is also very broad, including an on-site symmetry combined with the lattice translation symmetry, as featured in the original LSM theorems. Additionally, the symmetry can be purely internal but non-on-site. Furthermore, the internal symmetries can be either discrete or continuous. Besides incorporating long-range interactions, our theorems generalize the original LSM theorems in two ways. First, our theorems apply to a more general class of symmetries. Second, the original LSM theorems often concern about the Hamiltonians' spectra, but our results also govern the entanglement properties of general states. Our results have wide applicability, and we will discuss some examples below.

## II. Operator algebra formalism

To have a clean notion of locality, we wish to work with systems of infinite size. The operator algebra formalism deals with both finite and infinite systems conveniently. Below we first apply this formalism to infinite systems, which can be viewed as the thermodynamic limits where a sequence of finite systems converge to. From these results, we will extract important implications on finite systems.

For finite systems, the operator algebra formalism is just the usual quantum mechanics in the Heisenberg picture. Here we briefly review this formalism in the context of infinite systems before applying it. We start with the notions of operator algebras and states of infinite size. Then we discuss the symmetry actions and the associated anomaly index developed in Ref. [25], which characterizes the interplay between locality and symmetry.

Given an infinite lattice $\Lambda$ and its finite subset $\Gamma \subset \Lambda$, operators acting trivially outside $\Gamma$, including $c$-numbers, form a local operator algebra,[1] denoted by $\mathcal{A}^\ell_\Gamma$. The algebra of all local operators is defined as

$\mathcal{A}^\ell := \bigcup_{\Gamma \subset \Lambda, \ |\Gamma| < \infty} \mathcal{A}^\ell_\Gamma$, with $|\Gamma|$ the cardinality of $\Gamma$. A useful fact is that $\mathcal{A}^\ell$ factorizes as $\mathcal{A}^\ell = \mathcal{A}^\ell_{\Lambda_0} \otimes \mathcal{A}^\ell_{\Lambda_0^c}$, where $\Lambda_0$ is any (finite or infinite) subset of $\Lambda$, $\Lambda_0^c$ is the complement of $\Lambda_0$ in $\Lambda$, and $A^\ell_{\Lambda_0} := \bigcup_{\Gamma \subset \Lambda_0, |\Gamma| < \infty}$ and $A^\ell_{\Gamma_0^c} := \bigcup_{\Gamma \subset \Lambda_0^c, |\Gamma| < \infty}$ are the local operator algebras in $\Lambda_0$ and $\Lambda_0^c$, respectively. Later we often take $\Lambda_0 = (-\infty, 0)$ and $\Lambda_0^c = [0, \infty)$, i.e., the left and right half chains, respectively.

The Hilbert space $\mathcal{H}_\Gamma$ associated with the finite subset $\Gamma$ is the tensor product of the finite-dimensional on-site Hilbert space $\mathcal{H}_k$ for each site $k$ in $\Gamma$, i.e., $\mathcal{H}_\Gamma = \bigotimes_{k \in \Gamma} \mathcal{H}_k$. However, in contrast to finite systems, the total Hilbert space for infinite systems is not well-defined. So how should we represent a quantum state? Recall that a quantum state in finite systems can be specified by the expectation values of all operators with respect to it, so we can define states in infinite systems analogously. Concretely, a state $\psi$ is a linear functional $\psi : \mathcal{A}^\ell \to \mathbb{C}$ that satisfies positivity (i.e., $\psi(A^\dagger A) \geqslant 0$ for any local operator $A$) and normalization (i.e., $\psi(I) = 1$ with $I$ the identity operator).

We remark that, under this definition, states in different superselection sectors of an infinite system (i.e., states that cannot be related by local operators) can only form classical mixtures, but not quantum superpositions. For example, although the $N$-qubit GHZ state $\frac{1}{\sqrt{2}} (|0 \cdots 0\rangle + |1 \cdots 1\rangle)$ is pure, its infinite-system version is a mixed state, because no local operator can couple the states $|0 \cdots 0\rangle$ and $|1 \cdots 1\rangle$ (see Appendix A for more discussion).

Our proofs of the theorems below emphasize the entanglement aspect of states. To discuss the entanglement structure of quantum states in infinite systems, it is useful to introduce the split property [30–32]. When we cut the chain at any point, say, the origin, a pure state $\psi$ of the whole chain may not be factorized[2] as $\psi \simeq \psi_{<0} \otimes \psi_{\geqslant 0}$ for some *pure* states $\psi_{<0}$ on the left-half chain and $\psi_{\geqslant 0}$ on the right-half chain[3]. If $\psi$ factorizes in this way indeed, then we say that at the origin $\psi$ splits. Intuitively, states that split at the origin have limited entanglement between the left and right halves.

The above discussion on operators and states pertains to general infinite chains. However, we are specifically interested in quantum spin chains with symmetries. Symmetries in this formalism are described by automorphisms associated with the algebra $\mathcal{A}^\ell$. An automorphism is an invertible linear map $\varphi : \mathcal{A}^\ell \to \mathcal{A}^\ell$ satisfying $\varphi(AB) = \varphi(A)\varphi(B)$ and $\varphi(A^\dagger) = \varphi(A)^\dagger$ for any $A, B \in \mathcal{A}^\ell$. Automorphisms of $\mathcal{A}^\ell$ form a group un-

---

[1] An operator algebra means a set of operators that can add and multiply, such that this set is closed under finite additions and multiplications.

---

[2] The tensor product of states is defined by $(\psi_{<0} \otimes \psi_{\geqslant 0})(A \otimes B) = \psi_{<0}(A)\psi_{\geqslant 0}(B)$ for $A \in \mathcal{A}^\ell_{<0}$ and $B \in \mathcal{A}^\ell_{\geqslant 0}$.

[3] The equivalence "$\simeq$" of states here means that for any $\epsilon > 0$, there exists a finite region $\Gamma_\epsilon$, such that $|\psi(A) - (\psi_{<0} \otimes \psi_{\geqslant 0})(A)| < \epsilon ||A||$ for any $A \in \mathcal{A}^{ql}_{\Gamma_\epsilon^c}$, where $\Gamma_\epsilon^c$ is the complement of $\Gamma_\epsilon$.

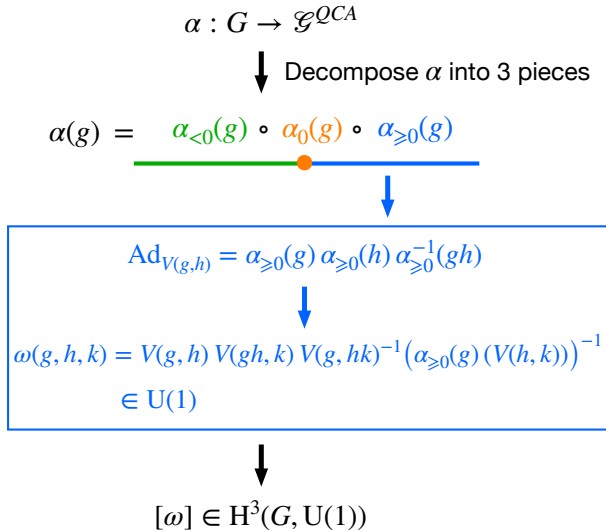

FIG. 1. An illustration on how to obtain anomaly index $\omega \in \mathrm{H}^3(G; \mathrm{U}(1))$ from the symmetry action $\alpha$.

der finite compositions, denoted by $\mathrm{Aut}(\mathcal{A}^\ell)$. There is a special subgroup of $\mathrm{Aut}(\mathcal{A}^\ell)$ called quantum cellular automata (QCA), denoted by $\mathcal{G}^{\mathrm{QCA}}$, which preserves the locality of operators. More precisely, an automorphism $\varphi$ is a QCA if $\varphi(A) \in \mathcal{A}^\ell_{B(\Gamma, r_\varphi)}$ for each $A \in \mathcal{A}^\ell_\Gamma$, where $B(\Gamma, r_\varphi) := \{x \in \Lambda \,|\, d(x, \Gamma) \leqslant r_\varphi\}$, with $r_\varphi > 0$ depending only on $\varphi$ and $d(x, \Gamma)$ the distance between $x$ and $\Gamma$. The structure of $\mathcal{G}^{\mathrm{QCA}}$ is well-understood in 1D [33]. In essence, 1D QCA are combinations of finite-depth quantum circuits and translations (see Refs. [34, 35] for review).

In the main text, we will focus on unitary symmetries implemented by QCA, as they preserve locality in the most strict sense (in the appendices, our considerations

are extended to a more general class of symmetry actions, i.e., locality preserving automorphisms, and our main theorems still hold). Concretely, given a symmetry group $G$, the symmetry action can be represented by a group homomorphism $\alpha : G \to \mathcal{G}^{\mathrm{QCA}}$. This symmetry may contain internal and/or translation symmetry, and the internal symmetry may be discrete or continuous, on-site or non-on-site.

Given such a symmetry action $\alpha : G \to \mathcal{G}^{\mathrm{QCA}}$, an important concept is the anomaly index, which takes values in $\mathrm{H}^3(G, \mathrm{U}(1))$ [25] (see Appendix B for a review of group cohomology). The construction of this anomaly index is similar to the previous work [36], and the innovation of this new anomaly index is that it applies to translation symmetries and continuous internal symmetries. Below we sketch the definition of the anomaly index, and more details can be found in Ref. [25] and Appendix C.

First, suppose $\alpha$ is an internal symmetry action (i.e., it contains no translation) and choose an arbitrary site, say, the origin, then it can be shown that $\alpha$ can be decomposed as

$$\alpha = \alpha_{<0}\,\alpha_0\,\alpha_{\geqslant 0} \;, \tag{II.1}$$

where $\alpha_{\geqslant 0}$ (resp. $\alpha_{<0}$) is an automorphism of $\mathcal{A}^\ell_{\geqslant 0}$ ($\mathcal{A}^\ell_{<0}$), and $\alpha_0$ is a local unitary (see Fig. 1). Although $\alpha$ is a group homomorphism, in general $\alpha_{\geqslant 0}$ is not. In fact, for any $g, h \in G$,

$$\alpha_{\geqslant 0}(g)\,\alpha_{\geqslant 0}(h) = \mathrm{Ad}_{V(g,h)}\,\alpha_{\geqslant 0}(gh) \;, \tag{II.2}$$

where $V : G \times G \to \mathcal{U}^\ell$ with $\mathcal{U}^\ell$ the group of local unitaries is not necessarily a homomorphism, and $\mathrm{Ad}_V(A) := VAV^\dagger$ for any $A \in \mathcal{A}^\ell$. The associativity of $\alpha_{\geqslant 0}$, i.e., $(\alpha_{\geqslant 0}(g)\,\alpha_{\geqslant 0}(h))\,\alpha_{\geqslant 0}(k) = \alpha_{\geqslant 0}(g)\,(\alpha_{\geqslant 0}(h)\,\alpha_{\geqslant 0}(k))$, puts further constraints on $V$: $\mathrm{Ad}_{\omega(g,h,k)} = 1$, where

$$\omega(g,h,k) = V(g,h)V(gh,k)V(g,hk)^{-1}(\alpha_{\geqslant 0}(g)(V(h,k)))^{-1} \;. \tag{II.3}$$

This means the above $\omega$ is actually a phase since it commutes with all local operators. It can be checked that $\omega$ satisfies the 3-cocycle condition, and multiplying $V(g,h)$ by a phase $\rho(g,h) \in \mathrm{U}(1)$ shifts $\omega$ by a 3-coboundary. Therefore, $\omega$ specifies an element in $\mathrm{H}^3(G, \mathrm{U}(1))$, and this element is defined as the anomaly index associated with the symmetry action $\alpha$.

If $\alpha$ contains translation, one can stack the system with another copy on which the translation acts oppositely. The symmetry action on this composite system (denoted by $\alpha_\otimes$) contains no translation, and the anomaly index of $\alpha$ is defined to be the index of $\alpha_\otimes$.

In Appendix C, we prove that this anomaly index

is independent of the choice of the site to decompose $\alpha$ in Eq. (II.1), which was not explicitly proved in Refs. [25, 36].

With the above definition of anomaly index, we say that the $G$-symmetry is anomalous if $\omega \neq 1 \in \mathrm{H}^3(G; \mathrm{U}(1))$. Otherwise, we say it is anomaly-free or non-anomalous.

To connect the above discussion with the more familiar notions, let us discuss an example. Consider a quantum spin chain with a symmetry $G = \mathbb{Z} \times G_{\mathrm{int}}$, where $\mathbb{Z}$ represents translation and $G_{\mathrm{int}}$ is an internal symmetry (taken as either a discrete group or a finite dimensional Lie group). Then $\mathrm{H}^3(G, \mathrm{U}(1)) \simeq \mathrm{H}^2(G_{\mathrm{int}}, \mathrm{U}(1)) \oplus$

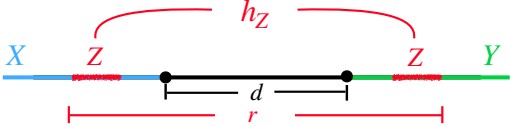

FIG. 2. The interaction of two disjoint intervals $X, Y$ seprated by distance $r$. The range of interaction is denoted by $Z$ with $\text{diam}(Z) = d$.

$H^3(G_{\text{int}}, U(1))$ [4]. The part $H^2(G_{\text{int}}, U(1))$ means if the degrees of freedom in a unit cell form a projective representation under $G_{\text{int}}$, which is precisely the condition of the original LSM theorems, the $G$-symmetry is anomalous. The part $H^3(G_{\text{int}}, U(1))$ means that even for a purely internal symmetry $G_{\text{int}}$, the $G$-symmetry can be anomalous if its anomaly index corresponds to a nontrivial element in $H^3(G_{\text{int}}, U(1))$. We will present an example of such internal symmetries below.

## III. LSM theorem in long-range interacting systems and generalizations

Now we proceed to our main theorems, which accommodate long-range and many-body interactions. Consider a 1D Hamiltonian with at most $k$-body interactions, $H = \sum_{|Z| \leqslant k} h_Z$, that satisfies [37]

$$\max_{i \in \mathbb{Z}} \left( \sum_{Z: Z \ni i, \ \text{diam}(Z) = d} ||h_Z|| \right) = O(r^{-\mathfrak{a}}), \ \mathfrak{a} > 2 , \tag{III.1}$$
$$\text{and } \max_{i \in \mathbb{Z}} \|h_i\| \leqslant B ,$$

where $\text{diam}(Z) = \sup_{x, y \in Z} |x - y|$, and $h_i$ is the on-site potential at site $i$. If $H$ satisfies Eq. (III.1), it is deemed as *admissible*. Specifically, if the Hamiltonian includes at most 2-body long-range interactions, Eq. (III.1) indicates that the interactions decay faster than $r^{-2}$, with $r$ the distance between the two interacting spins. Eq. (III.1) ensures that for any disjoint intervals $X, Y$ (separated by $d$) as in Fig. 2, their interaction $V_{X,Y} = \sum_{Z: Z \cap X \neq \emptyset, Z \cap Y \neq \emptyset} h_Z$ goes to 0 as $d \to \infty$.

To derive our theorem III.1, which extends the standard LSM theorem to long-range interacting systems with a general symmetry described by QCA, we present several lemmas about the properties of the ground states of admissible Hamiltonians.

**Lemma III.1.** *A gapped ground state of an admissible Hamiltonian in 1D must split at every site.*

This lemma is deduced by combining Sec. II of Ref. [37] and theorem 1.5 of Ref. [30].

**Lemma III.2 (Theorem A.5 in Appendix A).** *A locally-unique gapped ground state of an admissible*

Hamiltonian $H$ is pure.[4]

These two lemmas illustrate profound connections between the spectral property of Hamiltonians and the entanglement property of the ground states. The next lemma bridges the entanglement properties of states with the anomaly of the symmetry action.

**Lemma III.3 (Remark 4.1 of Ref. [25]).** *Given a symmetry action $\alpha : G \to \mathcal{G}^{\text{QCA}}$ on a quantum spin chain, if there exists a $G$-symmetric pure state $\psi$ which splits at any site, then the associated anomaly index $\omega = 1$.*

Our first main theorem can be obtained by considering lemmas III.1, III.2 and III.3.

**Theorem III.1.** *If $\alpha : G \to \mathcal{G}^{\text{QCA}}$ is a symmetry action on a quantum spin chain with an anomaly index $\omega \neq 1$, then there cannot be a locally-unique $G$-symmetric gapped ground state for a $G$-symmetric admissible $H$.*

To prove this theorem, we employ an argument by contradiction (see also Appendix D). Assume that $\psi$ is a locally-unique gapped ground state of a $G$-symmetric admissible $H$. According to lemmas III.1 and III.2, $\psi$ must be pure and split at every site. Additionally, it is $G$-symmetric by assumption. However, lemma III.3 states that the anomaly index $\omega = 1$, which contradicts our initial assumption.

Theorem III.1 concerns with infinite systems, but real systems are of finite size. To extract useful implications on finite systems, we utilize another theorem.

**Theorem III.2 (Theorem E.1 in Appendix E).** *Suppose a sequence of $G$-symmetric admissible Hamiltonians, $\{H_L\}$, converges to an admissible Hamiltonian $H$ as $L \to \infty$. If each $H_L$ has a unique $G$-symmetric gapped ground state, then this sequence of ground states converges to a $G$-symmetric locally-unique gapped ground state of $H$ as $L \to \infty$.*

Combining theorems III.1 and III.2, we deduce that if (a set of) large but finite systems described by admissible Hamiltonians with an anomalous symmetry have a well-defined thermodynamic limit, then they cannot have a unique $G$-symmetric gapped ground state.

Next, we move to theorem III.3, which generalizes the original LSM theorem and concerns with the entanglement property of a pure state with an anomalous symmetry in a finite system, and this state may not be

---

[4] A locally-unique ground state means the unique gapped ground state in a superselection sector. Namely, there may be other ground states, but they fall into other superselection sectors. See Appendix A for more details.

a ground state of any natural Hamiltonian.[5] Recall that a pure state in a sequence of finite but large systems is short-range entangled (SRE) if it can be deformed into a product state by a time evolution under a local Hamiltonian over a duration that does not diverge as the system size goes to infinity, otherwise it is long-range entangled. In addition, consider the following lemma.

**Lemma III.4 (Propositions E.1 and E.2 in Appendix E).** *In the thermodynamic limit, an SRE state must be pure and split at every site.*

Recall that pure states in finite systems can become mixed in the thermodynamic limit (e.g., the GHZ state). Lemma III.4 ensures that this does not occur for SRE states. It also clarifies the relation between being SRE and the split property, i.e., the former implies the latter, but the converse may not be true in general.

By using Lemmas III.3 and III.4, we deduce

**Theorem III.3.** *There cannot be a $G$-symmetric short-range entangled pure state for sufficiently long spin chains if the anomaly index $\omega \neq 1 \in \mathrm{H}^3(G; \mathrm{U}(1))$.*

Special versions of theorem III.3 were proved before [36, 38]. For example, in a spin-1/2 chain no SRE state is compatible with a $G = \mathbb{Z} \times SO(3)$ symmetry [38]. A widely studied $G$-symmetric state in this case is the ground state of the nearest-neighbor anti-ferromagnetic Heisenberg model, which indeed realizes a conformal field theory with long-range entanglement. Our theorem applies to general symmetries described by QCA, which is extended to symmetries described by locality preserving automorphisms in the appendices.

## IV. Examples and applications

The first example is the spin-1/2 XXZ chain with long-range interactions [39, 40]. The Hamiltonian is

$$H = \sum_{i>j} \frac{1}{|i-j|^{\mathfrak{a}}} \left( J_{ij}^z S_i^z S_j^z - S_i^x S_j^x - S_i^y S_j^y \right) \ , \quad \text{(IV.1)}$$

where $J_{ij} = O(|i-j|^{-\mathfrak{a}})$ when $|i-j| \to \infty$ with $\mathfrak{a} > 2$.

---

[5] SRE states have exponentially decaying correlation functions (see Appendix E), but the ground states of Hamiltonians with power-law interactions generically have power-law correlation functions, so they are generically not SRE, even without any anomalous symmetry.

where $J_{ij}^z$ can be positive or negative. Notice that $H$ satisfies the admissible condition in Eq. (III.1) when $\mathfrak{a} > 2$. This model has a $O(2) \times \mathbb{Z}$ symmetry, where $\mathbb{Z}$ is lattice translation and $O(2)$ is generated by rotation around the $z$-axis and reflection about the $xy$ plane. We have $\mathrm{H}^3(O(2) \times \mathbb{Z}; \mathrm{U}(1)) \simeq \mathbb{Z}_2$, which measures the on-site spin quantum number $S$. If $S \in \mathbb{Z} + \frac{1}{2}$, the anomaly index is nontrivial and our theorems apply, which means that this model cannot have a unique symmetric gapped ground state.

From Ref. [41], for $\mathfrak{a} > 2$ the phase diagram of Eq. (IV.1) contains a ferromagnetic phase, an antiferromagnetic phase and a continuous symmetry breaking phase, which all spontaneously break some symmetries, and an XY phase, which is symmetric but gapless. Indeed, none of these phases has a unique symmetric gapped ground state, agreeing with our theorem III.1. Moreover, the only symmetric phase (XY) is a conformal field theory with long-range entanglement, agreeing with our theorem III.3.

In our second example, the only relevant symmetry is an anomalous $\mathbb{Z}_2$ internal symmetry. For each lattice site, we place a qubit. This $\mathbb{Z}_2$ symmetry acts as [42]

$$\begin{aligned} \alpha(Z_i) &= -Z_i \ , \\ \alpha(X_i) &= Z_{i-1} X_i Z_{i+1} \ , \end{aligned} \quad \text{(IV.2)}$$

where $X_j, Z_j$ are usual Pauli matrices, we also denote $Y_j = iX_j Z_j$. Formally, this symmetry is generated by conjugation with the following infinite product

$$\prod_{j \in \mathbb{Z}} e^{\frac{i\pi}{4} Z_j Z_{j+1}} \prod_{k \in \mathbb{Z}} X_k \ . \quad \text{(IV.3)}$$

This choice of symmetry action corresponds the nontrivial anomaly class in $\mathrm{H}^3(\mathbb{Z}_2; \mathrm{U}(1))$ [25]. We consider the following Hamiltonian, which may be realizable in experimental setups similar to those in Refs. [39, 40]:

$$H = -\sum_{i,j} J_{ij} Z_i Z_j - \sum_i g_i (X_i + Z_{i-1} X_i Z_{i+1}) - \sum_j h_j Y_j (1 - Z_j Z_{j+1}) \ , \quad \text{(IV.4)}$$

For example, $J_{ij}$ can be chosen as

$$J_{ij} = \frac{C_{ij}}{|i-j|^{\mathfrak{a}}} \ , \quad \text{(IV.5)}$$

where $\mathfrak{a} > 2$, and $C_{ij}$ depend on $i, j$ and are bounded by constant $D$ for all $i, j$. This Hamiltonian can break the translation symmetry explicitly.

In the special case where $h_i = 0$, $g_i = 0$ and $C_{ij} = 1$,

this system is the classical long-range Ising model. The ground state is gapped and breaks the $\mathbb{Z}_2$ symmetry spontaneously, agreeing with theorem III.1. In the regime $g > 0$ and $J_{ij} = h_i = 0$, this model realizes a gapless symmetric long-range entangled Luttinger liquid [42], agreeing with theorems III.1 and III.3. For more general couplings such as Eq. (IV.5), other phases are also possible, which are left to future works.

## V. Discussions

In this work, we have proved and generalized the Lieb-Schultz-Mattis theorem in quantum spin chains with long-range interactions and anomalous symmetries. Our results apply to both discrete and continuous internal symmetries, as well as lattice translation symmetries. Currently it is unclear whether theorem III.1 can be extended to the case where the 2-body interactions decay as $1/r^{\mathfrak{a}}$ with $\mathfrak{a} \leqslant 2$, and it is interesting to better understand this. Nevertheless, it should be obvious that theorem III.1 cannot be extended to systems with extremely non-local interactions, such as a system whose Hamiltonian is simply a projector into a symmetric long-range entangled state. Also, it is useful to generalize our results to systems with time reversal and/or point-group symmetry, fermionic systems and higher dimensional systems.

## Acknowledgments

We thank Theo Johnson-Freyd and Michael Levin for helpful discussion. LZ thanks the Kavli Institute for Theoretical Physics for their hospitality during the course of this research. Research at Perimeter Institute is supported in part by the Government of Canada through the Department of Innovation, Science and Industry Canada and by the Province of Ontario through the Ministry of Colleges and Universities. This research was supported in part by grant NSF PHY-2309135 to the Kavli Institute for Theoretical Physics (KITP).

*Note added.* While completing this work, we became aware of an independent work by Ruochen Ma on related subjects, whose paper will appear in the same arXiv post [43].

## Appendices

In these appendices, we provide more details related to the main text. Specifically, we review the operator algebra formalism in Sec. A, which is aimed to provide a basic introduction to this formalism to readers unfamiliar with it. In Sec. B, we review the mathematical definitions of group cohomology and differentiable group cohomology. In Sec. C, we review the construction of the anomaly index. In particular, we prove that this anomaly index is independent of the choice of the cut. In Sec. D, we provide the proof of theorem III.1 in the main text. In Sec. E, we discuss the connection between infinite systems and finite systems, and also present the proof of the theorems 2 and 3 in the main text. Because we will employ a large number of notations in this supplemental material, for the convenience of the readers, we list the frequently used symbols at the end of this document.

## A. Review of the operator algebra formalism

As explained in the main text (see also Sec. A 1 below), working on infinite lattices can be challenging, due to the absence of a total Hilbert space. However, doing so comes with some advantages. For example, strictly speaking, phases of matter are defined for systems in the thermodynamic limit. Thus it is natural to work on infinite lattice while discussing these phases. Moreover, on infinite lattices, we have a better notion of locality (see Sec. A 2). Also, in finite-size systems, locality-preserving automorphisms (defined in definition A.6) do not form a group under finite compositions. Hence it is inconvenient to describe symmetry actions via these automorphisms.

The basic tool to study systems on infinite lattices is the operator algebra formalism, which is a generalization of the Heisenberg picture of the usual quantum mechanics. In this section, we review the operator algebra formalism. Our goal here is to introduce the essential aspects of this formalism to readers unfamiliar with it. We will not only give definitions and prove theorems, but also provide illuminating examples of various concepts.

### 1. Algebras of local and quasi-local operators

Here we introduce some general background of the algebras of local operators and quasi-local operators. Readers are referred to Refs. [44–47] for a more thorough treatment.

Throughout this section, we work on lattices of general spatial dimension $d$, i.e., our lattice is $\Lambda \simeq \mathbb{Z}^d$ unless otherwise specified. We assume that our on-site Hilbert space $\mathcal{H}_k$ is finite-dimensional, where $k$ labels a site in $\Lambda$

(we do *not* assume that $\dim \mathcal{H}_k$ is the same for different $k$'s). However, as we will see below, the "total" Hilbert space $\mathcal{H} := \bigotimes_{k \in \Lambda} \mathcal{H}_k$ is not well defined for infinite lattices. Especially, this naive infinite tensor space lacks a well-defined inner product, which is essential to the usual quantum mechanics. In more detail, let us say we have a "quantum state" in $\mathcal{H}$:

$$|\psi\rangle = \bigotimes_{k \in \Lambda} |\psi_k\rangle \tag{A.1}$$

where $|\psi_k\rangle \in \mathcal{H}_k$ is normalized vector for each $k \in \Lambda$. Given an arbitrary sequence $\{a_k\}_{k \in \Lambda}, a_k \in \mathbb{R}$, one can construct another "quantum state"

$$|\psi'\rangle := \bigotimes e^{ia_k} |\psi_k\rangle \tag{A.2}$$

The inner product between $|\psi\rangle$ and $|\psi'\rangle$ is

$$\langle\psi|\psi'\rangle = \exp(i \sum_{k \in \Lambda} a_k) \tag{A.3}$$

Since the sequence $\{a_k\}_{k \in \Lambda}$ is arbitrary and there is no obvious regularization scheme, this inner product has no definite answer. Thus $\mathcal{H}$ is not a well-defined object for quantum mechanics on infinite lattices. So one must be careful about the meaning of a quantum state on infinite lattices.

Nevertheless, the Heisenberg picture of quantum mechanics, which focuses on operators rather than states, is still applicable even for infinitely many degrees of freedom, such as spin systems on infinite lattices. Let us start with the notion of *local operators*. Given a *finite subset* $\Gamma \subset \Lambda$, one can talk about the Hilbert space $\mathcal{H}_\Gamma := \bigotimes_{k \in \Gamma} \mathcal{H}_k$ on $\Gamma$. The operator supported on $\Gamma$ is defined to be all operators on this finite dimensional Hilbert space $\mathcal{H}_\Gamma$. Note that any (finite) addition and multiplication of operators on $\mathcal{H}_\Gamma$ give another operator on $\mathcal{H}_\Gamma$, and hence these operators form an algebra, denoted by $\mathcal{A}_\Gamma^l$. We call $\mathcal{A}_\Gamma^l$ the algebra of local operators support on $\Gamma$. It is obvious that if $A \in \mathcal{A}_\Gamma^l$ then its Hermitian conjugate $A^\dagger \in \mathcal{A}_\Gamma^l$ as well.

Now we introduce a norm on this algebra. First, for a state $|\phi\rangle \in \mathcal{H}_\Gamma$, we define its norm as

$$\||\phi\rangle| = \sqrt{\langle\phi|\phi\rangle} \geqslant 0 \tag{A.4}$$

This norm satisfies the usual triangle inequality

$$\||\phi_1\rangle + |\phi_2\rangle| \leqslant \||\phi_1\rangle| + \||\phi_2\rangle| \tag{A.5}$$

Next, for a local operator $A \in \mathcal{A}_\Gamma^l$, we define its operator norm by

$$\|A\| := \sup_{\substack{|\psi\rangle \in \mathcal{H}_\Gamma \\ \langle\psi|\psi\rangle=1}} |A|\psi\rangle| \tag{A.6}$$

Namely, the norm of the operator $A$ is the square root of the largest eigenvalue of $A^\dagger A$ in the finite dimensional case.

By the definition in Eq. (A.6), for any vector $|\phi\rangle$ we have

$$|A|\phi\rangle|^2 \leqslant \|A\|^2 \||\phi\rangle|^2 \tag{A.7}$$

Another useful property of norms is the triangle inequality,

$$\|A_1 + A_2\| \leqslant \|A_1\| + \|A_2\|, \forall A_1, A_2 \in \mathcal{A}_\Gamma^l \tag{A.8}$$

for some finite subset $\Gamma$. To wit, note that for any $|\psi\rangle \in \mathcal{H}_\Gamma$, by Eq. (A.5),

$$|A_1|\psi\rangle + A_2|\psi\rangle| \leqslant |A_1|\psi\rangle| + |A_2|\psi\rangle| \tag{A.9}$$

The desired inequality Eq. (A.8) follows by taking supremes on both sides with respect to $\langle\psi|\psi\rangle = 1$.

Given any local operator $A \in \mathcal{A}_\Gamma^l$, if $\Gamma'$ is another finite subset containing $\Gamma$ (i.e., $\Gamma \subset \Gamma'$), then there is a natural way to extend $A$ to a local operator supported on $\Gamma'$,

$$\tilde{A} = A \bigotimes_{k \in \Gamma' \backslash \Gamma} I_k \tag{A.10}$$

where $\tilde{A}$ is the extension of $A$ on $\Gamma'$ and $I_k$ is the identity operator on $\mathcal{H}_k$. In this case, we say that $A$ acts as an identity outside $\Gamma$. It is often convenient to identify these two operators, i.e., we will not distinguish $\tilde{A}$ and $A$ in the following. One can easily check that $\|\tilde{A}\| = \|A\|$ so this identification is unambiguous on norms. We then make the following definition:

**Definition A.1.** *The algebra of local operators*

$$\mathcal{A}^l := \bigcup_{\Gamma \subset \Lambda, |\Gamma| < \infty} \mathcal{A}^l_\Gamma \tag{A.11}$$

*with the above identification $\tilde{A} \sim A$. We also define $\mathcal{A}^l_\emptyset = 0$.*

Here $|\Gamma|$ means the cardinality of $\Gamma$ and we write $|\Gamma| < \infty$ if $\Gamma$ is a finite set. More explicitly, $A \in \mathcal{A}^l$ if there is a finite subset $\Gamma$ such that $A \in \mathcal{A}^l_\Gamma$ (this $\Gamma$ is not unique due to the freedom to extend $A$). If $A_i \in \mathcal{A}^l, i = 1, 2$, then one can find a finite subset $\Gamma$ such that $A_i \in \mathcal{A}^l_\Gamma$. Because $\lambda_1 A_1 + \lambda_2 A_2 \in \mathcal{A}^l_\Gamma$, $\lambda_i \in \mathbb{C}, i = 1, 2$ and $A_1 A_2 \in \mathcal{A}^l_\Gamma$, $\mathcal{A}^l$ is an algebra. This algebra contains a special element, called a unit denoted by $I$, which satisfies $IA = AI = A$ for any $A \in \mathcal{A}^l$. In $\mathcal{A}^l$, this unit is given by identity operator which acts trivially on all sites.

One crucial property of $\mathcal{A}^l$ is locality. Given $A_i \in \mathcal{A}^l_{\Gamma_i}, i = 1, 2$. If $\Gamma_1 \cap \Gamma_2 = \emptyset$, then

$$[A_1, A_2] = 0 \tag{A.12}$$

We remark that $A_1, A_2$ in above equation really mean their extension on some $\Gamma$ which contains $\Gamma_1 \cup \Gamma_2$.

However, it is often insufficient to only work with $\mathcal{A}^l$. Consider a potentially non-on-site U(1) symmetry action, which is generated by its "conserved charge"[6],

$$Q = \sum_{|\Gamma| < \infty} Q_\Gamma \tag{A.13}$$

where $Q_\Gamma$ is some local term in the sense $Q_\Gamma = 0$ if $\mathrm{diam}(\Gamma) > R$ for some fixed $R > 0$, with $\mathrm{diam}(\Gamma) := \max_{x,y \in \Gamma} d(x, y)$ where $d$ is the distance. Given such a $Q$, the symmetry transformation is denoted by $\alpha_\theta$ for $\theta \in [0, 2\pi)$, which transforms local operator $A \in \mathcal{A}^l_\Gamma$ into

$$\alpha_\theta(A) = e^{i\theta Q} A e^{-i\theta Q} \tag{A.14}$$

Especially, we say that $Q$ is on-site if $Q_\Gamma = 0$ for all $|\Gamma| > 1$, and in this case the symmetry transformation generated by $Q$ is also said to be on-site.

As noted above, for a given $A \in \mathcal{A}^l_\Gamma$, if $B \in \mathcal{A}^l_{\Gamma'}$ is another local operator such that $\Gamma \cap \Gamma' = \emptyset$, then $[A, B] = 0$ by locality. However, $[\alpha_\theta(A), B] \neq 0$ in general and this commutator is estimated by the Lieb-Robinson bound[7] [48–51]

$$||[\alpha_\theta(A), B]|| \leqslant |\Gamma| C e^{-a(L - v\theta)} \tag{A.15}$$

where $C, a, v$ are non-universal positive constants[8] and $L := d(\Gamma, \Gamma')$ is the distance of $\Gamma$ and $\Gamma'$. The constant $v$ is usually called the Lieb-Robinson velocity. Actually, one version of the Lieb-Robinson bounds implies that $\alpha_\theta$ maps a quasi-local operator to a quasi-local operator (see theorem 3.2 and lemma 3.3 of Ref. [52]).

**Remark A.1.** *In quantum information theory, it is customary to call any generator like Eq. (A.13) of unitary transformations a Hamiltonian, and any transformation generated by a Hamiltonian a time evolution. We adopt these terminologies.*

To ensure that continuous symmetries can act on our operator algebra, one has to consider the *quasi-local* operator algebra $\mathcal{A}^{ql}$ (see below for definition), rather than $\mathcal{A}^l$ only. Intuitively, $\mathcal{A}^{ql}$ is obtained by taking sequential limits in $\mathcal{A}^l$. Concretely, an operator $A \in \mathcal{A}^{ql}$ if and only if there is a Cauchy sequence $A_j \in \mathcal{A}^l$ such that

$$A = \lim_{j \to \infty} A_j \tag{A.16}$$

By a Cauchy sequence, we mean that $\forall \epsilon > 0$, there exists $N \in \mathbb{Z}^{>0}$ such that $||A_j - A_{j'}|| < \epsilon$ for all $j, j' > N$, where $|| \cdot ||$ is the operator norm defined in Eq. (A.6). More precisely, one says that $\mathcal{A}^{ql}$ is the completion of $\mathcal{A}^l$ with respect to $|| \cdot ||$. An analogue for the relation between $A^{ql}$ and $\mathcal{A}^l$ is $\mathbb{R}$ and $\mathbb{Q}$. In the latter case, $\mathbb{R}$ can be

---

[6] It is in quote because the following expression involves infinite sum, so one needs a more careful definition for this operator and it will be discussed later. This paragraph is hence purely heuristic and motivating for more rigorous definitions later.

[7] There are many different versions of Lieb-Robinson bounds.

[8] It will be important later that all of these constants are independent of $|\Gamma'|$.

defined to be the limits of Cauchy sequences in $\mathbb{Q}$ (or the completion of $\mathbb{Q}$ with respect to usual absolute value of $\mathbb{Q}$) [53]. In essence, if $A \in \mathcal{A}^{ql}$, $A$ may not act as identity outside a finite region. However, it can be approximated by local operators with any desired accuracy. Besides, $\mathcal{A}^{ql}$ has a natural norm inheriting from the operator norm in Eq. (A.6) on $\mathcal{A}^l$. Explicitly, let $A \in \mathcal{A}^{ql}$ and $A_j \in \mathcal{A}^l$ be a sequence convergent to $A$, we define

$$||A|| := \lim_{j \to \infty} ||A_j|| \tag{A.17}$$

It is easy to check $||A||$ does not depend on the choice of the sequence $\{A_j\}_{j=1,2,\ldots}$. By this definition, the unit in $\mathcal{A}^{ql}$, i.e., the identity operator $I$, has norm 1.

To summarize,

**Definition A.2.** *The quasi-local operator algebra $\mathcal{A}^{ql}$ is defined to be the completion of $\mathcal{A}^l$ with respect to the operator norm in Eq. (A.6). We also denote the group of quasi-local unitary operators as $\mathcal{U}^{ql}$.*

Given a subset $\Gamma$ (which may be finite or not), we write $\mathcal{A}^{ql}_\Gamma$ for the algebra of quasi-local operators supported on $\Gamma$, i.e., it acts as identity outside of $\Gamma$.

Our $\mathcal{A}^{ql}$ is a special example of the so-called $C^*$-algebra in the mathematical literature [54–56].

**Definition A.3.** *A $C^*$-algebra $\mathcal{C}$ is an algebra equipped with an anti-linear involution $*$ (which models Hermitian conjugation in quantum mechanics) and a norm $|| \cdot ||$ (which may or may not be the operator norm defined in Eq. (A.6)), such that*

1. *$(A^*)^* = A$ for all $A \in \mathcal{C}$.*

2. *$(AB)^* = B^* A^*$ for all $A, B \in \mathcal{C}$.*

3. *$(\lambda A + B)^* = \bar{\lambda} A^* + B^*$ for all $A, B \in \mathcal{C}$, $\lambda \in \mathbb{C}$ and $\bar{\lambda}$ is the complex conjugate of $\lambda$.*

4. *(Banach property) $||AB|| \leqslant ||A|| \cdot ||B||$ and $||A^*|| = ||A||$ for all $A, B \in \mathcal{C}$.*

5. *($C^*$ property) $||A^* A|| = ||A||^2$*

**Example A.1.** *If $\mathcal{H}$ is a Hilbert space (not necessarily finite dimensional), then its algebra of bounded operators (i.e., operators with finite norms) $\mathcal{B}(\mathcal{H})$ is a $C^*$-algebra with $* = \dagger$. The first 3 properties are obvious. The Banach property is true according to Eq. (A.7)*

$$|AB|\psi\rangle| \leqslant ||A|| \cdot |B|\psi\rangle| \leqslant ||A|| \cdot ||B|| \cdot ||\psi\rangle| \tag{A.18}$$

*Hence $||AB|| \leqslant ||A|| \cdot ||B||$. For the $C^*$-property, again note that for a normalized $|\psi\rangle$*

$$|A|\psi\rangle|^2 = \langle\psi|A^\dagger A|\psi\rangle \leqslant ||A^\dagger A|| \tag{A.19}$$

*Thus $||A||^2 \leqslant ||A^\dagger A|| \leqslant ||A^\dagger|| \cdot ||A||$ by the Banach property. So we have $||A|| \leqslant ||A^\dagger||$. Note that $(A^\dagger)^\dagger = A$. Hence $||A|| = ||A^\dagger||$. The $C^*$-property then follows.*

**Example A.2.** *Our $\mathcal{A}^{ql}$ with $* = \dagger$ and the norm Eq. (A.17) is a $C^*$-algebra (see Sec. 3.2.3 of Ref. [46]). As a consequence, for any $U \in \mathcal{U}^{ql}$, we have $||U|| = 1$ since $||U||^2 = ||U^\dagger U|| = 1$.*

In this paper, we will focus on two examples of $C^*$-algebra, i.e., the algebra of quasi-local operators $\mathcal{A}^{ql}$ and the algebra of bounded operators $\mathcal{B}(\mathcal{H})$, but the discussions presented in this section apply to general $C^*$-algebras.

## 2. Quantum cellular automata and locality-preserving automorphisms

After introducing the basic notion of operator algebra, our next goal is to define a proper notion of symmetry action on local operators. This symmetry action is often required to preserve certain notion of locality. Given a symmetry group $G$, it is natural to define the symmetry action as a homomorphism $G \to \text{Aut}(\mathcal{A}^l)$, where $\text{Aut}(\mathcal{A}^l)$ is the automorphism group of $\mathcal{A}^l$. Recall that

**Definition A.4.** *We say $\alpha : \mathcal{A}^l \to \mathcal{A}^l$ $\alpha$ is an automorphism of $\mathcal{A}^l$, if*

1. *$\alpha(A + B) = \alpha(A) + \alpha(B)$*

*2. $\alpha(AB) = \alpha(A)\alpha(B)$*

*3. $\alpha(A^\dagger) = \alpha(A)^\dagger$*

*4. $\alpha(\lambda A) = \lambda\alpha(A), \forall \lambda \in \mathbb{C}$*

*5. $\alpha$ is invertible.*

*All automorphisms of $\mathcal{A}^l$ form a group under **finite** compositions, denoted by $\mathrm{Aut}(\mathcal{A}^l)$*

There are similar automorphism groups for other algebras, e.g., $\mathcal{A}^{ql}$ and $\mathcal{B}(\mathcal{H})$.
There is a special subgroup of $\mathrm{Aut}(\mathcal{A}^l)$ in the literature called quantum cellular automata [34, 35].

**Definition A.5.** *A quantum cellular automaton (QCA) is an automorphism $\alpha$ of $\mathcal{A}^l$ such that for a local operator $A \in \mathcal{A}^l_X$ (where $X$ is a finite subset), $\alpha(A) \in \mathcal{A}^l_{B(X,r_\alpha)}$ where $r_\alpha > 0$ does not depend on $A$.[9]*

From the definition, QCA preserves locality in the strongest sense and QCA form a group under finite compositions, which is denoted by $\mathcal{G}^{\mathrm{QCA}}$.

**Example A.3.** *One particular example of QCA is a (finite-depth unitary) circuit. For simplicity, we describe it in 1d. Let $\{P_k\}_{k\in\mathbb{Z}}$ be a set of disjoint intervals with $|P_k| < l$ for a constant length $l$ ($P_k$ can be empty for some $k$'s). Then, we define a block-partitioned unitary (BPU) as*

$$\alpha = \prod_{k=-\infty}^{\infty} \mathrm{Ad}_{U_k} \tag{A.20}$$

*where $U_k$ is a unitary operator supported on $P_k$ and $\mathrm{Ad}_{U_k}(A) = U_k A U_k^\dagger$ for local operator $A$. To see that a BPU is a QCA, let us assume $\mathrm{supp}(A) \subset P_k$ for some $k$. Thus $\alpha(A) \in P_k$ again has finite support. It is easy to generalize this argument to other local operators. Hence it is QCA by definition.*

*A circuit is a finite composition of BPU's (these BPU's may be defined for different partitions). The group of all circuits are denoted by $\mathcal{G}^{cir}$. Later we will see that not every QCA is a circuit.*

As explained before in the last subsection, one needs to consider not only QCA's, but the automorphism group $\mathrm{Aut}(\mathcal{A}^{ql})$ of the quasi-local operator algebra $\mathcal{A}^{ql}$ is also of fundamental importance.

**Definition A.6.** *An automorphism $\alpha$ of $\mathcal{A}^{ql}$ is called a locality-preserving automorphism if for each local operator $A \in \mathcal{A}^l_X$ and any $r > 0$, there exists a local operator $B$ such that*

$$||\alpha(A) - B|| < f_\alpha(r)||A|| \tag{A.21}$$

*where $f_\alpha(r)$ is a positive decreasing function independent of the choice of $A$ and $\lim_{r\to\infty} f_\alpha(r) = 0$.*
*The group of LPA's under finite composition is denoted by $\mathcal{G}^{lp}$*

More explicitly, if $\alpha \in \mathcal{G}^{lp}$ and $A \in \mathcal{A}^l_\Gamma$ is a local operator with $|\Gamma| < \infty$, then

$$\frac{||\alpha(A) - B||}{||A||} < f_\alpha(r) \tag{A.22}$$

In this situation, we say that $\alpha$ has an $f_\alpha$-tail[10]. Intuitively, $\alpha(A)$ is *not* a local operator any more, but it can be approximated by another local operator $B$ defined on a larger support $B(\Gamma, r)$ with an error controlled by $f_\alpha(r)$. We introduce another useful notation to deal with LPA's.

**Definition A.7.** *Given an operator $A \in \mathcal{A}^{ql}$ and a subalgebra $\mathcal{B}$ of $\mathcal{A}^{ql}$, fix a constant $\epsilon > 0$, we write $A \overset{\epsilon}{\in} \mathcal{B}$ if there is $B \in \mathcal{B}$ such that $||A - B|| \leqslant \epsilon||A||$. Similarly, for two subalgebras of $\mathcal{A}^{ql}$, we write $\mathcal{A} \overset{\epsilon}{\subset} \mathcal{B}$ if for $\forall A \in \mathcal{A}$ we have $A \overset{\epsilon}{\in} \mathcal{B}$.*

---

[9] Here $B(\Gamma, r_\alpha) := \{x \in \Lambda | d(x,\Gamma) \leqslant r_\alpha\}$ where $d$ is the distance on lattice and $0 \leqslant r_\alpha < \infty$ does not depend on $A$.
[10] The choice of $f_\alpha$ is not unique. Any other non-negative de-creasing function $h(r) \geqslant f(r)$ with $\lim_{r\to\infty} h(r) = 0$ also does the job. So for such $h_\alpha$, one can say $\alpha$ has $h_\alpha$-tail as well.

Thus, the condition Eq. (A.22) can be written as $\alpha(\mathcal{A}_\Gamma^l) \overset{f(r)}{\in} \mathcal{A}_{B(\Gamma,r)}^l$. The theorem 3.2 and lemma 3.3 of Ref. [52] give an easy criterion for an automorphism to be an LPA in 1d. In particular, a finite-time evolution generated by a local Hamiltonian is an LPA.

In order to study symmetry actions on lattice systems in more detail, one first needs the structure of $\mathcal{G}^{lp}$. In fact, the structures of $\mathcal{G}^{\mathrm{QCA}}$ and $\mathcal{G}^{lp}$ are rather clear in 1d,[11] thanks to the Gross-Nesme-Vogts-Werner (GNVW) index [33, 52]. Below we first give a brief introduction of the GNVW index of QCA's, and then review the GNVW index of LPA's. We do not provide a detailed construction of the GNVW index here since it is rather technical and we will not use it in any essential way. Interested readers are referred to Refs. [33–35, 52].

We assume that all on-site local Hilbert spaces $V$ have the same dimension $D$. This does not lose any generality, since it can always be achieved by tensoring the original degrees of freedom with some other degrees of freedom at each site, such that our QCA acts trivially on the additional degrees of freedom.

Roughly speaking, the GNVW index is a group homomorphism, denoted by

$$\mathrm{ind} : \mathcal{G}^{\mathrm{QCA}} \to \mathbb{Z}[\{\log(p_j)\}_{j\in J}] \tag{A.23}$$

where $\{p_j\}_{j\in J}$ is the set of all prime divisors of $D$.

The index map satisfies the following properties:

1. $\mathrm{ind}(\alpha\beta) = \mathrm{ind}(\alpha) + \mathrm{ind}(\beta), \forall\, \alpha, \beta \in \mathcal{G}^{\mathrm{QCA}}$,

2. $\ker(\mathrm{ind}) = \mathcal{G}^{cir}$,

3. $\mathrm{ind}$ is a surjection.

It can be verified that a circuit is a QCA with a vanishing GNVW index. On the other hand, one can verify that if $\tau$ is a shift on the lattice by $+1$

$$\mathrm{ind}(\tau) = \log D \in \mathbb{Z}[\{\log p_j\}_{j\in J}] \tag{A.24}$$

To show that ind is a surjection, note that $\mathbb{Z}[\{\log p_j\}_{j\in J}]$ is generated by $\log p_i$, it suffices to identify an element in $\mathcal{G}^{\mathrm{QCA}}$ whose index is $\log p_i$ for each $p_i$. To this end, one defines a generalized translation (or partial translation) which only shifts part of the degrees of freedom at each site and which has an index $\log p_i$. To be more precise, one can fix a $p_i$ (where $p_i$ is a prime divisor of $D$) dimensional subspace $V_i$ of $V$. One can factorize $V$ into $V \simeq V' \otimes V_i$ at each site, and a generalized translation $\tau_i$ only shifts the $V_i$-part while fixing $V'$. The GNVW index of $\tau_i$ is then verified to be

$$\mathrm{ind}(\tau_i) = \log p_i \tag{A.25}$$

Generalized translations form an Abelian group under finite compositions, and we denote it by $\mathcal{G}^T$. Thus, the GNVW index actually shows that $\mathcal{G}^{\mathrm{QCA}}$ is a semi-direct product

$$\mathcal{G}^{\mathrm{QCA}} = \mathcal{G}^{cir} \rtimes \mathcal{G}^T \tag{A.26}$$

Now we turn to the structure theory of LPA's. Recall that $\mathcal{A}^{ql}$ is obtained by taking limits of $\mathcal{A}^l$, or more formally, any quasi-local operator can be approximated by local operators. Any LPA can also be approximated by QCA's. The basic strategy to study LPA's is to approximate it by a sequence of QCA's. For example, if $\alpha \in \mathcal{G}^{lp}$, there exists a sequence $\{\beta_j\}_{j=1,2\ldots}$ of QCA's such that

$$\lim_{j\to\infty} \beta_j = \alpha \tag{A.27}$$

Then one defines the GNVW index of $\alpha$ as

$$\mathrm{ind}(\alpha) = \lim_{j\to\infty} \mathrm{ind}(\beta_j) \tag{A.28}$$

It can be checked that this index is well-defined (finite and independent of the choice of the sequence) [52]. Given the existence of this index map, many results of QCA's can be carried over to LPA's. For example,

$$\mathcal{G}^{lp} = \mathcal{G}^{loc} \rtimes \mathcal{G}^T \tag{A.29}$$

where $\mathcal{G}^{loc}$ is the subgroup of time evolution generated by local Hamiltonians (*which can be time-dependent*) over a finite duration.

––––––––

[11] Some classifications of higher dimensional QCA's are also pro-   posed recently, see e.g., Ref. [57].

### 3. States in the operator algebra formalism

In the above sections, we have introduced the algebra of operators, and the concepts of QCA and LPA to describe symmetry actions on local operators. In this section, we discuss the notion of states in the operator algebra formalism, in the context of infinite systems.

In quantum mechanics, a state $|\psi\rangle$ is a vector in some Hilbert space $\mathcal{H}$. Equivalently, this state can be represented by a density matrix $\rho_\psi = |\psi\rangle\langle\psi|$. Another slightly unusual point of view is to think of $\rho_\psi$ as a linear functional $\psi$ on the algebra of operators,

$$\psi(A) := \mathrm{tr}(\rho_\psi A) \tag{A.30}$$

for any operator $A$ in $\mathcal{H}$. Once $\psi$ is determined, the density matrix $\rho_\psi$ (hence the vector state $|\psi\rangle$) is also determined[12]. The linear functional corresponding to a quantum state must satisfy further properties, such as positivity

$$\psi(A^\dagger A) = \langle\psi|A^\dagger A|\psi\rangle \geqslant 0 \tag{A.31}$$

for any operator $A$ in $\mathcal{H}$. Besides, if $|\psi\rangle \neq 0$, one requires it to be normalized, that is

$$\psi(I) = \langle\psi|\psi\rangle = 1 \tag{A.32}$$

This motivates the following definition of states in infinite systems.

**Definition A.8.** *A quantum state of quasi-local operator algebra $\mathcal{A}^{ql}$ is a linear functional $\psi : \mathcal{A}^{ql} \to \mathbb{C}$ with the following properties:*

1. *Positivity: $\psi(A^\dagger A) \geqslant 0$ for all $A \in \mathcal{A}^{ql}$.*

2. *Normalization[13]: $\psi(I) = 1$ for nonzero[14] $\psi$.*

**Example A.4.** *Consider the algebra $\mathcal{B}(\mathcal{H})$ (see example A.1 for its definition) with $\dim \mathcal{H} < \infty$. Let $\psi$ be a state of $\mathcal{B}(\mathcal{H})$. Pick up an orthonormal basis $|e_i\rangle$ of $\mathcal{H}$, we define*

$$\begin{aligned} P_{ij} &:= |e_i\rangle\langle e_j| \in \mathcal{B}(\mathcal{H}) \\ \lambda_{ij} &:= \psi(P_{ij}) \end{aligned} \tag{A.33}$$

*Note that $P_{ij}^\dagger = P_{ji}$ and $P_{ij}P_{kl} = \delta_{jk}P_{il}$, where $\delta_{jk}$ is usual Kronecker delta. One can define a density matrix*

$$\rho_\psi = \sum_{i,j} \lambda_{ij} P_{ij} \tag{A.34}$$

*For any $A \in \mathcal{B}(\mathcal{H})$, we have*

$$\psi(A) = \mathrm{tr}(\rho_\psi A) \tag{A.35}$$

*Hence the abstract state $\psi$ is represented by the $\rho_\psi$ and one can check that $\rho_\psi$ is a density matrix indeed.*

The above example shows how to connect a quantum state in the language of operator algebra with a quantum state in terms of a usual density matrix. Below we discuss some basic properties of states, such as the norm of states, Cauchy-Schwarz inequality and orthogonality of states. This part can be safely skipped for a first reading. Readers only interested in the proof of our main theorems can move forward to definition A.10.

The positivity actually implies that $\psi(A) \geqslant 0$ for any positive operator $A$ (i.e., a self-adjoint operator whose eigenvalues are all non-negative). To see it, if $A$ is a positive local operator, then by linear algebra, there is a unique positive square root $\sqrt{A}$ of $A$. Thus $\psi(A) = \psi((\sqrt{A})^\dagger\sqrt{A}) \geqslant 0$ indeed. If $A$ is a positive quasi-local operator, one can find a sequence of positive operators $A_j \in \mathcal{A}^l$ and we define $\sqrt{A} := \lim_{j\to\infty} \sqrt{A_j}$. In this case,

---

[12] To see this, one chooses $A$ to be projectors to basis vectors, which is enough to reconstruct $|\psi\rangle$ up to an overall phase.

[13] For non-unital $C^*$-algebra, one can embed it into another unital $C^*$-algebra, see Sec. 2.2 of Ref. [46]. Hence this definition still works.

[14] A state $\psi$ is said to be zero if $\psi(A) = 0$ for all $A \in \mathcal{A}^{ql}$.

we again have $\psi(A) = \psi((\sqrt{A})^\dagger \sqrt{A}) \geqslant 0$. More generally, any linear functional $f : \mathcal{A}^{ql} \to \mathbb{C}$ is *positive* if $f(A) \geqslant 0$ for any positive operator $A \in \mathcal{A}^{ql}$.

It is also useful to define a norm on the space of linear functionals on $\mathcal{A}^{ql}$. Given a linear functional $f : \mathcal{A}^{ql} \to \mathbb{C}$, its norm is defined by

$$||f|| := \sup_{\substack{A \in \mathcal{A}^{ql}, \\ ||A|| = 1}} |f(A)| \tag{A.36}$$

As a norm, it satisfies the triangle inequality

$$||f_1 + f_2|| \leqslant ||f_1|| + ||f_2|| \tag{A.37}$$

To wit, note that for any $|f_1(A) + f_2(A)| \leqslant |f_1(A)| + |f_2(A)|$ for any $A$. The desired inequality follows by taking supremes on both sides with respect to $||A|| = 1$. Notice this is the triangle inequality of the functional norms, and previously we have proved the triangle inequality for operator norms in Eq. (A.8).

The corollary below will be useful.

**Corollary A.1.** *The following properties about state $\psi$ and positive linear functionals are true.*

1. *$\psi(A^\dagger) = \psi(A)^*$ for all $A \in \mathcal{A}^{ql}$, where $z^*$ means the complex conjugation of the complex number $z$.*

2. *Cauchy-Schwarz inequality: $|\psi(B^\dagger A)|^2 \leqslant \psi(B^\dagger B)\psi(A^\dagger A)$.*

3. *If $\psi$ is nonzero, then $||\psi|| = 1$. More generally, if $f$ is a positive linear functional, then $||f|| = f(I)$ where $I$ is the identity.*

4. *If $f_1, f_2$ are positive linear functionals, then $||f_1 + f_2|| = ||f_1|| + ||f_2||$.*

*Proof.* For the first property, note that each operator $A$ has the following decomposition

$$A = \frac{A + A^\dagger}{2} + i\frac{A - A^\dagger}{2i} \tag{A.38}$$

So we only have to show that $\psi(A) \in \mathbb{R}$ if $A$ is Hermitian. Assume $A$ is Hermitian. Note

$$\psi(A + \lambda I) = \psi(A) + \lambda, \ \forall \lambda \in \mathbb{C} \tag{A.39}$$

So without loss of generality, we can assume $A$ is positive by shifting $A \to A + \lambda I$ with $\lambda \geqslant 0$. In this case, $A$ has a square root $\sqrt{A}$ which is again positive. Then

$$\psi(A) = \psi((\sqrt{A})^\dagger \sqrt{A}) \geqslant 0 \tag{A.40}$$

In particular, $\psi(A) \in \mathbb{R}$ and the first property follows.

The second property (Cauchy-Schwarz inequality) can be proved as in the usual quantum mechanics. Consider

$$F(\lambda) := \psi((A + \lambda B)^\dagger (A + \lambda B)) = \psi(A^\dagger A) + \lambda\psi(A^\dagger B) + \lambda^*\psi(B^\dagger A) + |\lambda|^2\psi(B^\dagger B) \geqslant 0 \tag{A.41}$$

for any $\lambda \in \mathbb{C}$. This quadratic function $F(\lambda)$ has to be positive definite, which results in the desired inequality.

To see the third property, we will use the definition in Eq. (A.36), which requires us to maximize $|\psi(A)|$ over all $A \in \mathcal{A}^{ql}$ with $||A|| = 1$. To this end, we first note that we only have to restrict to self-adjoint operators $A$ by the Cauchy-Schwarz inequality,

$$|\psi(A)|^2 = |\psi(I^\dagger A)|^2 \leqslant \psi(A^\dagger A) \tag{A.42}$$

so we have $\sup_{||A||=1} |\psi(A)| \leqslant \sup_{||A||=1} \sqrt{\psi(A^\dagger A)}$. Below we show that $\sup_{||A||=1} \sqrt{\psi(A^\dagger A)} = 1$. Since $1 = \psi(I) \leqslant ||\psi|| \leqslant 1$, we then deduce that $||\psi|| = 1$.

To show that $\sup_{||A||=1} \sqrt{\psi(A^\dagger A)} = 1$, note that $A^\dagger A$ is self-adjoint and $||A^\dagger A|| = ||A||^2 = 1$ by the $C^*$-property. So to maximize $\psi(A^\dagger A)$ with $||A|| = 1$ amounts to maximizing $|\psi(T)|$ for *positive* $T$ with $||T|| = 1$, where $T = A^\dagger A$. For later convenience, we define a partial order on self-adjoint operators. Let $T$ and $B$ be two self-adjoint operators, we write $T \geqslant B$ if $T - B$ is positive. Since $\psi$ is a positive linear functional, this implies that $\psi(T - B) \geqslant 0$. Now note that for all self-adjoint $T \in \mathcal{A}^{ql}$, we have

$$-||T||I \leqslant T \leqslant ||T||I \tag{A.43}$$

Applying $\psi$, we deduce

$$-||T|| \leqslant \psi(T) \leqslant ||T|| \tag{A.44}$$

This means that $\sup_{||T||=1} |\psi(T)| \leqslant 1$ for positive $T$. However, $|\psi(I)| = 1$, hence $\sup_{||T||=1} |\psi(T)| = 1$. Therefore, we conclude that $||\psi|| = 1$. For more general positive linear functional $f$, we normalize it as $\psi := f/f(I)$ which is a state. Hence we have $||f|| = f(I)||\psi|| = f(I)$.

For the last property, note that

$$f_1(I) + f_2(I) \leqslant ||f_1 + f_2|| \leqslant ||f_1|| + ||f_2|| = f_1(I) + f_2(I) \tag{A.45}$$

The second "$\leqslant$" above used Eq. (A.37). Hence $||f_1 + f_2|| = ||f_1|| + ||f_2||$. $\qquad \square$

Note that for any two quantum states $\psi_1, \psi_2$, according to Eq. (A.37), we have

$$||\psi_1 - \psi_2|| \leqslant ||\psi_1|| + ||\psi_1|| = 2 \tag{A.46}$$

If this bound is saturated, which means these states are separated as far as possible, we say $\psi_1$ is orthogonal to $\psi_2$.

**Definition A.9.** *Two states $\psi_1, \psi_2$ are orthogonal if*

$$||\psi_1 - \psi_2|| = 2 \tag{A.47}$$

**Remark A.2.** *In Ref. [44], a different notion of orthogonality is used, which we call independence of states, see definition A.13. We avoid that usage of orthogonality because in that definition, any two different pure states are orthogonal even for $\mathcal{B}(\mathcal{H})$ with $\dim(\mathcal{H}) < \infty$. That means, their definition cannot reduce to the usual orthogonality in quantum mechanics.*

**Example A.5.** *Now we show that this definition reduces to the usual notion of orthogonality in quantum mechanics for $\mathcal{C} = \mathcal{B}(\mathcal{H})$ (where $\mathcal{H}$ is a finite dimensional Hilbert space). In this case, we represent these states by vectors in $\mathcal{H}$,*

$$\psi_1(A) - \psi_2(A) = \langle \psi_1 | A | \psi_1 \rangle - \langle \psi_2 | A | \psi_2 \rangle \tag{A.48}$$

*If $\langle \psi_1 | \psi_2 \rangle = 0$, i.e., they are orthogonal to each other in the usual sense, take*

$$A = |\psi_1\rangle\langle\psi_1| - |\psi_2\rangle\langle\psi_2| \tag{A.49}$$

*Note that $||A|| = 1$ and $\psi_1(A) - \psi_2(A) = 2$, thus we conclude that $||\psi_1 - \psi_2|| = 2$ indeed, provided that these states are orthogonal in the usual sense. More generally if $\langle \psi_1 | \psi_2 \rangle \neq 0$, with some linear algebra, one can show that*

$$||\psi_1 - \psi_2|| = 2(1 - |\langle \psi_1 | \psi_2 \rangle|^2) \tag{A.50}$$

*So conversely, $||\psi_1 - \psi_2|| = 2$ also implies that $\langle \psi_1 | \psi_2 \rangle = 0$.*

Given two quantum states $\psi_0, \psi_1$ and a real number $t \in [0, 1]$, one can construct another state

$$\psi_t = t\psi_1 + (1 - t)\psi_0 \tag{A.51}$$

It is easy to check $\psi_t$ is positive as well as normalized.

**Definition A.10.** *A state $\psi$ is called mixed if there exists $\psi_0, \psi_1$ and $0 < t < 1$ such that $\psi = \psi_t$. A state is pure if it is not mixed.*

In fact, this definition is equivalent to mixed state in quantum mechanics if there is only finitely many degrees of freedom. Besides, one can show that for $C^*$-algebra $\mathcal{B}(\mathcal{H})$ where $\mathcal{H}$ is a finite dimensional Hilbert space, a pure state defined in definition A.10 coincides with usual pure states, see Sec. 2.3 of Ref. [47].

There are many equivalent definitions of pure states, the following is often useful.

**Definition A.11** (Alternative definition for pure states). *A state $\psi$ of a $C^*$-algebra $\mathcal{C}$ is pure if and only if for any positive linear functional $\rho : \mathcal{C} \to \mathbb{C}$ (may not be normalized) majorized by $\psi$ (i.e., $\psi - \rho$ is again positive), we have $\rho = t\psi$ for some $t \in [0, 1]$.*

**Lemma A.1.** *Pure states defined by definitions A.10 and A.11 are equivalent.*

One can easily check this for $\mathcal{B}(\mathcal{H})$ with finite dimensional $\mathcal{H}$, where states are represented by density matrices. Thus lemma A.1 becomes obvious by diagonalizing this density matrix. For a more general proof, see theorem 2.3.15 of Ref. [44].

## 4. Gelfand-Naimark-Segal construction

Now we do have the notion of states in the context of operator algebra, then it is tempting to talk about the Hilbert space. Indeed, working with Hilbert spaces has benefits. For example, abstract states in definition A.8 do not form a linear space, which means we lack one of the most important ingredients in quantum mechanics, i.e., the coherent superposition of states (see remark A.4 for details). Moreover, it is often easier to work with matrices rather than abstract algebra of operators. So we want to represent our algebra $\mathcal{A}^{ql}$ on some Hilbert space. Furthermore, some usual notions of representation theory, such as irreducible representations and Schur's lemma can help us further decompose these matrices into simpler pieces, i.e., making these matrices block-diagonalized.

Recall what we have in hand is states and the algebra of (quasi-)local operators. In quantum many-body physics, the Hilbert space can be built up by applying local operators to a fixed reference state. There is a similar way to build up a Hilbert space in $C^*$-algebra, known as the Gelfand-Naimark-Segal (GNS) construction. We now present this construction for general $C^*$-algebra $\mathcal{C}$, but readers can keep only $\mathcal{B}(\mathcal{H})$ or $\mathcal{A}^{ql}$ in mind.

Let us fix a state $\psi$ which can be pure or mixed. We start with a *pure* state for simplicity. We define the GNS ideal $N_\psi$ as

$$N_\psi := \{A \in \mathcal{C} | \psi(A^*A) = 0\} \tag{A.52}$$

Remember that $\psi$ is now our input state (informally one can think it as $|\psi\rangle$ since it is pure and $N_\psi$ includes operators which annihilate $|\psi\rangle$). If $A \in N_\psi$ (i.e., $A$ annihilates $|\psi\rangle$), then $BA$ also annihilates $|\psi\rangle$. This means $BA$ is again in $N_\psi$. Similarly if $A, B \in N_\psi$ then $A + B \in N_\psi$. Thus mathematically we say that $N_\psi$ is a left ideal. We will make the relation between the abstract state $\psi$ and the vector state $|\psi\rangle$ precise in Eq. (A.55).

The GNS Hilbert space $\mathcal{H}_\psi$ is defined to be[15]

$$\mathcal{H}_\psi := \mathcal{C}/N_\psi \tag{A.53}$$

where the equivalence is defined as $A \sim B$ if $A - B \in N_\psi$ (that is, elements in $N_\psi$ is identified as 0). We denote the equivalence class of operator $A$ as $[A]$. The $\mathcal{H}_\psi$ defined above is a Hilbert space, where each vector in this space is an equivalence class of operators, the addition of vectors is inherent from the addition of these operators, and the inner product is given by $\langle [A], [B] \rangle := \psi(A^*B)$. More formally, we have defined a representation of $\mathcal{C}$ on $\mathcal{H}_\psi$ as follows

$$\pi_\psi(A)[B] := [AB] \tag{A.54}$$

The last ingredient in the GNS construction is a reference state $|\psi\rangle$. Given an abstract state $\psi$, we say a state $|\psi\rangle \in \mathcal{H}_\psi$ is a reference state representing $\psi$, if

$$\psi(A) = \langle \psi | \pi_\psi(A) | \psi \rangle, \forall A \in \mathcal{C} \tag{A.55}$$

We note that $|\psi\rangle = [I]$ provides a possible choice satisfying this equation, where $[I]$ means the equivalence class of the identity operator in above quotient $\mathcal{H}_\psi = \mathcal{C}/N_\psi$. So such a representative does exist.

**Definition A.12.** *The above constructed $\mathcal{H}_\psi$ (Eq. (A.53)), $\pi_\psi$ (Eq. (A.54)) together with the state $|\psi\rangle$ is called a GNS triple, denoted by $(\pi_\psi, \mathcal{H}_\psi, |\psi\rangle)$. In mathematics, this $|\psi\rangle$ is called a cyclic vector.*

However, it is important to note that the representative $|\psi\rangle$ in Hilbert space $\mathcal{H}_\psi$ of an abstract state $\psi$ is not unique. For example, we can replace it with $|\Psi\rangle = U|\psi\rangle$ and work with

$$\pi(A) := U\pi_\psi(A)U^{-1} \tag{A.56}$$

for any unitary operator $U \in \mathcal{B}(\mathcal{H}_\psi)$. We emphasize that this unitary operator $U$ is only defined on the particular GNS Hilbert space $\mathcal{H}_\psi$. This also defines a GNS triple $(\pi, \mathcal{H}_\psi, |\Psi\rangle)$. It turns out that given a state $\psi$, the GNS representation is unique up to unitary equivalence.

**Corollary A.2** (Theorem 2.5.3 of Ref. [46]). *GNS triple is unique in the following sense. Suppose that $\psi$ is a state of $\mathcal{A}^{ql}$, $(\pi_\psi, \mathcal{H}_\psi, |\psi\rangle)$ and $(\pi, \mathcal{H}_\psi, |\Psi\rangle)$ are two different GNS triples associated to $\psi$, then there exists a unitary operator $U \in \mathcal{B}(\mathcal{H}_\psi)$ such that*

$$U|\psi\rangle = |\Psi\rangle$$
$$U\pi_\psi(A)U^{-1} = \pi(A) \tag{A.57}$$

———————

[15] Actually, this quotient space is often incomplete (i.e., it is not well-behaved when taking limits) for infinite dimensional $\mathcal{C}$, so one needs to do further completion. We omit this detail.

One can think of the GNS construction in this case as a certain kind of *state-operator correspondence*, since all states in $\mathcal{H}_\psi$ can be obtained by applying $\pi_\psi(A)$ to the reference state $|\psi\rangle$ for some $A \in \mathcal{C}$. Thus, each state corresponds to certain equivalence class of states in $\mathcal{C}/N_\psi$.

Notice the above definition of the inner product in $\mathcal{H}_\psi$ indicates what it means for two vectors in $\mathcal{H}_\psi$ to be orthogonal. However, this orthogonality is in general unrelated to the orthogonality defined in definition A.9.

**Remark A.3.** *One may wonder, if we start with a pure state of $\mathcal{A}^{ql}$, as explained in the beginning of Sec. A 1, there is no Hilbert space for an infinite spin chain, then what is the GNS Hilbert space on earth? As we will see in Sec. A 5 (in particular, example A.10), the GNS Hilbert space only describes a superselection sector of our model.*

**Remark A.4.** *So far we have only defined the convex combinations of abstract states (i.e., normalized positive linear functionals), see Eq. (A.51), which model classical mixtures, rather than quantum coherent superpositions in quantum mechanics. In the operator algebra formalism applied to infinite systems, we can only talk about coherent superpositions within a GNS Hilbert space, which is given by the usual vector addition.*

The definition of GNS triple definition A.12 also applies to mixed states. To see the structure of the GNS Hilbert space $\mathcal{H}_\psi$ for a mixed state $\psi$ intuitively, note that one can always decompose a mixed state into a convex linear combinations of pure states (see Eq. (A.65) below) and then we build GNS Hilbert spaces for each pure states separately. The space $\mathcal{H}_\psi$ is the direct sum of these Hilbert spaces. Below we discuss the detailed structure of the GNS Hilbert space for mixed states. Readers can skip this discussion for a first reading and move forward to example A.8.

To study the GNS Hilbert space of $\mathcal{H}_\psi$ for mixed state $\psi$ in more detail, we need a notion called independent decomposition. To begin, we need the notion of independence of positive linear functionals (see the paragraph below definition A.8 for the definition of positive linear functionals).

**Definition A.13.** *Given two positive linear functionals $f_1, f_2 : \mathcal{A}^{ql} \to \mathbb{C}$, if there exists no nonzero positive linear functional $f$ such that $f_1 - f$ and $f_2 - f$ are again positive, then we say $f_1$ and $f_2$ are independent.*

**Example A.6.** *By definition A.11, any two different nonzero pure states $\psi_1, \psi_2$ are independent. To wit, suppose they are not independent, then there is a nonzero positive linear functional $f$ in definition A.13. But if $\psi_1 - f$ is positive, then $f = \lambda_1 \psi_1, 0 < \lambda_1 \leqslant 1$ since $\psi_1$ is pure. Similarly, $f = \lambda_2 \psi_2, 0 < \lambda_2 \leqslant 1$. Taking norms of these equations and noting that $||\psi_1|| = ||\psi_2|| = 1$ (see the third property of corollary A.1), we find $\lambda_1 = \lambda_2$ and hence $\psi_1 = \psi_2$, contradicting to our assumption.*

*Moreover, orthogonal states (in the sense of definition A.9) are independent. Let $\psi_1$ and $\psi_2$ be orthogonal states (not necessarily pure) and $f$ be a positive linear functional such that $\psi_1 - f$ and $\psi_2 - f$ are positive. By the third property of corollary A.1,*

$$||\psi_1 - f|| = \psi_1(I) - f(I) \leqslant ||\psi_1|| = 1 \tag{A.58}$$

*Similarly, $||\psi_2 - f|| \leqslant ||\psi_2|| = 1$. On the other hand,*

$$2 = ||\psi_1 - \psi_2|| \leqslant ||\psi_1 - f|| + ||f - \psi_2|| \leqslant ||\psi_1|| + ||\psi_2|| = 2 \tag{A.59}$$

*Note the equality holds only if $||\psi_1 - f|| = ||\psi_2 - f|| = 1$. However, by the fourth property in corollary A.1,*

$$1 = ||\psi_1|| = ||\psi_1 - f|| + ||f|| \tag{A.60}$$

*which implies $||f|| = 0$ and thus $f = 0$. Therefore, $\psi_1$ and $\psi_2$ are independent.*

**Example A.7.** *We also give an example where two linear functionals are **not** independent. By abusing terms, we say that two density matrices $\rho_1, \rho_2$ are independent if the associated linear functionals $\mathrm{tr}(\rho_1 \cdot), \mathrm{tr}(\rho_2 \cdot)$ are independent. Consider the following two density matrices in qubit system*

$$\rho_1 = \frac{1}{2}(|\uparrow\rangle\langle\uparrow| + |\downarrow\rangle\langle\downarrow|)$$
$$\rho_2 = |+\rangle\langle+| \tag{A.61}$$

*where $|\uparrow\rangle, |\downarrow\rangle$ are orthonormal basis on $\mathcal{H} = \mathbb{C}^2$ and $|+\rangle := \frac{1}{\sqrt{2}}(|\uparrow\rangle + |\downarrow\rangle)$. Note*

$$\rho_1 - \frac{1}{2}|+\rangle\langle+|, \ \rho_2 - \frac{1}{2}|+\rangle\langle+| \tag{A.62}$$

*are positive and $\frac{1}{2}|+\rangle\langle+|$ corresponds to a positive linear function. Hence by definition A.13 $\rho_1$ and $\rho_2$ are not independent.*

The importance of independent states lies in the following theorem,

**Theorem A.1** (Lemma 4.1.19 of Ref. [44]). *If $\psi_0, \psi_1$ are independent states, and*

$$\psi := t\psi_1 + (1-t)\psi_0, 0 < t < 1 \tag{A.63}$$

*then the GNS representation decomposes as*

$$\pi_\psi = \pi_{\psi_0} \oplus \pi_{\psi_1} \tag{A.64}$$

*The converse is also true.*

So the structure of the GNS representation of a mixed state $\psi$ is clear if we can decompose it into independent states. This decomposition is indeed possible due to the following theorem.

**Theorem A.2** (Theorem 4.4.9 of Ref. [44]). *Any state $\psi$ admits an independent decomposition*[16]

$$\psi = \sum_{i \in I} \lambda_i \psi_i \tag{A.65}$$

*where $I$ is an index set, $\psi_i$ is pure for each $i \in I$ and $0 < \lambda_i \leqslant 1$ with $\sum_{i \in I} \lambda_i = 1$. This decomposition is independent in the sense that, for any subset $J \subset I$, $\psi_J := \sum_{i \in J} \lambda_i \psi_i$ is independent of $\psi_{J^c} = \sum_{i \in I \setminus J} \lambda_i \psi_i$. Besides, the GNS representation $\pi_\psi$ is also decomposed as*

$$\pi_\psi = \bigoplus_{i \in I} \pi_{\psi_i} \tag{A.66}$$

*That means, the representation matrices $\pi_\psi(A)$ can be simultaneously block-diagonalized for all $A \in \mathcal{A}^{ql}$.*

**Proposition A.1.** *One says that a GNS representation $\pi_\psi$ is irreducible if Eq. (A.66) has only 1 direct summand, i.e., the state $\psi$ is pure.*

Given the notion of irreducibility, the usual Schur's lemma follows.

**Lemma A.2** (Schur). *For a GNS representation $(\pi_\psi, \mathcal{H}_\psi |\psi\rangle)$ of $C^*$-algebra $\mathcal{C}$, it is irreducible if and only if, for each $T \in \mathcal{B}(\mathcal{H}_\psi)$ commuting with all $\pi_\psi(\mathcal{C})$, we have $T = \lambda I$ for some $\lambda \in \mathbb{C}$.*

Now we are ready to give some examples.

**Example A.8.** *Consider a qubit which lives in $\mathcal{H} = \mathbb{C}^2$. The operator algebra is $M_2(\mathbb{C})$, the 2 by 2 matrix algebra over $\mathbb{C}$. We begin with the following state,*

$$\psi(A) = A_{11} \tag{A.67}$$

*where $A_{11}$ is the first matrix element of A. We denote a basis of $M_2(\mathbb{C})$ as*

$$E_{11} = \begin{pmatrix} 1 & 0 \\ 0 & 0 \end{pmatrix}, E_{12} = \begin{pmatrix} 0 & 1 \\ 0 & 0 \end{pmatrix}$$
$$E_{21} = \begin{pmatrix} 0 & 0 \\ 1 & 0 \end{pmatrix}, E_{22} = \begin{pmatrix} 0 & 0 \\ 0 & 1 \end{pmatrix} \tag{A.68}$$

*Then, it is easy to check*

$$\psi(E_{11}^\dagger E_{11}) = 1, \psi(E_{12}^\dagger E_{12}) = 0$$
$$\psi(E_{21}^\dagger E_{21}) = 1, \psi(E_{22}^\dagger E_{22}) = 0 \tag{A.69}$$

---

[16] It can happen that $\psi$ is decomposed into uncountably many pure states. In that case, the right hand side of Eq. (A.65) is replaced by a suitable integral (see Ref. [44]).

*Hence* $N_\psi = \mathrm{span}\{E_{12}, E_{22}\}$, *or equivalently* $E_{12}, E_{22}$ *annihilate* $\psi$. *Thus, reassuringly,* $\mathcal{H}_\psi = M_2(\mathbb{C})/N_\psi \simeq \mathbb{C}^2$ *is exactly the space we start with! Besides, under the basis* $[E_{11}], [E_{21}]$ *(or* $|E_{11}\rangle, |E_{21}\rangle$ *if one prefers) the representation* $\pi_\psi$ *is given by*

$$
\pi_\psi(E_{11}) = \begin{pmatrix} 1 & 0 \\ 0 & 0 \end{pmatrix}, \pi_\psi(E_{12}) = \begin{pmatrix} 0 & 1 \\ 0 & 0 \end{pmatrix}
$$
$$
\pi_\psi(E_{21}) = \begin{pmatrix} 0 & 0 \\ 1 & 0 \end{pmatrix}, \pi_\psi(E_{22}) = \begin{pmatrix} 0 & 0 \\ 0 & 1 \end{pmatrix}
\tag{A.70}
$$

*That is, these operators are represented in the standard way.*

*On the other hand, if we consider*

$$
\rho(A) = tA_{11} + (1 - t)A_{22}
\tag{A.71}
$$

*for some* $0 < t < 1$. *Then one can calculate*

$$
\rho(E_{11}^\dagger E_{11}) = t, \rho(E_{12}^\dagger E_{12}) = 1 - t
$$
$$
\rho(E_{21}^\dagger E_{21}) = t, \rho(E_{22}^\dagger E_{22}) = 1 - t
\tag{A.72}
$$

*Thus* $N_\rho = 0$ *and* $\mathcal{H}_\rho \simeq M_2(\mathbb{C})/N_\rho \simeq \mathbb{C}^4 \simeq \mathbb{C}^2 \oplus \mathbb{C}^2$.

*However, consider the following state*

$$
\omega = \mathrm{tr}(\Omega \cdot)
\tag{A.73}
$$

*where* $\Omega = \frac{1}{3}(|\uparrow\rangle\langle\uparrow| + |\downarrow\rangle\langle\downarrow| + |+\rangle\langle+|)$ *is the density matrix representing* $\omega$ *(see example A.7). This state contains 3 pure states in the decomposition. Naively we will get 3 copies of* $\mathbb{C}^2$, *so do we have* $\mathcal{H}_\omega = \mathbb{C}^6$? *However, we cannot get a 6-dimensional Hilbert space since* $\mathcal{B}(\mathcal{H})$ *is 4-dimensional after all. This is because this decomposition of* $\Omega$ *above is not independent, as is checked in example A.7.*

*Another example will be the Ising chain.*

**Example A.9.** *In this example, we have an on-site Hilbert space* $\mathcal{H}_k \simeq \mathbb{C}^2$ *for each site* $k \in \mathbb{Z}$, *and* $X, Y, Z$ *will be the usual Pauli operators. We consider the all-spin-up state* $\psi_+$ *as the reference state. More suggestively, we write it as* $|\uparrow\rangle$.

*Note that* $\psi_+((Z_k - 1)^\dagger(Z_k - 1)) = 0$, *and it is easy to see the left ideal* $N_{\psi_+}$ *is generated by* $(Z_k - 1)$ *for all* $k \in \mathbb{Z}$, *which means if* $A \in N_{\psi_+}$, *there exists* $B \in \mathcal{A}^{ql}$ *such that*

$$
A = B(Z_k - 1)
\tag{A.74}
$$

*for some* $k \in \mathbb{Z}$. *We claim,*

$$
\mathcal{H}_{\psi_+} \simeq \mathcal{A}^{ql}/N_{\psi_+} \simeq \mathrm{span}\{\prod_{k \in I}[X_k] | I \subset \mathbb{Z}, |I| < \infty\}
\tag{A.75}
$$

*This is because any local operator can be written as a linear combination of Pauli basis*

$$
\prod_{i \in I} X_i \prod_{j \in J} Z_j, \ |I|, |J| < \infty
\tag{A.76}
$$

*there is no Pauli* $Y_k$ *operator above since it is not independent, i.e.,* $Y_k = -iX_k Z_k$. *The quotient procedure amounts to regard* $Z_j = 1$ *for all* $j \in \mathbb{Z}$. *So the claim above follows.*

*We give a more intuitive explanation of the this example below. More intuitively,*

$$
\pi_+(Z_k)|\uparrow\rangle = |\uparrow\rangle
\tag{A.77}
$$

*for any* $k \in \mathbb{Z}$. *On the other hand,*

$$
\pi_+(X_k)|\uparrow\rangle \neq |\uparrow\rangle
\tag{A.78}
$$

*which can be seen by applying* $Z_k$ *at site* $k$. *The action of* $Y_k$ *is not independent since* $Y_k = iZ_k X_k$, *so we ignore it. Thus states in GNS Hilbert space* $\mathcal{H}_+$ *of* $|\uparrow\rangle$ *can be identified as*

$$
\prod_{j \in J} \pi_+(X_j)|\uparrow\rangle
\tag{A.79}
$$

where $J$ is a **finite** subset of $\mathbb{Z}$. That is, the states in $\mathcal{H}_+$ are configurations where almost all spins are up but only finitely many spins are flipped[17]. Also, it is easy to check that this representation is irreducible, hence $\psi_+$ is pure as expected.

Similarly, for $\psi_-$, the all-spin-down state (also denoted as $|\downarrow\rangle$), the GNS Hilbert space $\mathcal{H}_-$ in this case is similar to $\mathcal{H}_+$, but most of its spins are down while only finitely many of them are up. The associated representation is denoted as $\pi_-$.

### 5. Superselection sectors

In this subsection, we discuss an important application of the concept of GNS construction discussed in the previous subsection.

One drastic difference between infinite systems and finite cases is the notion of superselection sectors in infinite systems. Physically speaking, two states in an infinite system fall into different superselection sectors if and only if they cannot be connected by local operators (see Sec. 7.1 of Ref. [58] for a mathematical definition of superselection sectors).

For example, different ground states related to spontaneous symmetry breaking fall into different superselection sectors, and topologically degenerate ground states in the toric code are also in different superselection sectors.

A related notion in the GNS representations is as follows.

**Definition A.14.** *For $\mathcal{A}^{ql}$, the GNS representations of two different states $\psi, \sigma$ are said to be inequivalent if there is no unitary map $U : \mathcal{H}_\rho \to \mathcal{H}_\psi$ such that $U^\dagger \pi_\psi(A)U = \pi_\rho(A)$.*

As we will see later, the (in)equivalence of irreducible GNS representations is closely related to superselection sectors. For demonstration, we consider the following example.

**Example A.10** (Sec. 3.1 of Ref. [46])**.** *Let us continue our Ising model example A.9. Now we show that $\pi_+$ and $\pi_-$ defined earlier are inequivalent in the sense of definition A.14.*

*Suppose they are equivalent, i.e., there is a unitary operator $U : \mathcal{H}_+ \to \mathcal{H}_-$ such that $\pi_+(A) = U^\dagger \pi_-(A)U$ for all $A \in \mathcal{A}^{ql}$. We define the polarization operator*

$$m_N := \frac{1}{2N+1} \sum_{k=-N}^{N} Z_k \tag{A.80}$$

*where $N$ is arbitrarily big but stays finite. We **do not** talk about the limit of $m_N$ as $N \to \infty$. Note that*

$$\langle \uparrow |\pi_+(m_N)| \uparrow \rangle = 1 \tag{A.81}$$

*On the other hand,*

$$\langle \uparrow |\pi_+(m_N)| \uparrow \rangle = \langle \uparrow |U^\dagger \pi_-(m_N)U| \uparrow \rangle \tag{A.82}$$

*Note $U| \uparrow \rangle \in \mathcal{H}_-$, hence it has almost all spins being down, and only finitely many of them are up. Therefore, if $N$ is sufficiently large, one has*

$$\langle \uparrow |U^\dagger \pi_-(m_N)U| \uparrow \rangle \overset{N \gg 1}{\longrightarrow} -1 \tag{A.83}$$

*Thus one obtains a contradiction, which means there cannot be such a unitary operator $U$.*

The lesson from the above example is that $\pi_+$ is inequivalent to $\pi_-$ because $| \uparrow \rangle$ cannot be transformed into $| \downarrow \rangle$ by local operators only. Generalizing this example, one can obtain the following theorem.

**Theorem A.3** (Theorem 2.6.1 of Ref. [46])**.** *Given that $\mathcal{C}$ is a $C^*$-algebra and $\psi_1, \psi_2$ are two pure states, then their GNS representations $\pi_{\psi_1}$ and $\pi_{\psi_2}$ are equivalent if and only if $\psi_1$ and $\psi_2$ fall into the same superselection sector.*

———————

[17] As is noted before, one needs completion to make it really a     Hilbert space.

As a further consequence, if two pure states are in the same superselection sector, i.e., they can be related to each other by local operators, then they are almost the same outside a small region. Thus one concludes the following.

**Proposition A.2** (Proposition 3.2.8 of Ref. [46])**.** *Given 2 pure states $\psi_1, \psi_2$ of $\mathcal{A}^{ql}$, then the following statements are equivalent:*

1. *The GNS representation $\pi_{\psi_1}$ is equivalent to $\pi_{\psi_2}$.*

2. *For any $\epsilon > 0$, there is a finite subset $\Gamma_\epsilon$ such that*

$$|\psi_1(A) - \psi_2(A)| < \epsilon ||A|| \tag{A.84}$$

*for any $A \in \mathcal{A}^{ql}_{\Gamma_\epsilon^c}$, where $\Gamma_\epsilon^c$ means the complement of $\Gamma_\epsilon$.*

*If one of these conditions is satisfied, we say that $\psi_1$ is equivalent to $\psi_2$ and write $\psi_1 \simeq \psi_2$.*

For example, if $U \in \mathcal{U}^{ql}$ then $\psi \simeq \psi \circ \mathrm{Ad}_U$ in above sense.

**Remark A.5.** *As a remark on the above example, one can similarly show that ground states exhibiting spontaneous symmetry breaking (SSB) fall into different superselection sectors if there is a local order parameter $\mathcal{O}_k$ ($Z_k$ for Ising model) and unbroken translation symmetry. Especially, we do not need $\mathcal{O}_k$ to be Hermitian or to commute with Hamiltonian.*

Another lesson from the above example is that states in different superselection sectors cannot be superposed coherently. To see this, consider the GHZ-state defined on a finite system with size $L$,

$$|GHZ\rangle := \frac{1}{\sqrt{2}}(|\uparrow\rangle_L + |\downarrow\rangle_L) \tag{A.85}$$

where subscript $L$ means each state is defined on a chain of size $L$. Its density matrix is

$$\rho_{GHZ} = \frac{1}{2}(|\uparrow\rangle\langle\uparrow|_L + |\downarrow\rangle\langle\downarrow|_L) + \frac{1}{2}(|\uparrow\rangle\langle\downarrow|_L + |\downarrow\rangle\langle\uparrow|_L) \tag{A.86}$$

Note that the cross terms (in the second parenthesis) evaluate to 0 on any local operators in the thermodynamic limit. Thus, these two terms vanish as a state of $\mathcal{A}^{ql}$. As a result,

$$\rho_{GHZ} \stackrel{L \to \infty}{\Longrightarrow} \frac{1}{2}(|\uparrow\rangle\langle\uparrow| + |\downarrow\rangle\langle\downarrow|) \tag{A.87}$$

Namely, this GHZ-state becomes a mixed state in this limit.

As explained above, states in different superselection sectors are highly separated in the sense that no local operator can couple these states. So it is natural to expect that states in different superselection sectors must be orthogonal in the sense of definition A.9, and below we show this is indeed the case.

**Proposition A.3.** *If pure states $\psi_1$ and $\psi_2$ of $\mathcal{A}^{ql}$ are in two different superselection sectors (i.e., $\pi_{\psi_1} \not\simeq \pi_{\psi_2}$), then*

$$||\psi_1 - \psi_2|| = 2 \tag{A.88}$$

This proof is based on von-Neumann's double commutant theorem so we state it here without a proof, and interested readers are referred to Sec. C.20 of Ref. [47] for a proof. Suppose that $\mathcal{A}$ is a subalgebra of $\mathcal{B}(\mathcal{H})$ for some Hilbert space (finite or infinite dimensional), its commutant (a.k.a centralizer) is defined as

$$\mathcal{A}' = \{x \in \mathcal{B}(\mathcal{H}) | ax = xa, \forall a \in \mathcal{A}\} \tag{A.89}$$

Then, we have

**Theorem A.4** (Theorem C.127 of Ref. [47])**.** *For any vector $|\psi\rangle \in \mathcal{H}$ and any $A \in \mathcal{A}''$ (double commutant of $\mathcal{A}$, i.e., the commutant of $\mathcal{A}'$), there exists a sequence $A_n \in \mathcal{A}$ such that*[18]

$$\lim_{n \to \infty} A_n |\psi\rangle = A |\psi\rangle \tag{A.90}$$

*This convergence is with respect to the inner product of $\mathcal{H}$.*

---

[18] Mathematically, this means $\mathcal{A}$ is dense in $\mathcal{A}''$ in strong operator topology.

*Proof to proposition A.3.* Here we prove this proposition for pure states $\psi_1$ and $\psi_2$, but the proof can be straightforwardly generalized to mixed states. Consider a representation $\pi := \pi_1 \oplus \pi_2$ on $\mathcal{H} := \mathcal{H}_1 \oplus \mathcal{H}_2$, where $\pi_i := \pi_{\psi_i}, i = 1, 2$ is the GNS representation of $\psi_i$. The assumption that $\psi_1$ and $\psi_2$ are in two superselection sectors indicates that $\pi_1 \neq \pi_2$.

Note that

$$\pi(A) = \begin{pmatrix} \pi_1(A) & 0 \\ 0 & \pi_2(A) \end{pmatrix} \tag{A.91}$$

The commutant of $\pi(A)$ in $\mathcal{B}(\mathcal{H})$ is given by Schur's lemma,

$$\pi(\mathcal{A}^{ql})' = \{T \in \mathcal{B}(\mathcal{H}) | T = \begin{pmatrix} \lambda_1 & 0 \\ 0 & \lambda_2 \end{pmatrix}, \lambda_1, \lambda_2 \in \mathbb{C}\} \tag{A.92}$$

So the double commutant is

$$\pi(\mathcal{A}^{ql})'' = \{S \in \mathcal{B}(\mathcal{H}) | S = \begin{pmatrix} x & 0 \\ 0 & y \end{pmatrix}, x \in \mathcal{B}(\mathcal{H}_1), y \in \mathcal{B}(\mathcal{H}_2)\} \tag{A.93}$$

Especially, $\pi(A)''$ contains the following element with a unit norm

$$S = \begin{pmatrix} 1 & 0 \\ 0 & -1 \end{pmatrix} \tag{A.94}$$

By theorem A.4, there exists $A_n \in \mathcal{A}^{ql}$ such that $\lim_{n\to\infty} \pi(A_n)|\psi\rangle = S|\psi\rangle$ for any $|\psi\rangle \in \mathcal{H}$. Thus

$$\psi_1(A_n) - \psi_2(A_n) = \langle\psi_1|\pi_1(A_n)|\psi_1\rangle - \langle\psi_2|\pi_2(A_n)|\psi_2\rangle \overset{n\to\infty}{\Longrightarrow} 2 \tag{A.95}$$

So we have

$$||\psi_1 - \psi_2|| = 2 \tag{A.96}$$

$\square$

**Remark A.6.** *This proof does not work if $\psi_1$ and $\psi_2$ are in the same superselection sector. In that case $\pi_1(A) = \pi_2(A)$ after a proper choice of basis, so we have*

$$\pi(\mathcal{A}^{ql})' = \{T \in \mathcal{B}(\mathcal{H}) | T = \begin{pmatrix} \lambda_1 & \lambda_2 \\ \lambda_3 & \lambda_4 \end{pmatrix}, \lambda_i \in \mathbb{C}, i = 1, 2, 3, 4\} \tag{A.97}$$

*and the double commutant*

$$\pi(\mathcal{A}^{ql})'' = \{S \in \mathcal{B}(\mathcal{H}) | S = \begin{pmatrix} x & 0 \\ 0 & x \end{pmatrix}, x \in \mathcal{B}(\mathcal{H}_1)\} \tag{A.98}$$

*So the above proof does not apply.*

**Remark A.7.** *Proposition A.3 is still true if $\psi_1$ and $\psi_2$ are mixed states as long as they are in different superselection sectors. We say two mixed states are in two different superselection sectors if after decomposing them as Eq. (A.65), all of their pure components are in the different superselection sectors. Mathematically, one says $\psi_1$ and $\psi_2$ are disjoint (see def. 8.18 of Ref. [47]).*

By example A.6, we obtain

**Corollary A.3.** *If $\psi_1$ and $\psi_2$ are states of $\mathcal{A}^{ql}$ from different superselection sectors, then they are independent because they are orthogonal. Especially, for any*

$$\psi = t\psi_1 + (1-t)\psi_2, 0 < t < 1 \tag{A.99}$$

*we have $\pi_\psi = \pi_1 \oplus \pi_2$. That is, $\pi(A)$ is always block-diagonal and there is no local operator can couple $\psi_1$ and $\psi_2$.*

## 6.  Split property and area law

To get prepared to prove our main theorems, it is useful to discuss some entanglement properties of states, which is the subject of this subsection.

Given a subset $\Gamma$ (finite or infinite) of our infinite lattice $\Lambda$, the algebra of quasi-local operators always decomposes as

$$\mathcal{A}^{ql} \simeq \mathcal{A}^{ql}_\Gamma \otimes \mathcal{A}^{ql}_{\Gamma^c} \tag{A.100}$$

Recall that $\mathcal{A}^{ql}_\Gamma$ is the quasi-local operator algebra supported on $\Gamma$ and $\Gamma^c := \Lambda \setminus \Gamma$ is complement of $\Gamma$. Since $\Gamma$ is not necessarily finite, operators in $\mathcal{A}^{ql}_\Gamma$ are not necessarily local. Nevertheless, if $\Gamma$ is finite, $\mathcal{A}^{ql}_\Gamma$ is the local operator algebra supported on $\Gamma$ and hence it is finite dimensional.

Before we proceed, let us define the tensor product of states

**Definition A.15.** *For any two $C^*$-algebra $\mathcal{C}_1, \mathcal{C}_2$ and two states $\psi_i : \mathcal{C}_i \to \mathbb{C}, i = 1, 2$, the tensor product of states $\psi_1 \otimes \psi_2$ is a state of $\mathcal{C}_1 \otimes \mathcal{C}_2$[19], defined as*

$$(\psi_1 \otimes \psi_2)(A_1 \otimes A_2) := \psi_1(A_1)\psi_2(A_2) \tag{A.101}$$

*for any $A_i \in \mathcal{C}_i$.*

Given a pure state $\psi$ of $\mathcal{A}^{ql}$ and the above decomposition $\mathcal{A}^{ql} \simeq \mathcal{A}^{ql}_\Gamma \otimes \mathcal{A}^{ql}_{\Gamma^c}$, we may not be able to factorize $\psi$ into the form $\psi_\Gamma \otimes \psi_{\Gamma^c}$ for some pure states $\psi_\Gamma : \mathcal{A}^{ql}_\Gamma \to \mathbb{C}$ and $\psi_{\Gamma^c} : \mathcal{A}^{ql}_{\Gamma^c} \to \mathbb{C}$. In the case of 1d system and $\Gamma = (-\infty, 0)$, we have the notion of the split property[20].

**Definition A.16.** *In a 1d system with the following decomposition*

$$\mathcal{A}^{ql} \simeq \mathcal{A}^{ql}_{<0} \otimes \mathcal{A}^{ql}_{\geqslant 0} \tag{A.102}$$

*where $\mathcal{A}^{ql}_{<0}$ is a subalgebra of quasi-local operators which are supported on $\Gamma = (-\infty, 0)$ only and similarly for $\mathcal{A}^{ql}_{\geqslant 0}$. A pure state $\psi$ is said to split at the origin if*

$$\psi \simeq \psi_{<0} \otimes \psi_{\geqslant 0} \tag{A.103}$$

*where $\psi_{<0}$ (resp. $\psi_{\geqslant 0}$) is a pure state of $\mathcal{A}^{ql}_{<0}$ (resp. $\mathcal{A}^{ql}_{\geqslant 0}$) and the equivalence is in the sense of proposition A.2.*

Note that we only define the split property for *infinite* chains. Also, the split property is defined with respect to a particular site on the chain. In other words, if a state splits at one site, a priori, it may not split at another site. Nevertheless, currently we do not have an example of state that splits only at one site, nor can we prove that states that split at one site must split at all sites. Below when we discuss the split property of a state, we will always explicitly point out at which sites the state splits.

In quantum mechanics, one can tensor different states to get a state of the composite system. Conversely, one can do the partial trace operation to reduce the degrees of freedom. What does partial trace correspond to in operator algebra? Given that $\Gamma \subset \lambda$ is a subset (it can be finite or infinite) and a state $\psi : \mathcal{A}^{ql} \to \mathbb{C}$, note that $\mathcal{A}^{ql}_\Gamma$ can be viewed as a subalgebra of $\mathcal{A}^{ql}$, hence one can get a state $\psi|_\Gamma$ of $\mathcal{A}^{ql}_\Gamma$ by the restriction

$$\psi|_\Gamma(A) = \psi(A), \forall A \in \mathcal{A}^{ql}_\Gamma \tag{A.104}$$

Here $\psi|_\Gamma$ can be viewed as the state obtained from $\psi$ by partially tracing the degrees of freedom in $\Gamma^c$. Please do not confuse $\psi|_\Gamma$ with $\psi_\Gamma$ in the definition of the split property, i.e., even if the state $\psi$ splits between $\Gamma$ and $\Gamma^c$, $\psi_\Gamma$ may *not* be obtained by restricting $\psi$ to $\Gamma$ except for some exceptionally special examples. In particular, states obtained by restriction (partial trace), e.g., $\psi|_\Gamma$, are not necessarily pure, but $\psi_\Gamma$ is pure by definition.

To give an example of states with the split property, let us define *product states*.

---

[19] We suppress the subtleties in the tensor product of $C^*$-algebras, see Sec 3.2.2 of Ref. [46] for details.

[20] The split property in 2 dimension is also proposed in Ref. [32], but we do not need it in the present paper. Also, in this paper we only need the split property for pure states, although the split property can also be defined for mixed states [30].

**Definition A.17.** *For any lattice system (including higher dimensions), a state $\omega$ is said to be factorized or a product state if*

$$\omega(AB) = \omega(A)\omega(B) \tag{A.105}$$

*whenever $A$ and $B$ have disjoint supports. Besides, in the present paper, product states are always assumed to be pure.*

As we will see below, a product state is arguably the simplest state in quantum many-body physics, since it means that spins at different site are not entangled with each other at all. A state that is not a product state is referred to as an entangled state.

**Lemma A.3.** *Let $\Gamma \subset \Lambda$ (finite or infinite and $\Lambda$ is any lattice) and $\omega$ be a (pure) product state, we always have*

$$\omega = \omega|_\Gamma \otimes \omega|_{\Gamma^c} \tag{A.106}$$

*Moreover, $\omega|_\Gamma$ and $\omega|_{\Gamma^c}$ are again pure product states.*

Specializing to 1d, the above lemma shows that any product state splits at every site.

*Proof to lemma A.3.* For any $A \in \mathcal{A}_\Gamma^{ql}, B \in \mathcal{A}_{\Gamma^c}^{ql}$, we have

$$\omega(AB) = \omega(A)\omega(B) = \omega|_\Gamma(A)\omega|_{\Gamma^c}(B) \tag{A.107}$$

Then Eq. (A.106) follows. To show that $\omega|_\Gamma$ is pure, let $\rho : \mathcal{A}_\Gamma^{ql} \to \mathbb{C}$ be a positive linear functional majorized by $\omega|_\Gamma$, i.e., $\rho \leqslant \omega|_\Gamma$ (see definition A.11). We then have

$$\rho \otimes \omega|_{\Gamma^c} \leqslant \omega|_\Gamma \otimes \omega|_{\Gamma^c} = \omega \tag{A.108}$$

By assumption, $\omega$ is a pure state, hence $\rho \otimes \omega|_{\Gamma^c} = \lambda\omega = \lambda\omega|_\Gamma \otimes \omega|_{\Gamma^c}$ for some $\lambda \in [0,1]$. Thus for any $A \in \mathcal{A}_\Gamma^{ql}$, we have

$$\rho(A) = (\rho \otimes \omega|_{\Gamma^c})(A \otimes I) = \lambda\omega(A \otimes I) = \lambda\omega|_\Gamma(A) \tag{A.109}$$

Hence $\omega|_\Gamma$ is pure by definition A.11.

To see that $\omega|_\Gamma$ is a product state, let $A, B \in \mathcal{A}_\Gamma^{ql}$ be quasi-local operators with disjoint support. Then

$$\omega|_\Gamma(AB) = \omega(AB) = \omega(A)\omega(B) = \omega|_\Gamma(A)\omega|_\Gamma(B) \tag{A.110}$$

Thus $\omega|_\Gamma$ is indeed a pure product state, and so is $\omega|_{\Gamma^c}$ for similar reasons. $\square$

**Remark A.8.** *However, in general, there are non-product states which satisfy the split property. For example, as shown in Sec. E 3, short-range entangled states split at every point, but they are generically not product states.*

In the special case where $\Gamma$ is finite, $\psi|_\Gamma$ is a state of $\mathcal{A}_\Gamma^{ql} \simeq \otimes_{k \in \Gamma}\mathcal{B}(\mathcal{H}_k)$, i.e., the quasi-local algebra is finite dimensional. Therefore, one can represent $\psi|_\Gamma$ by a density matrix $\rho_\psi$, as we have seen in example A.4.

**Definition A.18.** *Let $\Gamma$ be a finite region as above and a density matrix $\rho_\psi$ represent a state $\psi$ on $\Gamma$ (i.e., $\psi|_\Gamma$), then one defines the entanglement entropy of $\psi|_\Gamma$ as*

$$S(\rho_\psi) = -\text{tr}(\rho_\psi \log \rho_\psi) \tag{A.111}$$

*A state $\psi$ is said to satisfy the area law if*

$$S = O(|\partial\Gamma|) \tag{A.112}$$

*for any finite region $\Gamma$. Here $\partial\Gamma$ means the boundary of $\Gamma$.*

Especially, in 1d, the area law means that the entanglement entropy is bounded by a constant. The following lemma is of fundamental importance.

**Lemma A.4** (Theorem 1.5 of Ref. [30])**.** *In 1d spin chains, a state $\psi$ splits at every site if it satisfies the area law.*

Readers are referred to Sec. 2 of Ref. [30] for a proof of lemma A.4.

The lemma below characterizes the "uniqueness" of the factorization of states.

**Lemma A.5.** *Given a decomposition $\mathcal{A}^{ql} \simeq \mathcal{A}^{ql}_\Gamma \otimes \mathcal{A}^{ql}_{\Gamma^c}$ where $\Gamma$ and $\Gamma^c$ are infinite, and given pure states $\psi_\Gamma, \psi'_\Gamma$ (reps. $\psi_{\Gamma^c}, \psi'_{\Gamma^c}$) of $\mathcal{A}^{ql}_\Gamma$ (resp. $\mathcal{A}^{ql}_{\Gamma^c}$), if*

$$\psi_\Gamma \otimes \psi_{\Gamma^c} \simeq \psi'_\Gamma \otimes \psi'_{\Gamma^c} \tag{A.113}$$

*Then $\psi_\Gamma \simeq \psi'_\Gamma$ and $\psi_{\Gamma^c} \simeq \psi'_{\Gamma^c}$.*

*Proof.* We use proposition A.2. By assumption, for any $\epsilon > 0$, there exists a finite region $P_\epsilon$ such that for any operator $S \in \mathcal{A}^{ql}_{P_\epsilon^c}$, we have

$$|(\psi_\Gamma \otimes \psi_{\Gamma^c})(S) - (\psi'_\Gamma \otimes \psi'_{\Gamma^c})(S)| < \epsilon ||S|| \tag{A.114}$$

Now we choose $S = A \otimes B$ for any $A \in \mathcal{A}^{ql}_{\Gamma \setminus P_\epsilon}$ and $B \in \mathcal{A}^{ql}_{\Gamma^c \setminus P_\epsilon}$, we have

$$|\psi_\Gamma(A)\psi_{\Gamma^c}(B) - \psi'_\Gamma(A)\psi'_{\Gamma^c}(B)| < \epsilon ||A|| \cdot ||B|| \tag{A.115}$$

Now let $B = I$, hence $\psi_{\Gamma^c}(B) = \psi'_{\Gamma^c}(B) = 1$,

$$|\psi_\Gamma(A) - \psi'_\Gamma(A)| < \epsilon ||A|| \tag{A.116}$$

We then conclude $\psi_\Gamma \simeq \psi'_\Gamma$ and similarly $\psi_{\Gamma^c} \simeq \psi'_{\Gamma^c}$. $\qquad\square$

**Remark A.9.** *This lemma can fail if $\Gamma$ or $\Gamma^c$ is finite. In that case, $\Gamma \setminus P_\epsilon$ or $\Gamma^c \setminus P_\epsilon$ can be empty and above proof does not apply.*

## 7. Hamiltonians and ground states

In quantum mechanics, concepts such as Hamiltonians and time evolution play important roles. In this and the next subsections, we define these concepts in the operator algebra formalism.

In this subsection, we define Hamiltonians, ground states, and energy gaps in the formalism of operator algebra. Especially. we will explain how these definitions reduce to our more familiar cases for systems with finitely many degrees of freedom.

Naively, one can define local Hamiltonians as

$$H = \sum_{j \in \Lambda} h_j \tag{A.117}$$

where $h_j \in \mathcal{A}^l_{B(j,R)}$ is a local term, and $B(j, R) := \{p \in \Lambda | d(p, j) \leqslant R\}$ for some fixed $R > 0$. However, this does not work in general since it contains an infinite sum and one has to be careful about convergence. Actually Eq. (A.117) does not converge in general, hence one needs a more careful definition. Nevertheless, recall that in the operator algebra formalism, we work in the Heisenberg picture. In the Heisenberg picture, the Hamiltonian $H$ governs the time evolution of a local operator $A$ as

$$\frac{\mathrm{d}A}{\mathrm{d}t} = i[H, A] = i \sum_{j \in \Lambda} [h_j, A] \tag{A.118}$$

If $H$ is a local Hamiltonian, then the commutator $[H, A]$ contains at most finitely many non-zero terms by locality. Thus one can define $H$ by this commutator. This motivates the following definition.

**Definition A.19.** *A Hamiltonian is a derivation $\delta_H$ defined by*

$$\delta_H(A) = \sum_Z [h_Z, A], \forall A \in \mathcal{A}^l \tag{A.119}$$

*This Hamiltonian is local if $h_Z \in \mathcal{A}^l_\Gamma$ for some finite subset $\Gamma$ for all $Z$.*

Here a local (symmetric) derivation (which characterizes a local Hamiltonian) $\delta$ on $\mathcal{A}^l$ means the following

**Definition A.20.** *A local symmetric derivation $\delta$ is a map with the following properties.*

1. $\delta : \mathcal{A}^l \to \mathcal{A}^l$ *is $\mathbb{C}$-linear.*

2. $\delta(A^\dagger) = -\delta(A)^\dagger$

3. $\delta(AB) = \delta(A)B + A\delta(B)$

Clearly, this definition of Hamiltonian reduces to the usual one in finite quantum systems.

**Remark A.10.** *With long-range interactions, one has to make sure that the right hand side of Eq.* (A.119) *is convergent. In many cases, $\delta_H$ is only defined on local operators $\mathcal{A}^l$ or one says that $\delta_H$ is only **densely** defined on $\mathcal{A}^{ql}$. See lemma A.6 for an example.*

Given the notion of Hamiltonians, now we define ground states in the operator algebra formalism.

**Definition A.21.** *A state $\psi$ is said to be the ground state of the Hamiltonian $\delta_H$ if*

$$\psi(A^\dagger \delta_H(A)) \geqslant 0, \forall A \in \mathcal{A}^l \tag{A.120}$$

**Remark A.11.** *Let us explain why this definition agrees with the usual definition of ground states in finite-dimensional quantum mechanics. Let us assume $\psi$ is a state vector in some Hilbert space $\mathcal{H}$ denoted by $|\psi\rangle$. We also assume there is a Hamiltonian operator $H$ on $\mathcal{H}$, which has $|\psi\rangle$ as a ground state of energy $E$. Then Eq.* (A.120) *is equivalent to*

$$\langle\psi|A^\dagger H A|\psi\rangle \geqslant E\langle\psi|A^\dagger A|\psi\rangle \tag{A.121}$$

*In a finite system, any state in $\mathcal{H}$ can be prepared by applying local operators to $|\psi\rangle$, and hence Eq.* (A.121) *really means that $|\psi\rangle$ has the lowest energy $E$ in Hilbert space $\mathcal{H}$ and is a ground state.*

**Remark A.12.** *As another remark, a drawback of this definition is that it is not obvious if a ground state is pure or not. For example, given a finite size system with the classical Ising Hamiltonian, although its ground state is pure, some of its ground states (i.e., the GHZ states) become mixed after taking thermodynamic limit. So given a Hamiltonian $\delta_H$, we have to check if its ground state is pure or not. This is drastically different from the finite dimensional cases. We will show that gapped ground states of Hamiltonians with sufficiently short-range interactions are pure indeed (see Eq.* (A.129) *and theorem A.5).*

Now we are ready to talk about the notion of locally-unique gapped ground states and the energy gaps.

**Definition A.22** (Locally-unique gapped ground state)**.** *A ground state $\psi$ of Hamiltonian $\delta_H$ is a locally-unique gapped ground state if there is a $\gamma > 0$ such that*

$$\psi(A^\dagger \delta_H A) \geqslant \gamma \psi(A^\dagger A) \tag{A.122}$$

*for any $A \in \mathcal{A}^l$ with $\psi(A) = 0$. The energy gap $\Delta$ is the largest possible $\gamma$ satisfying the above inequality. A locally-unique gapped ground state is unique if it is the only locally-unique gapped ground state.*

**Remark A.13.** *Again, let us check that this definition reduces to our familiar notions in finite systems. First, the condition $\psi(A) = 0$ is to exclude the case where $A$ is a constant multiple of the identity operator, and it can always be achieved by redefining $A \to A - \psi(A)I$. Thus Eq.* (A.122) *means that for any*

$$|A\rangle := A|\psi\rangle, \, A \in \mathcal{A}^{ql} \tag{A.123}$$

*which is not proportional to $|\psi\rangle$, we must have*

$$\frac{\langle A|H|A\rangle}{\langle A|A\rangle} \geqslant \Delta > 0 \tag{A.124}$$

*Recall that any state in $\mathcal{H}$ can be obtained by applying local operators to $|\psi\rangle$. The above inequality means that $|\psi\rangle$ is the **only** state that has an energy smaller than $\Delta$ in the Hilbert space $\mathcal{H}$, agreeing with our usual definition of a unique gapped ground state. In the context of infinite systems, there may be other ground states of $\delta_H$, but they are not in the same superselection sector as $\psi$. Hence follows the name locally-unique.*

**Remark A.14.** *The gapped ground state defined in Ref. [25] is actually our locally-unique gapped ground state.*

**Example A.11.** *We give another example to show that not all ground states in the usual quantum mechanics are locally-unique. The model is the (classical) Ising model with the following Hamiltonian*

$$\delta_H(A) = -\sum_{j \in \mathbb{Z}} [Z_j Z_{j+1}, A] \tag{A.125}$$

*where $Z_j$ is usual Pauli operator. Using definition A.21, one can choose the ground state of this model to be the infinite-system version of the GHZ state:*

$$\psi_{GHZ} = \frac{1}{2}(\psi_\uparrow + \psi_\downarrow) \tag{A.126}$$

*However, this state is* **not** *a locally-unique gapped ground state of the Ising model. Let $A = Z_k$ supported at site $k$. Note that*

$$\psi(A^\dagger \delta_H(A)) = 0 \tag{A.127}$$

*since $A$ commutes with Pauli-Z operators. On the other hand,*

$$\begin{aligned} \psi(A) &= 0 \\ \psi(A^\dagger A) &= 1 \end{aligned} \tag{A.128}$$

*hence $\psi$ is not a locally-unique gapped ground state because $0 = \psi(A\delta_H(A)) < \psi(A^\dagger A) = 1$ and Eq. (A.122) is violated.*

*In fact, we will show in theorem A.5 that all locally-unique gapped ground states are pure.*

## 8. Time evolution

In the above, we have introduced the concepts of operators, states and Hamiltonians in the operator algebra formalism. In this subsection, we introduce the notions of time evolution.

First, let us present the definition of admissible Hamiltonians [37] (see definition A.19 for defining Hamiltonians in operator algebra formalism).

**Definition A.23.** *In a 1d lattice system $\Lambda = \mathbb{Z}$ where sites are labeled by $i$, the Hamiltonian $\delta_H$ is admissible if $\delta_H = \sum_{Z:|Z| \leqslant k}[h_Z, \bullet]$ with $h_Z$'s satisfying*

$$\sup_{i \in \mathbb{Z}} \sum_{\substack{Z:|Z| \leqslant k, Z \ni i \\ \text{diam}(Z) = r}} ||h_Z|| < \frac{J}{r^{\mathfrak{a}}}, \text{ with } \mathfrak{a} > 2, \tag{A.129}$$

$$\sup_{i \in \mathbb{Z}} ||h_i|| < B$$

*where $\text{diam} Z := \max_{x,y \in Z} d(x,y)$, $J$ and $B$ are positive constants and $h_i$ is a one-body potential at site $i$.*

We have not shown that the above definition is well-defined. In particular, we have not shown that the $\delta_H$ is indeed a derivation from $\mathcal{A}^l$ to $\mathcal{A}^{ql}$ (see definition A.20 and remark A.10). Below we will see that this is indeed the case. More precisely, if $A \in \mathcal{A}^l_\Gamma$ is a local operator, one can define a sequence

$$H_n^A := \sum_{\substack{Z:Z \cap \Gamma \neq \emptyset \\ \text{diam}(Z) \leqslant n}} [h_Z, A] \tag{A.130}$$

then its limit $n \to \infty$ exists,

$$\delta_H(A) := \lim_{n \to \infty} H_n^A \in \mathcal{A}^{ql} \tag{A.131}$$

However, as is remarked before (see remark. A.10), $\delta_H$ cannot be defined on the whole $\mathcal{A}^{ql}$.

**Lemma A.6.** *Admissible Hamiltonians are well-defined, that is, if $\delta_H$ is admissible and $A \in \mathcal{A}^l$, then $\delta_H(A) \in \mathcal{A}^{ql}$, i.e., the limits in above definition exist in $\mathcal{A}^{ql}$.*

*Proof.* We first show that $H_n^A$ defined in Eq. (A.130) is a local operator. Note for each $n > 0$, there are only finitely many subsets $Z$ such that $Z \cap \Gamma \neq \emptyset$, hence $H_n^A \in \mathcal{A}_{B(\Gamma, n)}^l$. Below we show $\{H_n^A\}_{n=1,2,\ldots}$ is a Cauchy sequence hence it is convergent in $\mathcal{A}^{ql}$. To this end, consider $m \geqslant n$,

$$||H_m^A - H_n^A|| = ||\sum_{r=n}^m \sum_{\substack{Z : Z \cap \Gamma \neq \emptyset \\ \mathrm{diam}(Z)=r}} [h_Z, A]|| \leqslant 2|\Gamma| \cdot ||A|| \sum_{r=n}^m \frac{J}{r^{\mathfrak{a}}} \tag{A.132}$$

where we have used $||\sum_i B_i|| \leqslant \sum_i ||B_i||$ and $||[B, C]|| \leqslant 2||B|| \cdot ||C||$ for any operators $B_i, B$ and $C$ in $\mathcal{A}^{ql}$.

Note for $m \geqslant n \geqslant n_0$,

$$\sum_{r=n}^m \frac{1}{r^{\mathfrak{a}}} < \int_{n_0}^\infty r^{-\mathfrak{a}} dr = \frac{n_0^{1-\mathfrak{a}}}{\mathfrak{a} - 1} \tag{A.133}$$

which goes to 0 as $n_0 \to \infty$ since $\mathfrak{a} > 2$. Therefore, $\{H_n^A\}$ is indeed a Cauchy sequence and so it converges to some element in $\mathcal{A}^{ql}$. $\qquad\square$

Given the definition of admissible Hamiltonians, our next task is to define the time evolution generated by admissible Hamiltonians. This is tricky. Naively, one can use the following exponential

$$\alpha_t(A) \overset{?}{:=} \exp(t\delta_H)(A) = \sum_{k=0}^\infty \frac{t^k}{k!} \delta_H^k(A) \tag{A.134}$$

However, this definition does not work in general since $\delta_H$ is only defined on $\mathcal{A}^l$ and typically $\delta(A) \in \mathcal{A}^{ql}$ even if $A \in \mathcal{A}^l$! Thus, $\delta_H^2(A) = \delta_H(\delta_H(A))$ is ill-defined.

However, the proof of lemma A.6 hints us that we can define the $n$-truncated Hamiltonian

$$H_n := \sum_{Z : Z \subset [-n, n]} h_Z \tag{A.135}$$

This is a local operator supported on $[-n, n]$ hence it is essentially finite dimensional. One can exponentiate it to define

$$\alpha_n^t(A) := e^{iH_n t} A e^{-iH_n t} \tag{A.136}$$

where $A$ is a local operator and $\alpha_n^t$ is an automorphism of local operators (actually of $\mathcal{A}^{ql}$). Our goal is to show the limit

$$\alpha^t(A) := \lim_{n \to \infty} \alpha_n^t(A) \tag{A.137}$$

exists and it defines the time evolution of $\delta_H$.

**Proposition A.4** (Existence of dynamics for admissible Hamiltonians)**.** *The limit defined by Eq. (A.137) exists and it defines the time evolution generated by the admissible Hamiltonian $\delta_H$, which is a strongly continuous*[21]

This proposition can be easily proved with the help of theorem 2.2 of Ref. [59], which we review below. To this end, we introduce the notion of reproducing functions. Consider a lattice $\Lambda$ with a metric $d$ and let $F : \mathbb{R}^{\geqslant 0} \to \mathbb{R}^{\geqslant 0}$ be a decreasing function with $\lim_{r \to \infty} F(r) = 0$. This function $F$ is called reproducing if

$$\sup_{y \in \Lambda} \sum_{x \in \Lambda} F(d(x, y)) < \infty$$

$$\sum_{l \in \Lambda} F(d(n, l)) F(d(l, m)) < C F(d(n, m)), \, \forall \, m, n \in \Lambda \tag{A.138}$$

for some $0 < C < \infty$. Especially, if $\Lambda = \mathbb{Z}$, then it can be checked that $F(r) = (1 + r)^{-1-\epsilon}$ is reproducing for any $\epsilon > 0$. Now we come to the most important lemma in this section.

————

[21] By strongly continuous, we mean that $\lim_{t \to 0} \alpha_t(A) = A$ for $\quad$ any $A \in \mathcal{A}^{ql}$.

**Lemma A.7** (Theorem 2.2 of Ref. [59]). *If the Hamiltonian $H = \sum_Z h_Z$ satisfies*

$$||H||_F := \sup_{m,n \in \Lambda} \frac{1}{F(d(m,n))} \sum_{Z:m,n \in Z} ||h_Z|| < \infty \tag{A.139}$$

*for some reproducing function $F(r)$ defined above, then the limit in Eq. (A.137) exists and defines a strongly continuous one-parameter subgroup of $\mathrm{Aut}(\mathcal{A}^{ql})$. This convergence is uniform for $t$ in some compact sets. It also does not depend on the choice of $n$-truncation.*

*Proof to proposition A.4.* We only have to check that our admissible Hamiltonian satisfies Eq. (A.139) with some proper choice of $F$. We choose $F(r) = (1+r)^{-1-\epsilon}$ with $0 < \epsilon < \mathfrak{a} - 2$, then $||H||_F$ defined in Eq. (A.139) becomes

$$||H||_F = \sup_{m,n \in \mathbb{Z}} (1+d)^{1+\epsilon} \sum_{Z:m,n \in Z} ||h_Z|| = \sup_{m,n \in \mathbb{Z}} (1+d)^{1+\epsilon} \sum_{r=d}^{\infty} \sum_{\substack{Z:m,n \in Z \\ \mathrm{diam}(Z)=r}} ||h_Z|| \tag{A.140}$$

where $d := d(m,n)$ and note $\mathrm{diam}(Z) := \sup_{x,y \in Z} d(x,y) \geqslant d$. According to Eq. (A.129), we have

$$\sum_{\substack{Z:m,n \in Z \\ \mathrm{diam}(Z)=r}} ||h_Z|| \leqslant \sup_{n \in Z} \sum_{\substack{Z:n \in Z \\ \mathrm{diam}(Z)=r}} ||h_Z|| \leqslant \frac{J}{r^{\mathfrak{a}}} \tag{A.141}$$

Therefore, it only remains to note that

$$\sup_{d \in \mathbb{Z}^{\geqslant 0}} (1+d)^{1+\epsilon} \sum_{r=d}^{\infty} \frac{J}{r^{\mathfrak{a}}} < \sup_{d \in \mathbb{Z}^{\geqslant 0}} (1+d)^{1+\epsilon} \int_d^{\infty} \frac{J}{r^{\mathfrak{a}}} dr = \frac{J}{\mathfrak{a}-1} \sup_{d \in \mathbb{Z}^{\geqslant 0}} (1+d)^{1+\epsilon} d^{1-\mathfrak{a}} \tag{A.142}$$

For large $d$, the right hand side can be estimated as $d^{2+\epsilon-\mathfrak{a}} \to 0$ if $d \to \infty$. Hence, $||H||_F$ defined in Eq. (A.139) must be bounded. By lemma A.7, the limit defined in Eq. (A.137) exists and defines a strongly continuous one-parameter subgroup of $\mathrm{Aut}(\mathcal{A}^{ql})$. $\qquad \square$

Given the operator algebra $\mathcal{A}^{ql}$ with a well-defined time evolution $\alpha^t$, one says they together define a $C^*$-dynamical system.

**Remark A.15.** *One may wonder whether we can define the Hamiltonian on whole $\mathcal{A}^{ql}$ by*

$$\delta_H(A) \overset{?}{:=} \lim_{t \to 0} \frac{\alpha^t(A) - A}{t}, \forall A \in \mathcal{A}^{ql} \tag{A.143}$$

*However, despite that $\alpha_t(A)$ is continuous in $t$, it is in general not differentiable in $t$ if $A \notin \mathcal{A}^l$, so the derivative above does not exist in general. Nevertheless, from the definition in Eq. (A.136), one sees that the derivative exists if $A \in \mathcal{A}^l$ and it coincides with our earlier definition of $\delta_H(A)$. The quickest way to see this is to write $\alpha^t(A) = \lim_{n \to \infty} \alpha_n^t(A)$ and change the order of limits $t \to 0$ and $n \to \infty$ (this is valid by uniform convergence in lemma A.7). Note for finite $n$, $\frac{d}{dt}\alpha_n^t(A) = H_n(A)$ where $H_n(A)$ is defined in Eq. (A.130). It is shown in lemma A.6 that the limit $\lim_{n \to \infty} H_n(A)$ exists if $A$ is a local operator.*

In fact, the above proposition A.4 can be easily generalized to higher dimensions using exactly the same argument, but with a rather different choice of reproducing function $F$.

**Proposition A.5** (Higher dimensional version of theorem A.4). *For $D$-dimensional lattice $\Lambda \simeq \mathbb{Z}^D$, if the Hamiltonian satisfies admissible condition Eq. (A.129) with $\mathfrak{a} > 2D$, then the dynamics exists.*

The proof is also to apply lemma A.7 but with $F(r) = (1+r)^{-D-\epsilon}$.

Below we present a simple application of the concept of time evolution, which is expected from the usual quantum mechanics.

**Corollary A.4.** *Given a Hamiltonian $\delta_H$ which generates a time-evolution $\alpha^t$ and $\psi$ the ground state of this Hamiltonian (see Eq. (A.120)), then $\psi$ is invariant under $\alpha^t$, i.e., $\psi \circ \alpha^t = \psi$.*

*Proof of corollary A.4.* To show that $\psi \circ \alpha^t = \psi$, we first show that

$$\psi(\alpha^t(A)) = \psi(A), \forall A \in \mathcal{A}^l \tag{A.144}$$

Since $\alpha^t$ is differentiable with respect to $t$ on local operators, this amounts to showing

$$\psi(\delta_H(A)) = 0, \forall A \in \mathcal{A}^l \tag{A.145}$$

To this end, note that

$$A = \frac{A + A^\dagger}{2} + i\frac{A - A^\dagger}{2i} \tag{A.146}$$

So without any loss of generality, we can assume $A$ to be Hermitian due to the linearity of $\psi$ and $\delta_H$. Besides, we can always make a shift $A \to A + \lambda I$ where $\lambda \in \mathbb{R}$, so we can further assume that $A$ has only positive eigenvalues. Given the assumption that $A$ is positive (i.e., $A$ is self-adjoint and all eigenvalues are positive), there is a positive root of square of $A$, denoted as $\sqrt{A}$, which is again local. Thus, by definition A.20,

$$\psi(\delta_H(A)) = \psi(\delta_H(\sqrt{A}\sqrt{A})) = \psi(\delta_H(\sqrt{A})\sqrt{A}) + \psi(\sqrt{A}\delta_H(\sqrt{A})) \tag{A.147}$$

Note that

$$\psi(\sqrt{A}\delta_H(\sqrt{A}))^* = \psi((\sqrt{A}\delta_H(\sqrt{A}))^\dagger) = -\psi(\delta_H(\sqrt{A})\sqrt{A}) \tag{A.148}$$

where we have used that $(\delta_H(B))^\dagger = \sum_Z [h_Z, B]^\dagger = -\delta_H(B^\dagger)$ since $h_Z^\dagger = h_Z$. On the other hand, since $\psi$ is a ground state of $\delta_H$, we have $\psi(\sqrt{A}\delta_H(\sqrt{A})) \geqslant 0$, in particular, $\psi(\sqrt{A}\delta_H(\sqrt{A})) \in \mathbb{R}$. Therefore, $\psi(\sqrt{A}\delta_H(\sqrt{A})) = -\psi(\delta_H(\sqrt{A})\sqrt{A})$ and

$$\psi(\delta_H(A)) = 0 \tag{A.149}$$

So we have shown that

$$\psi(\alpha^t(A)) = \psi(A), \forall A \in \mathcal{A}^l \tag{A.150}$$

For general $A \in \mathcal{A}^{ql}$, we use the standard trick to approximate $A$ by local operators, and the result follows from the fact $\alpha^t(\lim_{j\to\infty} A_j) = \lim_{j\to\infty} \alpha^t(A_j)$ and $\psi$ is continuous (see proposition 2.3.11 of Ref. [44]). □

Notice the corollary above holds as long as the time evolution generated by the Hamiltonian $\delta_H$ exists, regardless of whether this Hamiltonian is admissible or not.

## 9. The GNS Hamiltonian

It is natural to ask whether there is a way to realize the abstract Hamiltonian $\delta_H$ as an operator on some Hilbert space (especially, the GNS Hilbert space $\mathcal{H}_\psi$ for some state $\psi$, see Sec. A 4 for the relevant construction). Now we address this question in this subsection. Also, using the concepts and techniques developed so far, in this subsection we will prove theorem A.5, which plays a vital role in our proof of the theorem III.1 in the main text.

**Proposition A.6** (Corollary 2.5.8 of Ref. [46])**.** *Suppose $\alpha$ is an automorphism of $\mathcal{A}^{ql}$ and $\psi$ is a state which is invariant under $\alpha$. Then on the GNS Hilbert space $\mathcal{H}_\psi$, there exists an operator $U_\alpha \in \mathcal{B}(\mathcal{H}_\psi)$ such that*

$$\pi_\psi(\alpha(A)) = U_\alpha \pi_\psi(A) U_\alpha^{-1} \tag{A.151}$$

*Proof.* Suppose $(\pi_\psi, \mathcal{H}_\psi, |\psi\rangle)$ is a GNS triple. To show the above proposition, according to corollary A.2, it suffices to show that $(\pi_\psi \circ \alpha, \mathcal{H}_\psi, |\psi\rangle)$ is also a GNS triple. To this end, first notice that $\pi_\psi \circ \alpha$ is a homormorphism from $\mathcal{A}^{ql}$ to $\mathcal{B}(\mathcal{H}_\psi)$. Next, notice that for any $A \in \mathcal{A}^{ql}$, $\langle\psi|\pi_\psi \circ \alpha(A)|\psi\rangle = \langle\psi|\pi_\psi(\alpha(A))|\psi\rangle = \psi(\alpha(A)) = \psi \circ \alpha(A) = \psi(A)$. Combining these two observations, we conclude that $(\pi_\psi \circ \alpha, \mathcal{H}_\psi, |\psi\rangle)$ is indeed also a GNS triple. Then the proposition follows due to corollary A.2.

□

As a corollary, we have

**Corollary A.5.** *Let $\psi$ be a ground state of a Hamiltonian $\delta_H$, which generates a time evolution (i.e., a strongly continuous one-parameter subgroup) on the GNS Hilbert space $\mathcal{H}_\psi$, then there exists a so-called GNS Hamiltonian $H_{\mathrm{GNS}}$ such that*

$$\pi_\psi(\delta_H(A)) = [H_{\mathrm{GNS}}, \pi_\psi(A)], \forall A \in \mathcal{A}^l \tag{A.152}$$

*Typically, the GNS Hamiltonian is not in $\pi_\psi(A^{ql})$ but in its double commutant $\pi_\psi(\mathcal{A}^{ql})''$.*

*Proof.* Corollary A.4 implies that

$$\psi \circ \alpha^t = \psi \tag{A.153}$$

Thus by proposition A.6, there exists a unitary operator $U_t \in \mathcal{B}(\mathcal{H}_\psi)$, such that

$$\pi_\psi(\alpha^t(A)) = U_t \pi_\psi(A) U_t^\dagger, \ \forall A \in \mathcal{A}^{ql} \tag{A.154}$$

When $A \in \mathcal{A}^l$, taking derivatives on both sides and writing $H_{\mathrm{GNS}} = -i\frac{\mathrm{d}}{\mathrm{d}t}U_t$, we get

$$\pi_\psi(\delta_H(A)) = [H_{\mathrm{GNS}}, \pi_\psi(A)], \ \forall A \in \mathcal{A}^l \tag{A.155}$$

which proves Eq. (A.152). To see that the GNS Hamiltonian is in the double commutants $\pi_\psi(\mathcal{A}^{ql})''$, we refer to corollary 3.2.48 of Ref. [44]. $\qquad\square$

Now we are ready to establish the following important theorem.

**Theorem A.5.** *Let $\psi$ be a locally-unique gapped ground state of an admissible Hamiltonian, then*

    *1. It is non-degenerate (i.e., unique) in the GNS Hilbert space.*

    *2. This state $\psi$ is pure.*

The proof of this theorem relies on the following few lemmas.

**Lemma A.8** (Riesz representation theorem)**.** *Let $\mathcal{H}$ be a Hilbert space (finite or infinite-dimensional) and $F$ be a bounded linear functional on $\mathcal{H}$, then there exists a unique state $|F\rangle$ in $\mathcal{H}$, such that*

$$\langle F|\psi\rangle = F(|\psi\rangle) \tag{A.156}$$

*for any $|\psi\rangle \in \mathcal{H}$. This justifies the usual notion that bras are dual to kets.*

For a proof, we refer to standard textbooks on functional analysis e.g., theorem 2.E in Ref. [60]. As a corollary, we have

**Corollary A.6.** *Consider any bounded sesquilinear forms $(\bullet, \bullet) : \mathcal{H} \times \mathcal{H} \to \mathbb{C}$, i.e., $(\bullet, \bullet)$ is linear in the second argument and anti-linear in the first argument, and furthermore,* [22]

$$\frac{|(u, v)|}{||u|| \cdot ||v||} < \infty, \forall\, u, v \in \mathcal{H} \tag{A.157}$$

*There exists a bounded linear operator $T$ on $\mathcal{H}$ such that*

$$(u, v) = \langle u, Tv\rangle, \forall\, u, v \in \mathcal{H} \tag{A.158}$$

*where $\langle \bullet, \bullet \rangle$ is the inner product on $\mathcal{H}$.*

To understand this corollary, one defines a bounded linear functional by $(u, \bullet) : \mathcal{H} \to \mathbb{C}$. By the Riesz representation theorem, there exists another vector $T^\dagger(u) \in \mathcal{H}$ such that

$$(u, v) = \langle T^\dagger u, v\rangle \tag{A.159}$$

It is easy to see that $T^\dagger$ defined above is a linear operator. Hence its adjoint exists,

$$(u, v) = \langle T^\dagger u, v\rangle = \langle u, Tv\rangle \tag{A.160}$$

---

[22] We are not using bra-ket notations in this example because it makes everything messy.

**Lemma A.9** (Theorem 9.17 of Ref. [47]). *The GNS time evolution operator $e^{itH_{\mathrm{GNS}}} \in \pi_\psi(\mathcal{A}^{ql})''$, where $\psi$ is the ground state of $\delta_H$ and $\pi_\psi(\mathcal{A}^{ql})''$ means the double commutant of $\pi(\mathcal{A}^{ql})$ in $\mathcal{B}(\mathcal{H}_\psi)$ (see theorem A.4 for its definition).*

*Proof to theorem A.5.* Our proof here is rather heuristic and readers are refer to theorem A.3 of Ref. [61] for a more rigorous one. The technique is exactly the same.

First of all, by the definition of locally-unique ground state, other possible ground states fall into different superselection sectors. Hence $H_{\mathrm{GNS}}$ has a non-degenrate ground state $|\psi\rangle$.

Next, we show that $\psi$ is pure. To this end, we invoke definition A.11. Let $\rho : \mathcal{A}^{ql} \to \mathbb{C}$ be another positive linear functional, such that $\psi - \rho$ is again positive. We have to show that $\rho = \lambda\psi$ for some $\lambda \in [0, 1]$.

Firstly, we define a sesquilinear form on the GNS Hilbert space $\mathcal{H}_\psi$,

$$(|B\rangle, |A\rangle) := \rho(B^\dagger A) \tag{A.161}$$

By corollary A.6, there exists a linear operator $T_\rho$ on $\mathcal{H}_\psi$, such that

$$(|B\rangle, |A\rangle) = \langle B|T_\rho|A\rangle = \langle\psi|\pi_\psi(B)^\dagger T_\rho \pi_\psi(A)|\psi\rangle \tag{A.162}$$

Now note for any $A, B, C \in \mathcal{A}^{ql}$, we have

$$\langle\psi|\pi_\psi(B)^\dagger T_\rho \pi_\psi(C)\pi_\psi(A)|\psi\rangle = \rho(B^\dagger CA) = \rho((C^\dagger B)^\dagger A) = \langle\psi|\pi_\psi(C^\dagger B)T_\rho \pi_\psi(A)|\psi\rangle$$
$$= \langle\psi|\pi_\psi(B)^\dagger \pi_\psi(C)T_\rho \pi_\psi(A)|\psi\rangle \tag{A.163}$$

Hence $\pi_\psi(C)T_\rho = T_\rho \pi_\psi(C), \forall C \in \mathcal{A}^{ql}$. Equivalently, we have $T_\rho \in \pi_\psi(\mathcal{A}^{ql})'$. To show that $\psi$ is pure, it suffices to show $T_\rho = \lambda I$ for some $0 \leqslant \lambda \leqslant 1$. To this end, note that $H_{\mathrm{GNS}} \in \pi_\psi(\mathcal{A}^{ql})''$, especially $[H_{\mathrm{GNS}}, T_\rho] = 0$, so

$$H_{\mathrm{GNS}}T_\rho|\psi\rangle = T_\rho H_{\mathrm{GNS}}|\psi\rangle = 0 \tag{A.164}$$

Therefore, $T_\rho|\psi\rangle$ is another ground state of $H_{\mathrm{GNS}}$, which must be proportional to $|\psi\rangle$ since $H_{\mathrm{GNS}}$ is non-degenerate on $\mathcal{H}_\psi$. So $T_\rho|\psi\rangle = \lambda|\psi\rangle$ and hence $T_\rho = \lambda I$ on $\mathcal{H}_\psi$. So $\rho = \lambda\psi$. By assumption, $\rho$ is positive and $\rho \leqslant \psi$, thus we conclude $0 \leqslant \lambda \leqslant 1$, i.e., $\psi$ is pure.

$\square$

In the above proof, the condition that the Hamiltonian is admissible is used to show that the GNS Hamiltonian or its corresponding dynamics exists. In Sec. E 2, we will define a set of Hamiltonians known as nearly local Hamiltonians. These Hamiltonians are not admissible, but their GNS Hamiltonian and dynamics still exist. Hence there one can show that the locally-unique gapped ground states of nearly local Hamiltonians are also pure.

## B. Group cohomology and differentiable group cohomology

In this section, we review the basics of group cohomology and differentiable group cohomology. For group cohomology, there are many materials in the literature [62–64]. For differentiable group cohomology, see appendix A.1 of Refs. [25] and [65]. We will only cover the motivations and basics here.

### 1. Projective representations in quantum mechanics

To motivate group cohomology, we start with projective representations in quantum mechanics. Suppose we have a symmetry group $G$ (assumed to be unitary and discrete for simplicity) acting on a Hilbert space $\mathcal{H}$. Usually this symmetry action is given by a homomorphism $\rho : G \to U(\mathcal{H})$, i.e., a unitary representation of $\mathcal{H}$. More explicitly, for each $g \in G$, we assign a unitary operator $\rho(g)$ such that

$$\rho(g)\rho(h) = \rho(gh), \forall g, h \in G \tag{B.1}$$

However, in quantum mechanics, states are *not* really a vector in $\mathcal{H}$, but a *ray*. That means a state $|\psi\rangle$ is the same as $e^{i\theta}|\psi\rangle$ as a quantum state. Thus, the space of states is not literally $\mathcal{H}$, but the projective space $P(\mathcal{H})$. This for allows more general symmetry actions as

$$\rho(g)\rho(h) = \omega(g, h)\rho(gh) \tag{B.2}$$

where[23] $\omega(g,h) \in \mathrm{U}(1)$. This $\rho$ is a representation up to a phase $\omega$ and is called a projective representation. Moreover, the matrix multiplication is associative, so

$$(\rho(g)\rho(h))\rho(k) = \rho(g)(\rho(h)\rho(k)) \tag{B.3}$$

This imposes the following constraint on $\omega$,

$$\omega(g,h)\omega(gh,k) = \omega(g,hk)\omega(h,k) \tag{B.4}$$

Any function $G \times G \to \mathrm{U}(1)$ satisfying Eq. (B.4) is called a 2-cocycle. Furthermore, one can redefine the phase of $\rho(g) \to \tilde{\rho}(g) = \rho(g)\eta(g), \eta(g) \in \mathrm{U}(1)$ (we do not require $\eta : G \to \mathrm{U}(1)$ to be a homomorphism), and the resulting 2-cocycle is

$$\tilde{\omega}(g,h) = \omega(g,h)\eta(g)\eta(h)\eta(gh)^{-1} \tag{B.5}$$

One can easily check that $\tilde{\omega}$ again satisfies the 2-cocycle condition, Eq. (B.4). If there exists $\eta(g)$ such that $\tilde{\omega}(g,h) = 1$ for all $g, h \in G$, we say that $\omega$ is a 2-coboundary or trivial. Any two 2-cocycles $\omega$ and $\tilde{\omega}$ related by Eq. (B.5) are viewed as equivalent, since they differ only by the artificial choice of phase factors $\eta(g)$ of representation matrix $\rho(g)$. We write $\omega \sim \tilde{\omega}$ if $\omega$ and $\tilde{\omega}$ are equivalent. The space of 2-cocycles modulo this equivalence $\sim$ is the so-called the degree 2 group cohomology of $G$, denoted by $\mathrm{H}^2(G; \mathrm{U}(1))$.

**Example B.1.** *Let us consider $G = \mathbb{Z}_2 \times \mathbb{Z}_2$. We write its elements as $(a,b)$ where $a,b = 0,1 \mod 2$. Then we define a projective representation $\rho$ as follows:*

$$\begin{aligned} \rho(0,0) &= I, \ \rho(1,0) = \sigma_x \\ \rho(0,1) &= \sigma_y, \ \rho(1,1) = \sigma_z \end{aligned} \tag{B.6}$$

*Note that $\rho(0,1)\rho(1,0) = i\rho(1,1)$ hence $\omega((0,1),(1,0)) = i$. Similarly, $\omega((1,0),(0,1)) = -i$. One can show that this 2-cocycle is not a 2-coboundary and hence defines the nontrivial class in $\mathrm{H}^2(\mathbb{Z}_2 \times \mathbb{Z}_2; \mathrm{U}(1)) \simeq \mathbb{Z}_2$. In the context of symmetry-protected topological phases, this projective representation describes the boundary of the cluster state [67].*

**Example B.2.** *Consider the case $G = SO(3)$, the spin rotation symmetry [24]. One can show that $\mathrm{H}^2(SO(3); \mathrm{U}(1)) \simeq \mathrm{Hom}(\pi_1(SO(3)), \mathrm{U}(1)) \simeq \mathbb{Z}_2$, and this class is trivial if the (total) spin quantum number $S \in \mathbb{Z}$ and it is nontrivial if $S \in \mathbb{Z} + \frac{1}{2}$.*

A projective representation provides the following constraint on quantum states.

**Proposition B.1.** *If $G$ acts on the Hilbert space $\mathcal{H}$ via a projective representation $\rho$ whose associated 2-cocycle $\omega \neq 1 \in \mathrm{H}^2(G; \mathrm{U}(1))$, then there cannot be a nonzero $G$-symmetric state.*

*Proof.* Suppose $|\psi\rangle$ is a $G$-symmetric state, that is

$$\rho(g)|\psi\rangle = \eta(g)^{-1}|\psi\rangle \tag{B.7}$$

where $\eta(g) \in \mathrm{U}(1)$ is any $\mathrm{U}(1)$-valued function on $G$. Then one redefines $\tilde{\rho}(g) = \rho(g)\eta(g)$, this shifts $\omega$ by a 2-coboundary and the resulting $\tilde{\omega}$ (see Eq. (B.5)) is nontrivial, i.e., there exists $g, h \in G$ such that $\tilde{\omega}(g,h) \neq 1$. Now

$$\tilde{\rho}(g)|\psi\rangle = |\psi\rangle, \ \forall \, g \in G \tag{B.8}$$

One can calculate $\tilde{\rho}(g)\tilde{\rho}(h)|\psi\rangle$ in 2 different ways

$$\begin{aligned} \tilde{\rho}(g)(\tilde{\rho}(h)|\psi\rangle) &= \tilde{\rho}(g)|\psi\rangle = |\psi\rangle \\ (\tilde{\rho}(g)\tilde{\rho}(h))|\psi\rangle &= \tilde{\omega}(g,h)\tilde{\rho}(gh)|\psi\rangle = \tilde{\omega}(g,h)|\psi\rangle \end{aligned} \tag{B.9}$$

By assumption, $\tilde{\omega}(g,h) \neq 1$ for some $g, h \in G$. Hence $|\psi\rangle = 0$, which shows that there is no nonzero $G$-symmetric state. $\qquad\square$

---

[23] In principle, one has to show that the phase $\omega(g,h)$ is the same on each quantum state. This relies the coherence of these states and one can find the proof in Sec. 2.2 of Ref. [66].

[24] Actually, this a subtler case because $SO(3)$ is a Lie group so it requires more careful treatment, which will be left to later sections. We omit this subtlety for now.

As a corollary, consider a $G$-symmetric Hamiltonian $H$ which has a $G$ symmetry that acts projectively. We have

**Corollary B.1.** *Given $\rho$ is nontrivial projective representation, then a $G$-symmetric Hamiltonian must have degenerate ground states which break the $G$-symmetry.*

This can be viewed as $(0+1)d$ version of anomaly constraints.

**Example B.3.** *Consider a system made of $N$ qubits (or equivalently, spin $\frac{1}{2}$'s), whose Hamiltonian $H$ has a $G = SO(3)$ symmetry encountered in example B.2. If $N = 1 \mod 2$, then this system must be at least 2-fold degenerate. For example, consider $N = 1$, for the Hamiltonian $H$ to be $SO(3)$-symmetric, it has to commute with all Pauli operators. It is easy to check that $H$ must be $\lambda I$ for some $\lambda \in \mathbb{C}$ and $I$ is the identity operator. Hence the ground states are trivially 2-fold degenerate. However, for $N = 2$ where the total spin is an integer, one can take*

$$H = J\vec{S}_1 \cdot \vec{S}_2, J > 0 \tag{B.10}$$

*where the ground state is non-degnerate.*

## 2. Group cohomology

Now we present the definition of group cohomology in general. Let $G$ be a discrete group, one defines a space $BG$ which is a collection of spaces $\{G^n\}_{n=1,2,\dots}$ equipped with a collection of maps $d_k : G^n \to G^{n-1}$, $k = 0, 1, \dots, n$ (called face maps). Explicitly,

$$d_k(g_1, g_2, \dots, g_n) = \begin{cases} (g_2, \dots, g_n), k = 0 \\ (g_1, \dots, g_k g_{k+1}, \dots, g_n), 0 < k < n \\ (g_1, \dots, g_{n-1}), k = n \end{cases} \tag{B.11}$$

One can check that if $d = \sum_{k=0}^{n}(-1)^k d_k$, then $d^2 = 0$. Let $A$ be an Abelian group (with *discrete topology*). For example $A$ can be $\mathbb{Z}_2$, $\mathbb{Z}$, $\mathbb{R}$ or $U(1)$. We denote all $A$-valued functions on $BG$ as $C^\bullet(BG, A)$. For example, one writes $\omega \in C^2(BG, A)$ if $\omega : G^2 \to A$. Consider an $A$-valued function $\omega$ on $G^{n-1}$. The maps $d_k : G^n \to G^{n-1}$ induces a pullback of $\omega$, i.e., $d_k^*\omega := \omega \circ d_k$ on $G^n$. We denote $\delta = d^*$ (it follows that $\delta^2 = 0$), thus $C^\bullet(BG, A)$ together with $\delta$ becomes a cochain complex.

**Definition B.1.** *A function $\omega : G^n \to A$ is said to be an n-cocycle if $\delta\omega = 0$. We denote the space of all n-cocycles by $\mathrm{Z}^n(G; A)$. Besides, if an n-cocycle $\omega$ satisfies $\omega = \delta\eta$ for some $\eta \in C^{n-1}(G; A)$, it is called an n-coboundary. The space of all n-coboundary is denoted as $\mathrm{B}^n(G; A), n > 1$. Besides, $\mathrm{B}^1(G; A)$ is defined to be 0.*

**Definition B.2.** *The degree $n$ group cohomology of $G$ is defined to be*

$$\mathrm{H}^n(G; \mathrm{U}(1)) = \frac{\mathrm{Z}^n(G; A)}{\mathrm{B}^n(G; A)} \tag{B.12}$$

*In more details, $\mathrm{H}^n(G; A)$ are defined to be equivalence classes of n-cocycles under the equivalence relation $\omega \simeq \omega + \delta\eta$ where $\omega \in \mathrm{Z}^n(G; A)$ and $\delta\eta \in \mathrm{B}^n(G; A)$.*

**Example B.4.** *Let us consider a function $\omega : G \to A$ or equivalently $\omega$ here is a 1-cochain. Now we compute $\delta\omega$*

$$\delta\omega(g_1, g_2) = (d_0^*\omega - d_1^*\omega + d_2^*\omega)(g_1, g_2) = \omega(g_1) + \omega(g_2) - \omega(g_1 g_2) \tag{B.13}$$

*where we have used Eq. (B.11), e.g.,*

$$d_1^*\omega(g_1, g_2) = \omega(d_1(g_1, g_2)) = \omega(g_1 g_2) \tag{B.14}$$

*Then $\omega$ is a 1-cocycle iff it is a homomorphism, i.e., $\omega(g_1 g_2) = \omega(g_1) + \omega(g_2)$. We conclude*

$$\mathrm{H}^1(G; A) = \mathrm{Hom}(G, A) \tag{B.15}$$

**Example B.5.** *Now we consider a 2-cochain, again denoted by $\omega : G^2 \to \mathcal{A}$. Then one calculates $\delta\omega$ as follows*

$$\delta\omega(g_1, g_2, g_3) = \omega(g_2, g_3) - \omega(g_1 g_2, g_3) + \omega(g_1, g_2 g_3) - \omega(g_1, g_2) \tag{B.16}$$

*If one writes the group action in $A$ as multiplication rather than addition, one immediately recognizes $\delta\omega = 0$ is exactly the 2-cocycle condition Eq. (B.4) in projective representations. One can shift $\omega$ by a 2-coboundary $\delta\eta$. As we computed in the last example, this corresponds to*

$$\omega(g_1, g_2) \to \tilde{\omega}(g_1, g_2) = \omega(g_1, g_2) + \eta(g_1) + \eta(g_2) - \eta(g_1 g_2) \tag{B.17}$$

*In the context of projective representation, this amounts to redefining our representation matrices by a phase Eq. (B.5).*

Group cohomology of higher degrees are used to classify 't Hooft anomalies in physics. We will explain this in some more details in Sec. C.

**Remark B.1.** *The geometry behind Eqs. (B.11) and (B.12) is that we are doing simplicial cohomology on the space $BG$ (which is known as classifying space in mathematics), see Ref. [68] for example.*

### 3. Differentiable group cohomology

It is tempting to generalize the above definition of group cohomology of discrete to group cohomology of Lie group. Naively, we should require group cochains to be smooth, i.e., $\omega : G^n \to A$ should be a smooth function on $G^n$ where $A$ is an abelian Lie group. We denote the cohomology of smooth cochains by $H_s^*(G; A)$. However, this definition does not work due to the van Est theorem, which says that for connected compact Lie group $G$ we have [69]

$$H_s^n(G; A) = 0, n > 0 \tag{B.18}$$

Moreover, if $G$ is compact but not connected, we have

$$H_s^n(G; A) \simeq H^n(\pi_0(G); A), n \geqslant 0 \tag{B.19}$$

This means that this cohomology group does not capture the smooth structure of $G$ at all.

Roughly speaking, there are three different ways to define useful cohomology theory for Lie groups [70].

1. The first is to use measurable cochains rather than smooth ones. The resulting cohomology is known as the Borel group cohomology in the physics literature[25], see Refs. [16, 64]. Following the convention in the physics literature, we denote this cohomology by $H_B^*(G; A)$.

2. The second way is to replace smooth cochains by *locally* smooth cochains. By locally smooth we mean that $\omega : G^n \to A$ is smooth in a neighborhood of $(1, 1, ..., 1)$. We denote this cohomology theory by $H_{loc,s}^*(G; A)$.

3. The last way is to use simplicial method. Intuitively, one fixes a set of charts $\{U_i\}_{i \in J}$ on the Lie group $G$, and cochains are defined as smooth functions on each of trivialization chart $U_i$. On intersections, one needs *transition functions* to patch them together. The resulting cohomology theory is denoted as $H_{\text{diff}}^*(G; A)$, which is exactly the same as $H_{\text{simp},s}^*(G; A)$ in Ref. [70].

Let us look at an example of differential group cohomology.

**Example B.6.** *Suppose $\rho$ is a smooth projective representation of $G$ on a finite dimensional Hilbert space $V$, i.e., $\rho : G \to \mathrm{PU}(V) := \mathrm{U}(V)/\mathrm{U}(1)$. On each trivialization chart $U_i$ of $G$, one can lift $\rho$ to be $\rho_i : U_i \to \mathrm{U}(V)$, which may not be a representation of $G$ in general. On each chart,*

$$\rho_i(g)\rho_i(h) = \rho_i(gh)\omega_i(g, h) \tag{B.20}$$

*where $\omega_i(g, h) \in \mathrm{U}(1)$ and $g, h, gh \in U_i$. Of course, $\omega_i$ is further constrained by the usual 2-cocycle condition Eq. (B.4). On the intersection $U_i \cap U_j$, one notes that two different liftings at most differ by a phase, i.e.,*

$$\rho_i(g) = \rho_j(g)\eta_{ij}(g) \tag{B.21}$$

---

[25] We remark that this cohomology has nothing to do with the so-called Borel equivariant cohomology, which is often referred to as Borel cohomology in the mathematical literature.

where $\eta_{ij}(g) \in \mathrm{U}(1)$ and $g \in U_i \cap U_j$. Thus we have

$$\omega_i(g, h) = \delta\eta_{ij}(g, h)\omega_j(g, h) \tag{B.22}$$

We say $[\omega, \eta]$ defines a differentiable group cohomology class in $\mathrm{H}^2_{\mathrm{diff}}(G; \mathrm{U}(1))$. Note that here $\eta_{ij}$ plays the role of transition functions in usual bundle theory.

A priori, these cohomology groups may not be the same. However, it turns out that they are isomorphic for finite dimensional Lie groups with suitable coefficients.

**Theorem B.1** (Corollary IV.9 and remark IV.13 of Ref. [70]). *For finite dimensional Lie group $G$ which acts smoothly on $\mathrm{U}(1)$, we have*

$$\mathrm{H}^*_{loc,s}(G; \mathrm{U}(1)) \simeq \mathrm{H}^*_{\mathrm{B}}(G; \mathrm{U}(1) \simeq \mathrm{H}^*_{\mathrm{diff}}(G; \mathrm{U}(1)) \tag{B.23}$$

Despite of being isomorphic, $\mathrm{H}^*_{\mathrm{diff}}(G; \mathrm{U}(1))$ is more convenient for constructing the anomaly index (see Ref. [25]). There are some useful properties of $\mathrm{H}_{\mathrm{diff}}$. We list them here and the readers are referred to Refs. [65, 70] for proofs.

**Proposition B.2.** *If either $G$ or $A$ is discrete, then*

$$\mathrm{H}^*_{\mathrm{diff}}(G; A) \simeq \mathrm{H}^*(BG; A) \tag{B.24}$$

*where the right hand side is the singular cohomology of the classifying space $BG$.*

**Proposition B.3.** *If $G$ is compact*

$$\mathrm{H}^n_{\mathrm{diff}}(G; \mathbb{R}) = 0, n > 0 \tag{B.25}$$

By Bockstein homomorphism,

**Corollary B.2.** *For compact Lie group $G$, we have*

$$\mathrm{H}^n_{\mathrm{diff}}(G; \mathrm{U}(1)) \simeq \mathrm{H}^{n+1}(BG; \mathbb{Z}), \ n \geqslant 1 \tag{B.26}$$

**Proposition B.4** (Kunneth formula, Appendix B of Ref. [71]). *Let $G$ and $H$ be finite-dimensional Lie groups (including discrete groups) then*

$$\mathrm{H}^n_{\mathrm{diff}}(G \times H; \mathrm{U}(1)) \simeq \bigoplus_{p+q=n} \mathrm{H}^p_{\mathrm{diff}}(G; \mathrm{H}^q_{\mathrm{diff}}(H; \mathrm{U}(1))) \tag{B.27}$$

## C. Construction of anomaly index

### 1. Decomposition

We now present the construction of the anomaly index in Ref. [25]. The spirit of this construction is similar to Ref. [36], i.e., we need to cut our chain at the origin and decompose the symmetry action . This cut induces a factorization on $\mathcal{A}^{ql}$, such that

$$\mathcal{A}^{ql} \simeq \mathcal{A}^{ql}_{<0} \otimes \mathcal{A}^{ql}_{\geqslant 0} \tag{C.1}$$

To study how symmetry action decomposes under this cut, it is useful to define some more subgroups of $\mathcal{G}^{lp}$, the group of all symmetry actions.

**Definition C.1.** *We define the following useful subgroups of $\mathcal{G}^{lp}$,*

1. $\mathcal{G}^{lp}_0$: *The subgroup generated by $\mathrm{Ad}_U$, where $U$ is a quasi-local unitary operators.*

2. $\mathcal{G}^{lp}_{<0}$: *The subgroup that acts trivially on $\mathcal{A}^{ql}_{\geqslant 0}$ and maps $\mathcal{A}^{ql}_{<0}$ to itself.*

3. $\mathcal{G}^{lp}_{\geqslant 0}$: *The subgroup acts trivially on $\mathcal{A}_{<0}$ and maps $\mathcal{A}^{ql}_{\geqslant 0}$ to itself.*

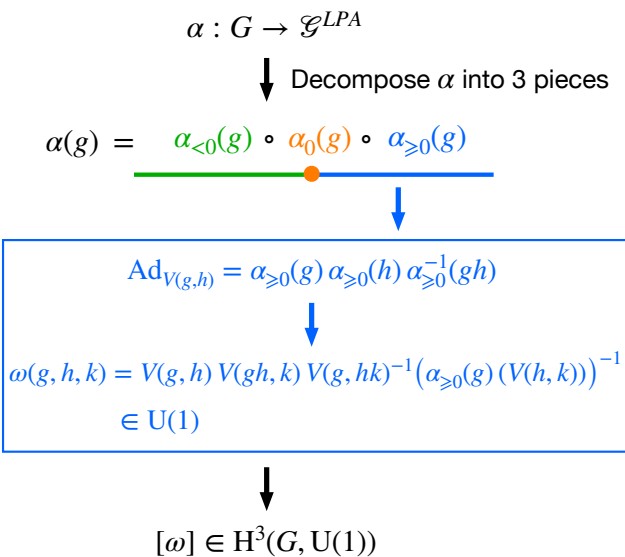

FIG. 3. The anomaly index $\omega \in \mathrm{H}^3(G; \mathrm{U}(1))$ from decomposing the symmetry action.

4. $\mathcal{G}_+^{lp}$ (resp. $\mathcal{G}_-^{lp}$) is the subgroup generated by $\mathcal{G}_{\geqslant 0}^{lp}\mathcal{G}_0^{lp}$ (resp. $\mathcal{G}_{<0}^{lp}\mathcal{G}_0^{lp}$).

Note that $\mathcal{G}_{\geqslant 0}^{lp}$ and $\mathcal{G}_{<0}^{lp}$ act on $\mathcal{G}_0^{lp}$ by conjugation. More explicitly, for $\alpha \in \mathcal{G}_{\geqslant 0}^{lp}$ or $\alpha \in \mathcal{G}_{<0}^{lp}$ and $U$ a quasi-local unitary,

$$\alpha \triangleright \mathrm{Ad}_U := \alpha \mathrm{Ad}_U \alpha^{-1} = \mathrm{Ad}_{\alpha(U)} \tag{C.2}$$

To accomplish the decomposition in Fig. 3, we need a few lemmas in Ref. [25]. Readers are referred to Ref. [25] for rigorous proofs of these lemmas. Here we only state the lemmas and comment on why some of these lemmas are intuitively true.

**Lemma C.1** (Lemma 2.1 of Ref. [25]). *Let $\alpha \in \mathcal{G}^{lp}$ be an LPA, then the following statements are equivalent,*

  1. *The GNVW index (see Sec.A 2 for a brief review) $\mathrm{ind}(\alpha) = 0$.*

  2. *The element $\alpha$ admits following decomposition*

$$\alpha = \alpha_{<0}\alpha_0\alpha_{\geqslant 0} \tag{C.3}$$

Given the decomposition Eq. (C.1), one may want to restrict the symmetry action $\alpha$ to each half chain, which gives the $\alpha_{\geqslant 0}$ and $\alpha_{<0}$ parts. However, generically $\alpha$ can expand the support of an operator. If an operator is supported on one of the two half chains, under $\alpha$ it will generically acquire some support on the other half chain. But $\alpha_{\geqslant 0}$ and $\alpha_{<0}$ cannot achieve this, so the $\alpha_0$ part is also expected.

**Lemma C.2** (Lemma 2.2 of Ref. [25]). *Suppose $\alpha \in \mathcal{G}^{lp}$ has a vanishing GNVW index and it admits two different decompositions*

$$\alpha = \alpha_{<0}\alpha_0\alpha_{\geqslant 0} = \tilde{\alpha}_{<0}\tilde{\alpha}_0\tilde{\alpha}_{\geqslant 0} \tag{C.4}$$

*then $\alpha_{<0}\tilde{\alpha}_{<0}^{-1} \in \mathcal{G}_0^{lp}$ and $\alpha_{\geqslant 0}\tilde{\alpha}_{\geqslant 0}^{-1} \in \mathcal{G}_0^{lp}$.*

The following lemma is also often useful when dealing with $\mathcal{G}_+^{lp}$ and $\mathcal{G}_-^{lp}$.

**Lemma C.3** (Corollary 2.1 of Ref. [25]). *The intersection $\mathcal{G}_+^{lp} \cap \mathcal{G}_-^{lp} = \mathcal{G}_0^{lp}$*

Note that any $\alpha_+ \in \mathcal{G}_+^{lp}$ can be uniquely written as

$$\alpha_+ = \alpha_0\alpha_{\geqslant 0}, \ \alpha_0 \in \mathcal{G}_0^{lp}, \ \alpha_{\geqslant 0} \in \mathcal{G}_{\geqslant 0}^{lp} \tag{C.5}$$

Note $\mathrm{Ad}_U\alpha_{\geqslant 0} = \alpha_{\geqslant 0}\mathrm{Ad}_{\alpha(U)}$. Similarly, one can always write $\beta_- \in \mathcal{G}_-^{lp}$ as $\beta_0\beta_{<0}, \beta_0 \in \mathcal{G}_0^{lp}, \beta_{<0} \in \mathcal{G}_{<0}^{lp}$. So one conclude the third lemma by comparing these two.

**Lemma C.4** (Remark 2.3 of Ref. [25]). *The subgroups $\mathcal{G}_+^{lp}$ and $\mathcal{G}_-^{lp}$ are both normal subgroups of $\mathcal{G}^{lp}$.*

## 2. Anomaly index from decomposition

With the above background, now we can perform the decomposition (see Fig. 3) and derive the anomaly index.

Let $\alpha : G \to \mathcal{G}^{lp}$ (it is assumed to be smooth if $G$ is a Lie group) be a symmetry action implemented by LPA, i.e.,

$$\alpha(g)\alpha(h) = \alpha(gh), \forall\, g, h \in G \tag{C.6}$$

We first assume that $\mathrm{ind}(\alpha(g)) = 0, \forall\, g \in G$. Thus, for each $g$, it admits the following decomposition as in Eq. (C.3)

$$\alpha(g) = \alpha(g)_{<0}\alpha(g)_0\alpha(g)_{\geqslant 0} \tag{C.7}$$

where $\alpha(g)_{<0} \in \mathcal{G}^{lp}_{<0}, \alpha(g)_0 \in \mathcal{G}^{lp}_0$ and $\alpha(g)_{\geqslant 0} \in \mathcal{G}^{lp}_{\geqslant 0}$. On the other hand, note that $\alpha(g)_{\geqslant 0}$ trivially commutes with $\alpha(h)_{<0}$ because they act on disjoint domains,

$$\alpha(g)_{<0}\alpha(g)_0\alpha(g)_{\geqslant 0}\alpha(h)_{<0}\alpha(h)_0\alpha(h)_{\geqslant 0} = \alpha(g)_{<0}\alpha(h)_{<0}\beta_0(g,h)\alpha(g)_{\geqslant 0}\alpha(h)_{\geqslant 0} \tag{C.8}$$

where $\beta_0(g,h) := (\alpha_{<0}(h) \triangleright \alpha_0(g))(\alpha_{\geqslant 0}(g) \triangleright \alpha_0(h)) \in \mathcal{G}^{lp}_0$ (see Eq. (C.2) for definition of group action $\triangleright$). According to lemma C.2, there exists a map $V : G \times G \to \mathcal{U}^{ql}$ (which may *not* be a homomorphism) such that

$$\alpha(g)_{\geqslant 0}\alpha(h)_{\geqslant 0} = \mathrm{Ad}_{V(g,h)}\alpha(gh)_{\geqslant 0} \tag{C.9}$$

Furthermore, this $V$ is constrained by the associativity of $\alpha_{\geqslant 0}$ as follows

$$\alpha(g)_{\geqslant 0}(\alpha(h)_{\geqslant 0}\alpha(k)_{\geqslant 0}) = (\alpha(g)_{\geqslant 0}\alpha(h)_{\geqslant 0})\alpha(k)_{\geqslant 0} \tag{C.10}$$

The above equation further simplifies to

$$\mathrm{Ad}_{\omega(g,h,k)} = 1 \tag{C.11}$$

where

$$\omega(g,h,k) = V(g,h)V(gh,k)V(g,hk)^{-1}(\alpha_{\geqslant 0}(g)(V(h,k)))^{-1} \tag{C.12}$$

Since $\omega(g,h,k)$ commutes with all quasi-local operators, it must be a pure phase i.e., $\omega(g,h,k) \in \mathrm{U}(1)$. It can be checked that $\omega$ satisfies the 3-cocycle condition (a.k.a pentagon identity) and shifting $V(g,h)$ by a phase $\rho(g,h) \in \mathrm{U}(1)$ will change $\omega$ by a 3-coboundary (see appendix B of Ref. [25] for more details). Thus $\omega : G^3 \to \mathrm{U}(1)$ is a well-defined degree 3 group cohomology class, i.e., $[\omega] \in \mathrm{H}^3(G; \mathrm{U}(1))$ (see Sec. B for a review of group cohomology).

If GNVW index of $\alpha(g)$ is nontrivial, $\alpha$ induces a map $\tau : G \xrightarrow{\alpha} \mathcal{G}^{lp} \to \mathcal{G}^T$ where $\mathcal{G}^T$ is the group of generalized translations. In this case, we stack our system with another (decoupled) copy, on which the $G$-symmetry acts as $\tau(g)^{-1}$. On the composite system, this $G$-symmetry acts as

$$\beta(g) := \alpha(g) \otimes \tau(g)^{-1} \tag{C.13}$$

This is again a group action thanks to the fact that $\mathcal{G}^T$ is abelian. Besides, $\mathrm{ind}(\beta(g)) = 0$, which allows us to define

$$\omega_\alpha := \omega_\beta \tag{C.14}$$

This construction *internalizes* the translation symmetry.

We call $\omega$ the anomaly index of the symmetry action $\alpha$. If $\omega \neq 1 \in \mathrm{H}^3(G; \mathrm{U}(1))$, we say that the symmetry action $\alpha$ is anomalous, otherwise $\alpha$ is said to be anomaly-free or non-anomalous.

Although the cohomology class of $\omega$ does not depend on the choice of $\alpha_{\geqslant 0}$ and $V$ [25], in the previous work such as Refs. [25, 36], it is not clear if the cohomology class of $\omega$ depends on where we cut the chain. For example, one can also cut the chain at 1 rather than 0, do they yield the same anomaly index? Below we show that the anomaly index is independent of the choice of the cut.

**Proposition C.1.** *Assume the dimension of the local Hilbert space $\mathcal{H}_k$ is bounded by a constant $K$, i.e., $\dim \mathcal{H}_k < K$ for any $k \in \mathbb{Z}$. Then the anomaly index $\omega$ constructed above does not depend on the choice of the cut.*

*Proof to proposition C.1.* If all local Hilbert spaces are of the same dimension, we have an operation which is translation by $+1$, denoted by $\tau$. Otherwise, we add some decoupled degrees of freedom at each site, on which our symmetry acts trivially. This step is to ensure that all local Hilbert spaces have the same dimension. After this prescription, we again have a well defined translation operation.

Define $\tilde{\alpha}(g) := \tau\alpha(g)\tau^{-1}$, then the decomposition of $\tilde{\alpha}$ at 0 gives a decomposition of $\alpha$ at 1. We fix a decomposition of $\tilde{\alpha}$ at 0 as

$$\tilde{\alpha} = \tilde{\alpha}_{<0}\tilde{\alpha}_0\tilde{\alpha}_{\geqslant 0} \tag{C.15}$$

Similarly, we fix a decomposition of $\alpha$ at 0 as

$$\alpha = \alpha_{<0}\alpha_0\alpha_{\geqslant 0} \tag{C.16}$$

We only have to show that they give rise to the same anomaly index. To this end, note that by lemma C.4, there exist $\beta_0' \in \mathcal{G}_0^{lp}$ and $\beta_{\geqslant 0} \in \mathcal{G}_{\geqslant 0}^{lp}$ such that

$$\tau\alpha_{\geqslant 0}\tau^{-1} = \beta_0'\beta_{\geqslant 0} \tag{C.17}$$

Similarly,

$$\tau\alpha_{<0}\tau^{-1} = \beta_{<0}\beta_0'' \\ \tau\alpha_0\tau^{-1} = \beta_0''' \tag{C.18}$$

So

$$\tilde{\alpha}_{<0}\tilde{\alpha}_0\tilde{\alpha}_{\geqslant 0} = \tilde{\alpha} = \beta_{<0}\beta_0\beta_{\geqslant 0} \tag{C.19}$$

where $\beta_0 := \beta_0''\beta_0'''\beta_0'$. Thus, we have two different decompositions for $\tilde{\alpha}$ at 0. Combining this result, the lemma 2.2 of Ref. [25] and proposition 3.1 of Ref. [25], we conclude that the two decompositions give the same anomaly index. $\square$

**Corollary C.1** (Edgeablity [72, 73])**.** *If $\omega \neq 1 \in \mathrm{H}^3(G; \mathrm{U}(1))$, then this spin chain is not edgeable, i.e., one cannot put this chain on a manifold with boundaries while preserving the $G$-symmetry.*

*Proof.* We imagine the chain with boundary is a half chain obtained by cutting an infinite chain at some point (say, 0). Then the symmetry acts on the half chain by

$$\alpha(g)_{\geqslant 0}\alpha(h)_{\geqslant 0} = \mathrm{Ad}_{V(g,h)}\alpha(gh)_{\geqslant 0} \tag{C.20}$$

To preserve the $G$ symmetry on this half chain, one needs $\alpha_{\geqslant 0}$ to be a group homomorphism, i.e., $\mathrm{Ad}_{V(g,h)}$ is trivial. Hence $V(g,h)$ commutes with all local operators, and it must be a phase. Thus, by definition

$$\omega(g,h,k) = (\delta V)(g,h,k) \tag{C.21}$$

This means the anomaly index must vanish. In other words, if $\omega \neq 1$, the chain cannot have a symmetric boundary condition.

However, one can still refine $V(g,h) \to V(g,h)\eta(g,h)$ where $\eta \in \mathrm{H}^2(G; \mathrm{U}(1))$, which leaves the relation $\omega = \delta V$ invariant. That is, the symmetric boundary conditions is a module over $\mathrm{H}^2(G; \mathrm{U}(1))$. One can understand this shift as stacking a 1d SPT phase to our spin chain. $\square$

**Remark C.1.** *Note that even when the anomaly index vanishes, i.e., $\omega = 1 \in \mathrm{H}^3(G; \mathrm{U}(1))$, generally there are non-symmetric boundary conditions, i.e., we may not be able to choose $V(g,h)$ to be a phase sometimes.*

## D. Proof of theorem III.1

Now we are ready to prove our theorem III.1 in the main text.

**Theorem D.1** (Theorem III.1 in the main text)**.** *Let the $G$-symmetry act on a quantum spin chain via a locality-preserving automorphisms (see Eq. (A.22)). Suppose this symmetry action has a non-vanishing anomaly index and $H$ is an admissible Hamiltonian (see Eq. (A.129)) that commutes with the $G$-action. Then there cannot be a locally-unique gapped $G$-symmetric ground state of $H$.*

To prove this theorem, we use the following established lemmas.

**Lemma D.1** (Sec. II of Ref. [37]). *If $\psi$ is a gapped ground state of an admissible Hamiltonian $H$, then the entanglement entropy of $\psi$ satisfies the area law (see definition A.18).*

**Lemma D.2** (Theorem 1.5 of Ref. [30]). *If the entanglement entropy of a state $\psi$ satisfies the area law, then it splits at every site (see definition A.16).*

Combining lemmas D.1 and D.2, one deduces that

**Corollary D.1.** *If $\psi$ is a gapped ground state of an admissible Hamiltonian $H$, then it splits at every site.*

The final lemma is

**Lemma D.3** (Theorem 2 and remark 4.1 of Ref. [25]). *Given a symmetry action $\alpha : G \to \mathcal{G}^{lp}$ (or $\mathcal{G}^{al}$ for continuous $G$) on a quantum spin chain. If there exists a pure state $\psi$ which splits at any site and is $G$-symmetric, then the associated anomaly index $\omega = 1$.*

*Proof to theorem III.1.* By corollary D.1 and lemma D.3, we only have to show the locally-unique gapped ground state of an admissible $H$ is pure and this is exactly the content of theorem A.5. $\square$

## E. Thermodynamic limit of finite-size systems and entanglement properties

Many of our theorems are proved for infinite systems, but real physical systems are all finite. In this section, we bridge finite and infinite systems together and use our results to extract some important implications on finite systems. In particular, we will prove the theorems 2 and 3 in the main text.

### 1. Thermodynamic limit of the unique gapped ground state of admissible Hamiltonians

Let $\{H_L\}$ be a sequence of admissible Hamiltonians (see Eq. (A.129) for its definition) defined on a finite system, with each $H_L$ having a size $L$. We say that this sequence of Hamiltonians is convergent if for any local operator $A$,

$$\lim_{L \to \infty} [H_L, A] = \delta_H(A) \tag{E.1}$$

where $\delta_H$ is a Hamiltonian on an infinite chain. If for each $L$, $H_L$ has a uniquely gapped ground state $|\psi_L\rangle$ with a gap $\Delta_L > 0$, then one can define a state $\psi$ on infinite chain

$$\psi(A) := \lim_{L \to \infty} \langle \psi_L | A | \psi_L \rangle \tag{E.2}$$

Of course, this limit may not exist. But the Banach-Alaoglu theorem ensures that there always exists a convergent subsequence, i.e., a sequence $L_n$ which increases to $\infty$ such that $\lim_{L_n \to \infty} \langle \psi_{L_n} | A | \psi_{L_n} \rangle$ exists for all local operators $A$ [61]. Below we only focus on this particular subsequence.

Our here goal is to show the following (theorem 2 in the main text)

**Theorem E.1.** *The state $\psi$ defined by Eq. (E.2) is a locally-unique gapped ground state of $\delta_H$, if $|\psi_L\rangle$ is the unique gapped ground state of $H_L$ with $\Delta_L > \Delta > 0$ for all $L$.*

*Proof.* Without loss of generality, let $A$ be a local operator such that $\psi(A) = 0$. By definition A.22, it suffices to show that

$$\psi(A^\dagger \delta_H(A)) \geqslant \Delta \psi(A^\dagger A) \tag{E.3}$$

To proceed, we denote $\psi_L(A) := \langle \psi_L | A | \psi_L \rangle$ and $A_L := A - \psi_L(A)$. By assumption,

$$\psi_L(A_L^\dagger [H_L, A_L]) \geqslant \Delta \psi_L(A_L^\dagger A_L) \tag{E.4}$$

Now we show that both sides of Eq. (E.4) converge to corresponding sides of Eq. (E.3) as $L \to \infty$, which shows that $\psi$ is a locally-unique gapped ground state of $\delta_H$.

For the left hand side, we have

$$
\begin{aligned}
&|\psi(A^\dagger \delta_H(A)) - \psi_L(A_L^\dagger[H_L, A_L])| \\
=&|\psi(A^\dagger(\delta_H - \delta_{H_L})(A)) + \psi(A^\dagger \delta_{H_L}(A)) - \psi_L(A_L^\dagger[H_L, A_L])| \\
=&|\psi(A^\dagger(\delta_H - \delta_{H_L})(A)) + \psi(A^\dagger \delta_{H_L}(A)) - \psi_L((A^\dagger - \psi_L(A)^*)[H_L, A - \psi_L(A)])| \\
\leqslant&|\psi(A^\dagger(\delta_H - \delta_{H_L})(A))| + ||\psi - \psi_L|| \cdot ||A^\dagger[H_L, A]|| + |\psi_L(\psi_L(A))^*[H_L, A])|
\end{aligned}
\tag{E.5}
$$

The third term vanishes because $\psi_L(A) = 0$ in the limit $L \to \infty$ by definition. The second term goes to zero since $\psi_L$ converges to $\psi$ and $||A^\dagger[H_L, A]||$ is bounded (since $||A^\dagger(\delta_H - \delta_{H_L})(A)|| \leqslant ||A^\dagger|| \cdot ||(\delta_H - \delta_{H_L})(A)|| \to 0$ and $||A^\dagger \delta_H(A)|| < \infty$ since $A \in \mathcal{A}^l$ and $\delta_H(A) \in \mathcal{A}^{ql}$). So we only have to take care of the first term, which satisfies

$$
|\psi(A^\dagger(\delta_H - \delta_{H_L})(A))| \leqslant ||A^\dagger|| \cdot ||\delta_H(A) - \delta_{H_L}(A)||
\tag{E.6}
$$

Here $||A^\dagger||$ is finite since $A$ is local. The last factor goes to zero since $\{H_L\}$ is a convergent sequence. Therefore, the left hand side of Eq. (E.4) converges to the left hand side of Eq. (E.3).

Treated in a similar manner, the right hand side of Eq. (E.4) can be shown to converge to the right hand side of Eq. (E.3).

$\qquad\qquad\qquad\qquad\qquad\qquad\qquad\qquad\qquad\qquad\qquad\qquad\qquad\qquad\qquad\qquad\qquad\qquad\qquad\qquad\qquad\quad\square$

Hence by lemma D.1, lemma D.2 and theorem A.5, we obtain

**Corollary E.1.** *The state $\psi$ defined by Eq. (E.2) is pure and splits at every site.*

If we have a sequence of states, which are $G$-symmetric, then the state of infinite chain defined by above limit is again $G$-symmetric. This can be seen as follows. For a finite chain with size $L$, we write $\alpha_L(g)(A) = \rho_L(g)A\rho_L^{-1}(g)$, where $\rho_L$ is a representation of $G$. Let $\alpha : G \to \mathcal{G}^{lp}$ be a symmetry on the infinite chain that a sequence of finite chains converge to, then we have $\lim_{L\to\infty} \alpha_L(g)(A) = \alpha(g)(A)$ for all local operators $A$. So

$$
\psi(\alpha(g)(A)) = \lim_{L\to\infty} \langle \psi_L | \rho_L(g)(A)\rho_L(g)^{-1} | \psi_L \rangle = \lim_{L\to\infty} \langle \psi_L | A | \psi_L \rangle = \psi(A)
\tag{E.7}
$$

We conclude $\psi = \psi \circ \alpha(g)$ and hence $\psi$ is $G$-symmetric.

## 2. Thermodynamic limit of short-range entangled (SRE) states

In order to talk about the thermodynamic limit of an SRE state, we have to define the notion of a sequence of SRE states. We will focus on 1d but most of our discussion can be generalized to higher dimensions unless otherwise specified.

Recall that by definition, in a finite-size system, an SRE state can be deformed to a product state by a finite-time evolution generated by some (potentially time dependent) local Hamiltonian. On the other hand, in finite-size systems, each product state (see definition A.17) is the unique gapped ground state of a single-body Hamiltonian, i.e., $H_0 = \sum_i h_i$ where $h_i$ is only supported on site $i$.

Therefore, in finite systems, any SRE state is the unique gapped ground state of the following type of Hamiltonian

$$
H = \sum_j \tau(h_j)
\tag{E.8}
$$

where $\tau$ is the constant time evolution of a local Hamiltonian. Note that $\tau(h_j)$ is *not* local in general. At far distances, it is controlled by the improved Lieb-Robinson bound [49, 50], i.e.,

$$
||[\tau(h_j), B]|| \leqslant C_1 ||h_j|| \cdot ||B|| e^{-C_2 r}
\tag{E.9}
$$

for $B \in \mathcal{A}^{ql}_{B(j,r)^c}$, where $B(j, r)^c$ is the complement of the ball $B(j, r) := [j - r, j + r]$ and $C_1, C_2 > 0$ are constants, which do not depend on the support of $B$. The following lemma gives a more direct characterization of $\tau(h_j)$.

**Lemma E.1** (Lemma 2.4 of Ref. [52]). *Let $\mathcal{A}, \mathcal{B} \subset \mathcal{B}(\mathcal{H})$ be two $C^*$-algebras. If $\mathcal{A} \overset{\epsilon}{\subset} \mathcal{B}'$ is a near inclusion[26], then*

$$
||[a, b]|| \leqslant 2\epsilon ||a|| \cdot ||b||
\tag{E.10}
$$

---

[26] Here $\mathcal{B}'$ means the commutant of $\mathcal{B}$, which is defined as $\mathcal{B}' := \qquad \{x \in \mathcal{B}(\mathcal{H}) | xb = bx, \forall b \in \mathcal{B}\}$.

*for any $a \in \mathcal{A}$ and $b \in \mathcal{B}$. Conversely, if $\mathcal{B}$ is a hyperfinite von-Neumann algebra[27] and*

$$||[a,b]|| \leqslant \epsilon ||a|| \cdot ||b|| \tag{E.11}$$

*then one has $\mathcal{A} \overset{\epsilon}{\subset} \mathcal{B}'$.*

See def. A.7 for the meaning of $a \overset{\epsilon}{\in} \mathcal{B}'$. Choose $\mathcal{A} = \tau(\mathcal{A}_i^l)$ and $\mathcal{B} = \mathcal{A}_{B(j,r)^c}^{ql}$, where $\mathcal{A}_i^l$ means the operators supported on site $i$. Note we have $\mathcal{B}' = \mathcal{A}_{B(j,r)}^l$. By lemma E.1, this amounts to saying that there exists a local operator $A_{j,r} \in \mathcal{A}_{B(j,r)}^l$ such that

$$||\tau(h_j) - A_{j,r}|| < f(r)||h_j|| \tag{E.12}$$

where $f(r) = C_1 e^{-C_2 r}$. Roughly speaking, this Hamiltonian Eq. (E.8) is not local since each term $h_j$ may have arbitrarily large support, although it *decays* exponentially at far distances so we say the Hamiltonian Eq. (E.8) has exponential tails. This is not an admissible Hamiltonian in Eq. (A.129), since we require at most finite-body interaction there. But the Hamiltonian Eq. (E.8) is indeed very special. It is an *almost-local Hamiltonian* defined in Refs. [74, 75]. Very roughly, for almost local Hamiltonians we allow infinite-body interactions, as long as they decay as their ranges increase faster than any polynomial functions of their ranges. Readers are referred to Refs. [25, 74, 75] for more details about almost local observables and almost local Hamiltonians.

For our purpose, it is useful to define

**Definition E.1** (Nearly local Hamiltonians). *A Hamiltonian $H = \sum_{j \in \mathbb{Z}} \tau(h_j)$ is said to be nearly local if each $h_j$ is only supported on site $j$ and $\tau$ is a constant time evolution by some local Hamiltonian.*

From above discussion we conclude that every nearly local Hamiltonian is an almost local Hamiltonian with exponential tails.

On a finite chain of size $L$, we can also define a nearly local Hamiltonian $H_L$, where each term in $H_L$ is obtained from a single-body Hamiltonian by an evolution $\tau$ (same as in above definition). If each $h_j$ in $H_L$ is gapped, $H_L$ has a unique ground state that is SRE. Conversely, as mentioned before, all SRE pure states in finite systems can be viewed as the unique gapped ground state of some nearly local Hamiltonian. Recall that we say $H_L$ converges to $H$ in the $L \to \infty$ limit if for any local operator $A$ on this finite chain we have

$$\lim_{L \to \infty} \delta_{H_L}(A) = \delta_H(A) \tag{E.13}$$

Now we define a sequence of SRE states as follows.

**Definition E.2** (Sequence of SRE states). *Let $H_L$ be a convergent sequence of nearly local Hamiltonians. If $\{|\psi_L\rangle\}$ is a sequence of states such that for each $L$, $|\psi_L\rangle$ is the unique gapped ground state of $H_L$ with a gap $\Delta_L \geqslant \Delta > 0$. Then $\{|\psi_L\rangle\}$ is called a sequence of SRE states.*

**Lemma E.2.** *Let $\{|\psi_L\rangle\}$ be a sequence of SRE states for some nearly local Hamiltonian $H_L$. For any local operator $A$, one defines*

$$\psi(A) := \lim_{L \to \infty} \langle \psi_L|A|\psi_L\rangle \tag{E.14}$$

*If this limit exists[28] for all such local operator $A$, then $\psi$ is a locally unique gapped ground state (see definition A.22) of $H$.*

The proof is identical to theorem E.1 and will be omitted here.

The following lemma is useful

**Lemma E.3** (Proposition E.2 of Ref. [75]). *The time evolution generated by an almost local Hamiltonian exists, in the sense of theorem A.4.*

————

[27] We will not give a rigorous definition on hyperfinite von-Neumann algebra, readers are referred to Ref. [52] for a definition. In particular, this lemma holds for $\mathcal{B} = \mathcal{A}_\Gamma^{ql}$ for any $\Gamma$ (finite or infinite), see Ref. [52].

[28] Of course it can happen that this limit may not exist. How-ever, Banach-Alaoglu theorem ensures that there is always a convergent subsequence. We restrict to such a subsequence if necessary.

Now we combine this lemma with theorem A.5. Note that the proof of theorem A.5 only requires locally-unique gapped ground state and the existence of time evolution. So by lemma (E.3),

**Corollary E.2.** *The locally-unique gapped ground state of an almost local Hamiltonian is pure.*

Thus, we have shown that

**Proposition E.1.** *Suppose $\{|\psi_L\rangle\}$ is a sequence of SRE states associated to a convergent sequence of nearly local Hamiltonians, $\{H_L\}$, then the thermodynamic limit of $|\psi_L\rangle$ is pure (if exists[29] ).*

Alternatively, instead of talking about SRE states in finite-size systems and taking the thermodynamic limit as above, one can also directly define SRE states in the thermodynamic limit.

**Definition E.3.** *A state $\psi$ of $\mathcal{A}^{ql}$ is called short-range entangled[30] if there exists an evolution $\tau$ generated by some (possibly time-dependent) local Hamiltonian, such that*

$$\psi = \omega \circ \tau \tag{E.15}$$

*where $\omega$ is a product state, see definition A.17.*

By this definition, one immediately obtains

**Corollary E.3.** *An SRE state $\psi$ defined above is pure.*

*Proof.* By definition, there is an evolution by local Hamiltonian $\tau$ such that

$$\psi = \omega \circ \tau \tag{E.16}$$

for some pure product state $\omega$. Let $\rho : \mathcal{A}^{ql} \to \mathbb{C}$ be a positive linear functional majorized by $\psi$, i.e., $\rho \leqslant \psi$. This implies

$$\rho \circ \tau^{-1} \leqslant \omega \tag{E.17}$$

because $\rho(\tau^{-1}(A^\dagger A)) \leqslant \psi(\tau^{-1}(A)^\dagger \tau^{-1}(A)) = \omega(A^\dagger A)$ for any $A \in \mathcal{A}^{ql}$. Hence $\rho \circ \tau^{-1} = \lambda \omega$ because $\omega$ is pure, according to definition A.11. Consequently, $\rho = \lambda \psi$, which implies $\psi$ is pure because of definition A.11. $\qquad\square$

### 3. Correlation functions of SRE state and the split property

In this subsection, we prove that 2-point functions of an SRE state must exhibit exponential decay. Our proof applies to finite and infinite chains in any dimension. Moreover, we show that SRE states on infinite chains split at every site.

**Theorem E.2.** *Let $\psi$ be an SRE (on finite or infinite chain) and $A \in \mathcal{A}_X^l, B \in \mathcal{A}_Y^l$ for disjoint intervals $X, Y$, then*

$$|\psi(AB) - \psi(A)\psi(B)| < C_0 ||A|| \cdot ||B|| e^{-kd} \tag{E.18}$$

*where $d := d(X, Y) \gg 1$ and $C_0, k$ are positive constants.*

*Proof.* Without loss of generality, we assume $\psi(A) = \psi(B) = 0$ (otherwise we shift $A \to A - \psi(A)$).
By assumption, there is a finite-time evolution $\tau$ by some local Hamiltonian such that

$$\psi = \omega \circ \tau \tag{E.19}$$

where $\omega$ is a product state or factorized state in the sense

$$\omega(xy) = \omega(x)\omega(y) \tag{E.20}$$

---

[29] As is explained before, we can always find a convergent subsequence and restrict to it if necessary.

[30] In Ref. [74], a slightly different definition of SRE states on infinite chains was proposed. Our SRE states are subclass of theirs.

whenever $x, y$ are local operators with disjoint support. Thus

$$\psi(AB) = \omega(\tau(A)\tau(B)) \tag{E.21}$$

By lemma E.1 and the Lieb-Robinson bound, for any positive $r$, there exists a local operator $A_r \in \mathcal{A}^l_{B(X,r)}$ such that

$$||\tau(A) - A_r|| < C_A e^{-ar}||A|| \tag{E.22}$$

for some positive constants $a, C_A$. Similarly,

$$||\tau(B) - B_{r'}|| < C_B e^{-ar'}||B|| \tag{E.23}$$

Without any loss of generality, we assume $\omega(A_r) = \omega(B_r) = 0$, since the shift $A_r \to A_r - \omega(A_r)$ and $B_{r'} \to B_{r'} - \omega(B_{r'})$ only affect the constants $C_A$ and $C_B$ on the right hand sides.

Then we fix $r = r' = s = \lceil \frac{d}{3} \rceil$ for $d > 3$. This choice ensures that $A_s$ and $B_s$ have disjoint supports. Then we write $\tau(A) = A_s + \delta A_s$ and $\tau(B) = B_s + \delta B_s$. By definition we have

$$\begin{aligned} ||\delta A_s|| &< C_A e^{-as}||A|| \\ ||\delta B_s|| &< C_B e^{-as}||B|| \end{aligned} \tag{E.24}$$

Note

$$\omega(\tau(A)\tau(B)) = \omega((A_s + \delta A_s)(B_s + \delta B_s)) \tag{E.25}$$

There are four terms after expanding the right hand side. We estimate them term by term. For the first term,

$$\omega(A_s B_s) = \omega(A_s)\omega(B_s) = 0 \tag{E.26}$$

since $A_s$ and $B_s$ are supported on disjoint sets and $\omega$ is factorized. For the second and third terms (mixing terms),

$$|\omega(A_s \delta B_s)| \leqslant ||A_s|| \cdot ||\delta B_s|| \tag{E.27}$$

where we have used $||\omega|| = 1$, see corollary A.1. Note $||A_s|| \leqslant ||\tau(A)|| + C_A e^{-as} < C_1 ||\tau(A)||$ for some positive constant $C_1$ if $s$ is large enough. Also $||\delta B_s|| \leqslant C_B e^{-as}||B||$. So we conclude

$$|\omega(A_s \delta B_s)| \leqslant C_1 C_B ||A|| \cdot ||B|| e^{-as} \tag{E.28}$$

And similarly, the third term is also bounded by a constant multiple of $||A|| \cdot ||B|| e^{-as}$.

For the last term

$$|\omega(\delta A_s \delta B_s)| \leqslant ||\delta A_s|| \cdot ||\delta B_s|| < e^{-2as}||A|| \cdot ||B|| < C_2 e^{-as}||A|| \cdot ||B|| \tag{E.29}$$

for some positive constant $C_2$. Thus all of 4 terms in Eq. (E.25) are of $e^{-as}$ tails. Recall that $s = \frac{d}{3}$, so we conclude that

$$|\psi(AB) - \psi(A)\psi(B)| < C_0 ||A|| \cdot ||B|| e^{-kd} \tag{E.30}$$

for large $d$ and $k = \frac{a}{3}$.

$\square$

Now we turn to split property. As indicated by lemma A.4, to show that SRE states split at every site, it suffices to show that SRE states satisfy the area law of entanglement entropy (see definition A.18). It is shown in Ref. [76] that this is indeed the case. More precisely,

**Theorem E.3** (Ref. [76])**.** *A gapped quantum many-spin system on an arbitrary lattice satisfies an area law for the entanglement entropy if and only if any other state with which it is adiabatically connected (i.e., any state in the same phase) also satisfies the area law.*

By definition, SRE states can be connected to product state adiabatically, hence it satisfies the area law. By lemma A.4, we deduce

**Proposition E.2.** *If $\psi$ is an SRE state on an infinite spin chain, then it splits at every site.*

Below we give a direct proof of proposition E.2 based on the decomposition of LPA's (see lemma C.1).

**Lemma E.4.** *Let $U \in \mathcal{U}^{ql}$, then for any $\epsilon > 0$, there exists $R_\epsilon > 0$ such that*

$$||\mathrm{Ad}_U(A) - A|| < \epsilon ||A|| \tag{E.31}$$

*for any $A \in \mathcal{A}^{ql}_{(-\infty, -R_\epsilon]}$ or $A \in \mathcal{A}^{ql}_{[R_\epsilon, \infty)}$.*

*Proof to lemma E.4.* By definition, there exists a sequence of local unitary operators $U_j \in \mathcal{A}^l$ having $U$ as its limit. For any $\epsilon > 0$, there exists $j_\epsilon$ such that for all $j \geqslant j_\epsilon$

$$||U_j - U|| < \epsilon \tag{E.32}$$

Note that this implies

$$||U^{-1} - U_{j_\epsilon}^{-1}|| = ||U^{-1}(U_{j_\epsilon} - U)U_{j_\epsilon}^{-1}|| \leqslant ||U^{-1}|| \cdot ||U_{j_\epsilon} - U|| \cdot ||U_{j_\epsilon}^{-1}|| < \epsilon \tag{E.33}$$

where we have used $||ab|| \leqslant ||a|| \cdot ||b||$ and $||a|| = 1$ for unitary $a$. Since $U_{j_\epsilon}$ is a local operator, there exists $R_\epsilon$ such that $\mathrm{supp}(U_{j_\epsilon}) \subset (-R_\epsilon, R_\epsilon)$. Thus, for $A \in \mathcal{A}^{ql}_{(-\infty, -R_\epsilon]}$ or $A \in \mathcal{A}^{ql}_{[R_\epsilon, \infty)}$,

$$||U^{-1}AU - A|| \leqslant ||U^{-1}AU - U_{j_\epsilon}^{-1}AU_{j_\epsilon}|| + ||U_{j_\epsilon}^{-1}AU_{j_\epsilon} - A|| \tag{E.34}$$

The second term is 0 since $[A, U_{j_\epsilon}] = 0$. The first term is estimated as

$$||U^{-1}AU - U_{j_\epsilon}^{-1}AU_{j_\epsilon}|| \leqslant ||U^{-1}AU - U_{j_\epsilon}^{-1}AU|| + ||U_{j_\epsilon}^{-1}AU - U_{j_\epsilon}^{-1}AU_{j_\epsilon}|| \tag{E.35}$$

Note

$$\begin{aligned}
||U^{-1}AU - U_{j_\epsilon}^{-1}AU|| &\leqslant ||A|| \cdot ||U^{-1} - U_{j_\epsilon}^{-1}|| < \epsilon ||A|| \\
||U_{j_\epsilon}^{-1}AU - U_{j_\epsilon}^{-1}AU_{j_\epsilon}|| &\leqslant ||A|| \cdot ||U - U_{j_\epsilon}|| < \epsilon ||A||
\end{aligned} \tag{E.36}$$

Thus we conclude

$$||UAU^{-1} - A|| < 2\epsilon ||A|| \tag{E.37}$$

The desired inequality follows by replacing $\epsilon$ with $\frac{\epsilon}{2}$. $\qquad\square$

Now we are ready to prove proposition E.2.

*Proof to proposition E.2.* Recall the definition of the split property (definition A.16), we must construct pure states $\psi_{<0}$ and $\psi_{\geqslant 0}$ such that

$$\psi \simeq \psi_{<0} \otimes \psi_{\geqslant 0} \tag{E.38}$$

By proposition A.2, it suffices to show that for any $\epsilon > 0$, there exists $R_\epsilon > 0$ such that

$$|\psi(A \otimes B) - \psi_{<0}(A)\psi_{\geqslant 0}(B)| < \epsilon ||A|| \cdot ||B|| \tag{E.39}$$

for any $A \in \mathcal{A}^{ql}_{(-\infty, R_\epsilon]}$ and $B \in \mathcal{A}^{ql}_{[R_\epsilon, \infty)}$.

By the definition of SRE states, there exists a finite time evolution $\tau$ of a local Hamiltonian such that $\psi = \omega \circ \tau$, where $\omega$ is a product state. By lemma C.1, $\tau$ admits following decomposition,

$$\tau = \tau_{<0}\tau_0\tau_{\geqslant 0} \tag{E.40}$$

where $\tau_{<0} \in \mathcal{G}^{lp}_{<0}, \tau_0 \in \mathcal{G}^{lp}_0$ and $\tau_{\geqslant 0} \in \mathcal{G}^{lp}_{\geqslant 0}$. We define[31] $\psi_{<0} := \omega|_{<0} \circ \tau_{<0}$ and $\psi_{\geqslant 0} := \omega|_{\geqslant 0} \circ \tau_{\geqslant 0}$. By lemma A.3, $\omega_{<0}$ and $\omega_{\geqslant 0}$ are again pure product states. Applying corollary E.3, we see that $\psi_{<0}$ and $\psi_{\geqslant 0}$ are pure.

———————

[31] The naive definition $\psi_{<0} = \psi|_{<0}$ does not work since $\psi|_{<0}$ is not pure in general. However, this restriction (partial trace) indeed produces a pure state for product states, such as $\omega$.

By lemma E.4, for any $\epsilon > 0$, there exists $R_1$ such that $||\tau_0(A) - A|| < \epsilon||A||$ for $A \in \mathcal{A}^{ql}_{(-\infty, -R_1]}$ and $R_2$ such that $||(\tau^{-1}_{\geqslant 0} \rhd \tau_0)(B) - B|| < \epsilon||B||$ for $B \in \mathcal{A}^{ql}_{[R_2, \infty)}$ (see Eq. (C.2) for the definition of $\rhd$). We define $R_\epsilon = \max\{R_1, R_2\}$ and with this choice, we have

$$
\begin{aligned}
||\tau_0(A) - A|| &< \epsilon||A|| \\
||(\tau^{-1}_{\geqslant 0} \rhd \tau_0)(B) - B|| &< \epsilon||B||
\end{aligned}
\tag{E.41}
$$

for $A \in \mathcal{A}^{ql}_{(-\infty, R_\epsilon]}$ and $B \in \mathcal{A}^{ql}_{[R_\epsilon, \infty)}$. Now the left hand side of (E.39) becomes

$$
|\omega(\tau(A)\tau(B) - \tau_{<0}(A)\tau_{\geqslant 0}(B))| \leqslant ||\tau(A)\tau(B) - \tau_{<0}(A)\tau_{\geqslant 0}(B)||
\tag{E.42}
$$

where we have used $||\omega|| = 1$, see corollary A.1. On the other hand

$$
||\tau(A)\tau(B) - \tau_{<0}(A)\tau_{\geqslant 0}(B)|| \leqslant ||\tau(A)\tau(B) - \tau_{<0}(A)\tau(B)|| + ||\tau_{<0}(A)\tau(B) - \tau_{<0}(A)\tau_{\geqslant 0}(B)||
\tag{E.43}
$$

The first term on the right hand side above can be estimated as

$$
||\tau(A)\tau(B) - \tau_{<0}(A)\tau(B)|| \leqslant ||\tau_0(A) - A|| \cdot ||B|| < \epsilon||A|| \cdot ||B||
\tag{E.44}
$$

The second term can be treated similarly

$$
||\tau_{<0}(A)\tau(B) - \tau_{<0}(A)\tau_{\geqslant 0}(B)|| \leqslant ||A|| \cdot ||\tau_{<0}\tau_0\tau_{\geqslant 0}(B) - \tau_{\geqslant 0}(B)||
\tag{E.45}
$$

Notice $\tau_{<0}\tau_0\tau_{\geqslant 0}(B) - \tau_{\geqslant 0}(B) = \tau_{<0}(\tau_0\tau_{\geqslant 0}(B) - \tau_{\geqslant 0}(B))$ since $\tau_{<0}$ acts trivially on $\tau_{\geqslant 0}(B) \in \mathcal{A}^{ql}_{\geqslant 0}$. So we have

$$
||\tau_{<0}\tau_0\tau_{\geqslant 0}(B) - \tau_{\geqslant 0}(B)|| \leqslant ||\tau_0(\tau_{\geqslant 0}(B)) - \tau_{\geqslant 0}(B)||
\tag{E.46}
$$

Note the fact that $\tau_{<0}$ and $\tau_{\geqslant 0}$ are isometries, so

$$
||\tau_0\tau_{\geqslant 0}(B) - \tau_{\geqslant 0}(B)|| = ||\tau_{\geqslant 0}((\tau^{-1}_{\geqslant 0} \rhd \tau_0)(B) - B)|| = ||((\tau^{-1}_{\geqslant 0} \rhd \tau_0)(B) - B)||
\tag{E.47}
$$

See Eq. (C.2) for the definition of $\rhd$. By assumption $||(\tau^{-1}_{\geqslant 0} \rhd \tau_0)(B) - B|| < \epsilon||B||$, so we have

$$
||\tau(A)\tau(B) - \tau_{<0}(A)\tau(B)|| < \epsilon||A|| \cdot ||B||
\tag{E.48}
$$

Combining the above together, we obtain

$$
|\psi(A \otimes B) - \psi_{<0}(A)\psi_{\geqslant 0}(B)| < 2\epsilon||A|| \cdot ||B||
\tag{E.49}
$$

The desired inequality follows by replacing $\epsilon \to \frac{\epsilon}{2}$. $\qquad\square$

## 4. Proof of theorem III.3

Here we prove theorem III.3 in the main text.

**Theorem E.4.** *There cannot be a $G$-symmetric short-range entangled state for sufficiently long spin chains if the anomaly index $\omega \neq 1 \in \mathrm{H}^3(G; \mathrm{U}(1))$ or $\mathrm{H}^3_{\mathrm{diff}}(G; \mathrm{U}(1))$.*

*Proof.* Let $|\psi_L\rangle$ be a $G$-symmetric SRE state defined on a spin chain of size $L$. If there exists a sequence of SRE states $|\psi_L\rangle$ in definition E.2, thus in thermodynamic limit, it defines a $G$-symmetric state $\psi$, which is pure and splits at every site. Hence the anomaly index $\omega = 1$ according to lemma D.3, contradicting to our assumption. Thus there cannot be such sequence, which means there cannot be $G$-symmetric SRE state for sufficiently long spin chains. $\qquad\square$

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

## Symbols

| | |
|---|---|
| $\lvert \cdot \rvert$ | Absolute value of a complex number or norm of a vector, Eq. (A.4) |
| $\lVert \cdot \rVert$ | norm of an operator or a functional, Eq. (A.6), Eq. (A.36) |
| $\mathcal{A}^l$ | algebra of local operators, definition A.1 |
| $\mathcal{A}^l_\Gamma$ | algebra of local operators supported on $\Gamma$ |
| $\mathcal{A}^{ql}$ | algebra of quasi local operators, definition A.2 |
| $\mathcal{A}^{ql}_\Gamma$ | algebra of quasi local operators supported on $\Gamma$ |
| $\mathcal{B}(\mathcal{H})$ | algebra of bounded operators on Hilbert space $\mathcal{H}$ |
| $\lvert \Gamma \rvert$ | cardinality of set $\Gamma$ |
| $\mathcal{G}^{cir}$ | group of circuits, example A.3 |
| $\mathcal{G}^{\mathrm{QCA}}$ | group of quantum cellular automata (QCA), definition A.5 |
| $\mathcal{G}^{lp}$ | group of locality-preserving automorphisms (LPA), definition A.6 |
| $a \overset{\epsilon}{\in} B$ | $a$ is nearly included in $B$, definition A.7 |
| $A \overset{\epsilon}{\subset} B$ | $A$ is nearly included in $B$, definition A.7 |
| ind | GNVW index map, Eq. (A.23) |
| $\psi$ | an abstract quantum state, definition A.8 |
| $N_\psi$ | The GNS ideal associated to state $\psi$, definition A.52 |
| $\mathcal{H}_\psi$ | The GNS Hilbert space associated to state $\psi$, definition A.12 |
| $\lvert \psi \rangle$ | The representative of a state $\psi$ in GNS Hilbert space $\mathcal{H}_\psi$, definition A.12 |
| $\pi_\psi$ | The GNS homomorphism of a state $\psi$, definition A.12 |
| $\psi\rvert_\Gamma$ | The restriction of a state $\psi$ on $\Gamma$ |
| $\omega$ | pure product state, definition A.17; anomaly index, Eq. (C.12) |
| $\delta_H$ | derivation a.k.a Hamiltonian $H$ |
| $\alpha^t$ | time evolution, Eq. (A.137) |
| $\lVert H \rVert_F$ | Norm on Hamiltonians associated to a reproducing function $F$, Eq. (A.139) |
| $\mathrm{H}^n(G; A)$ | degree $n$ group cohomology of $G$ with coefficients in $A$, Eq. (B.12) |
| $\mathcal{G}^{lp}_{0, <0, \geqslant 0, \pm}$ | subgroups of $\mathcal{G}^{lp}$, definition C.1 |
| $\tau$ | finite time evolution of a local Hamiltonian (possibly time-dependent) |