# Peer review of "Lieb-Schultz-Mattis theorem in long-range interacting systems and generalizations"

_SciPost Physics_

## Round 1 · Referee Report · Anonymous (Referee 1) · 2024-11-17

Report

The authors have generalized the LSM Theorem to long-range interacting systems. Overall the result seems solid and - with a perhaps minor issue that we discuss below in point 3 - correct. Therefore it seems suitable for SciPost Physics eventually.

However we felt that the authors could improve a number of points in the paper at present, which we list below.

1) It seems to us like the new content in the paper is essentially what is in Appendices D and E, and then summarized in the main text -- do the authors agree with this assertion? In some sense this is rather awkward, because the vast majority of the appendices are in A, B and C. Our impression is that this paper essentially just points out that certain results -- maybe Lemma D.1 in particular -- can just be used to generalize LSM to long-range interacting systems with very little work. But in this respect the length of the paper seems a bit confusing, as the actual contribution of the authors was not so lengthy. If Appendices A, B and C are really just intended as a review for the reader, we felt that should be a lot more explicit, and in some sense perhaps the paper could be better organized to highlight the aspects of the work that are new. For example, the proof of Theorem A.5 is not rigorous and the authors refer to Ref. [61] for a rigorous treatment, but this seems like a crucial step in the proofs in Appendices D/E. We assume then that this result is not new. Similarly, it's not clear based on the presentation whether it was known in prior literature how to use C* algebraic results for power-law interacting systems.

2) Relatedly, we are not sure why the authors use the C*-algebraic formalism which seems to take up the vast majority of the paper, given that they state in Theorem III.2 that real systems are finite-size after all. It might be worth commenting on how much simpler some of the key proofs could be if the authors restricted to a genuinely finite system.

3) As far as we can tell, Ref. [37] does not use a C*-algebraic formalism. Is it clear that their results can be ported to C* algebra formalism? Also, Ref. [37] only deals with unique ground states, but in this paper they wish to use it for only locally-unique ones. Why is that justified?

4) Below Definition E.1, depending on the definition of "local Hamiltonian", it is not true that exponential tails can be achieved in finite time evolution, see e.g. the strongest Lieb-Robinson bounds for nearest-neighbor systems in Ref. [51]. If local allows for exponential tails already then the result is true but seems rather semantic as one might as well have just included exponential tails in h_j in Definition E.1 to begin with. The wording can be clarified here.

5) The authors mention locality preserving automorphisms in the main text but do not explain its meaning. From the appendix, this notion is like QCA with the strict locality condition relaxed; some heuristic understanding could be pointed out in the main text.

Recommendation

Ask for minor revision

---

## Round 1 · Referee Report · Jules Lamers (Referee 2) · 2024-11-27

Strengths

1. Given at least two preprints that appeared around the same time and with similar results, the contents are clearly timely and of interest to the community.
2. The background material in the appendices, with its intuitive explanations and examples, and the tone of informal blackboard discussions, can serve as a useful reference for theoretical physicists who wish to learn the operator-algebraic approach to quantum statistical mechanics.

Weaknesses

1. The paper risks falling in a gap between physics and mathematics. While the definition-lemma-theorem-proof format suggests mathematical rigour, I do not find the text very satisfactory from a mathematical viewpoint. In particular, definitions and proofs are given in detail when simple, but else are vague sketches or deferred to the literature.
2. The exposition should be improved: the short main text is hard to read by itself; in several places the discussion is more general than needed; it is very hard to navigate the huge appendix.
3. The relation to the relevant literature is not sufficiently discussed; in particular, the overlap/differences with two recent related papers is not discussed.

Report

The authors present a Lieb–Schultz–Mattis-type no-go theorem for quantum spin chains with long-range interactions (with at most k-spin interactions, subject to an 'admissibility' condition controlling the rate of decay) that are symmetric under a group G whose unitary action (by 'quantum cellular automata') preserves locality. If such a system on an infinite lattice is anomalous (has a nontrivial anomaly index), then it cannot have a locally-unique gapped G-invariant ground state (Theorem III.1). This has implications for spin chains on finite but sufficiently large lattices such that the thermodynamic limit exits (Theorem III.2). Namely, such a system does not admit G-invariant short-range entangled pure states (Theorem III.3), which means it cannot have a G-symmetric nondegenerate gapped ground state.
The authors give two examples: a spin-1/2 XXZ chain with pairwise power-law interactions, which has U(1) \times \mathbb{Z} symmetry (spin-z and translations); and a second model with a discrete global \mathbb{Z}_2 symmetry.

I am not an expert on the operator-algebraic approach to quantum spin systems or the Lieb–Schultz–Mattis theorem. Judging from the other recent papers on the topic, and the motivation to understand long-range interacting systems coming from trapped-ion and cold-atom experiments, I believe the topic is certainly interesting, and the results seem to warrant publication. It is also clear that the authors have made substantial effort to try and make their (rather technical) results understandable for non-experts in physics. On the other hand, while I do not feel qualified to judge the correctness of the results and especially proofs without familiarity with the relevant literature, the presentation of the paper does not inspire confidence in its mathematical rigour. In addition, I find it highly unlikely that, as the author have indicated, all four possible journal expectations are fulfilled.

I thus recommend major revisions.

In more detail, I have the following concerns.

1. The paper risks falling in a gap between physics and mathematics.
a. From the physics side, the context, relevance and implications could be discussed in more detail, and with more examples.
The physical significance of the Lieb–Schultz–Mattis theorem could be elaborated.
Regarding the examples, the second example in the text, eq (IV.4), really consists of two disjoint special cases, while any discussion of the more general setting containing both special cases at the same time is deferred to future work.
It would be interesting if the authors would also include the Inozemtsev spin chain

V I Inozemtsev, 'On the connection between the one-dimensional s = 1/2 Heisenberg chain and Haldane–Shastry model', J Stat Phys 59 (1990) 1143
V I Inozemtsev, 'Integrable Heisenberg–Van Vleck chains with variable range exchange', Phys Part Nucl 34 (2003), 166, arXiv:hep-th/0201001

with SU(2) \times \mathbb{Z} symmetry, comprising an elliptic (finite chain) or hyperbolic (infinite lattice) potential whose imaginary period sets the interaction range, with long-range limit the 1/r^2 Haldane–Shastry chain. See also the summary in section 2 of

R Klabbers and J Lamers, 'How coordinate Bethe ansatz works for Inozemtsev model', Commun Math Phys 390 (2022) 827, arXiv:2009.14513

b. Mathematically, the paper only seems to give simple definitions and proofs, deferring the hard part to the literature. While this may have some benefit for physicists who wish to learn the operator-algebraic approach to quantum many-body systems, I do not find the result satisfactory. Since this is a strong statement, let me give many examples to support it.
- The authors mention lattices in dim > 1 in several places, but fail to state clearly from the start that the actual results concern lattice models in 1d. The latter should be clarified from the start, and the unnecessary generality of dim > 1 should be avoided (except if particularly illuminating, in which case that should be explained).
- The explanation of 'operator algebra' in footnote 2 (p2) in fact is an (unclear) definition of a ring; it misses scalar multiplication (-> algebra), norm (-> Banach algebra), as well as adjoints (-> C* algebra) -- even though the latter setting is actually what is used. The sentence with the footnote does not make sufficiently clear that, presumably, it defines what is meant by 'local'.
- The author state twice (on p2, and in Sect A.1 on p7) that the total Hilbert space for infinite systems is not well-defined. As far as I understand, this is misleading if not incorrect: starting from a (pure, infinite) ground state in some superselection sector, one can consider all vectors differing from it at finitely many sites only, and take the norm-closure to get a Hilbert space. See e.g. Sect 1.3 of

M Jimbo and T Miwa, 'Algebraic analysis of solvable lattice models', AMS (1994)

- The simple notion of a 'state' is defined, but the terms 'pure', 'mixed', 'classical mixture' are used without definition.
- Fig 1 (= Fig 3) fails to show the asymmetry between the green (<0) and blue (\geqslant 0) parts.
- The statement of Theorem III.1 does not make clear that it assumes an infinite lattice, or that H is the hamiltonian.
- The statement of Theorem III.2 does not make clear what L actually is (presumably the authors are thinking of a 1d lattice with L sites).

2. The exposition and formatting should be improved. The two-column format and short main text with huge appendix, suggesting rejection from a earlier submission with another journal, is far from helpful.
a. Given the absence of text limit for the journal, the main text is unnecessarily dense and short, and should be expanded to make it easier to read and more clear. The stylefile for SciPost Physics should be used, or at least a single column format.
b. The appendix is too long and almost impossible to navigate. It lacks a table of contents. In definitions, it should be clear which term is actually defined. Examples should not use italics throughout. The independent numbering of definitions, lemmas, propositions, theorems, corollaries, remarks, and examples is painful. For instance, it turns out that Lemma III.2 = Theorem A.5 comes a little after Remark A.15, namely in section A.9, after Proposition A.6 and Corollary A.5, before Lemma A.8… :|

3. The relation to the relevant literature is not sufficiently discussed.
a. As far as I understand, the main results follow by combining the literature with some simple proofs.
Indeed, the lemmas all rely crucially on the literature
Lemma III.1: [30] + [37]
Lemma III.2 = Theorem A.5: [61]
Lemma III.3: [25]
Lemma III.4 = Propositions E.1–2; Prop E.1 follows from Cor E.2 to Lem E.3: [75]; Prop E.2 follows from Lem A.4 and Thm E.3: [76], or a separate proof using amongst others Lem C.1: [25].

While it is of course fine to build on the literature, this should be clarified:
- It should be indicated in the main text that Lemma III.3 is from [61].
- The key references deserve more explicit credit and mention. After the introduction, the authors should include a section to put their work in context and discuss the most important recent developments in non-technical terms, "Recently, in [refs] it was shown that … . Here we will leverage these results in order to … ." From my understanding of the paper, particularly relevant references include [25], [30], [33], [36], [37], [61], [75], [76].

b. The overlap/differences with two recent related papers is not discussed. These are

R Ma, 'Lieb-Schultz-Mattis theorem with long-range interactions', arXiv:2405.14949
Y-N Zhou and X Li, 'Validity of the Lieb-Schultz-Mattis theorem in long-range interacting systems', arXiv:2406.08948

The former is mentioned in a 'Note added' after the Acknowledgements. However, the seeming similarity between the results warrants a more detailed comparison. A subsection should be added to the Discussion in which it is stated to which the degree the papers agree, if possible not just for the theorems and their assumptions, but also for the general strategy and technology used in the proofs.

Requested changes

0- Reread the entire paper to improve grammar, which is excellent in some parts but much less so in others.
1- Use SciPost Physics style file or at least remove two-column formatting, as per 2a above.
2- Elaborate on the physical significance of the Lieb–Schultz–Mattis theorem as a no-go theorem and its physical meaning for the possible phases.
3- Add a section with context as per 3a above.
4- State from the beginning (abstract, introduction, main text, discussion) that the paper concerns spin chains, namely spin systems on 1 dimensional lattices.
5- Omit unnecessary generality. The general lattice \Lambda in Sect II can be \mathbb{Z}^d with in fact d=1.
6- p2, left: Rephrase 'including c-numbers' (= constant multiples of the identity operator) to clarify that this is about the operators rather than \Gamma.
7- Correct footnote 2 as per the above.
8- p2, right: as written the (continuous) interval \Lambda_0 is not a subset of the (discrete) lattice \Lambda.
9- Correct/clarify the statement that the total Hilbert space for infinite systems is not well-defined (as per the above, in main text and appendix)
10- Recall the definitions of pure states and classical mixtures, and explain the meaning of 'GHZ' state (and repeat on p20)
11- Correct Figs 1 and 3 as per the above
12- p3, left: explain in words that B(\Gamma,r_\varphi) is a 'ball' around \Gamma of radius r_\varphi
13- p3, right: consider defining 'homomorphism'; 'automorphism' was defined before avoiding this word
14- Comment on the role of parity (\mathbb{Z}_2 acting by spatial reflection) and, if possibe, charge and time reversal, which presumably are not an internal symmetries either. Are any of them QCA? Are they covered by the results? If not, comment on this in the Discussion, and indicate whether it might be possible to include them.
15- clarify that, since cohomology with values in the group U(1) is used, multiplicative notation is used, and in particular the neutral element is denoted by 1. Repeat this on p32.
16- p3, right: 'this new anomaly index' could be taken to mean it is newly introduced in the present paper. Clarify, which, if any, aspect of it is new wrt [25, 36].
17- below (II.1): arguments g of \alpha_{\geqslant 0}, \alpha_{<0} and \alpha_0 are missing, needed to get elements of \mathcal{G}^\textsc{gca}.
18- before (II.3): include e.g. 'namely' before the condition Ad_{\omega(g,h,k)} = 1 to clarify the statement
19- define or at least intuitively describe the meaning of '3-cocycle condition' and '3-coboundary'
20- the additive language 'shifts \omega by' should be adjusted to multiplicative 'multiplies'
21- p4, left: clarify statement in first full sentence. What does 'means' mean? Is the 'if' a 'whether', or is this an 'if/then' statement?
22- (III.1): should this hold for some d? for all d?
23- Lemma III.2: Theorem A.5 should be hyperref'd. The proof in the appendix suggests that this result is due to [61], which should be attributed here already.
24- Theorem III.1: it should be clarified that the lattice is supposed to be infinite, and that H is a hamiltonian
25- The reference to appendix D should come after the outline of the proof
26- Either the reference to Lemmas III.1 and III.2 or the 'pure and split at every site' following it should be reversed
27- Theorem III.2: it should be clarified what L is
28- Can the authors provide bounds on the gap? If not, comment on this in the Discussion.
29- To me, SU(2) and U(1) seem more appropriate than SO(3) and SO(2) in the context of spin. Change or comment on this.
30- p5, right: to which extent is J_{ij}^z allowed to depend on i,j? Can it be a function of i,j separately, of |i-j|, … ?
31- Depending on J_{ij}^z the hamiltonian may be parity invariant. Does this \mathbb{Z}_2 symmetry have consequences?
32- It should be mentioned that J_{ij}^z = 1 enhances O(2) \cong U(1) to SU(2), and consequences should be discussed
33- It would be interesting to include the Inozemtsev chain as another example, as per above
34- The second example should be split into two separate examples, namely the one with (IV.5) at C_{ij} = 1 and h_i = g_i = 0, and the case (g_i = ?) g>0 and J_{ij} = h_i = 0. If they wish, the authors can end by mentioning (IV.5) and work in progress.
35- Discussion: include a subsection discussing related recent works, as per 3b above
36- Add a table of contents to the appendix
37- Change numbering of definitions, lemmas, propositions, theorems, corollaries, remarks, and examples as per 2b above.
38- Include punctuation as appropriate to all equations in the appendix.
39- p7, Rephrase 'The operator supported … is defined to be all operators …' to clarify ('An operator supported … is defined to be any operator …'?)
40- Here superscripts 'l' is used instead of '\ell' from the main text. Make consistent throughout.
41- Following (A.6), 'Namely, …' is presumably only true when \Gamma is the whole lattice.
42- (A.10) an \otimes is missing between A and the \bigotimes
43- In definitions, use e.g. \emph{…} to indicate which word is being defined
44- p8: clarify the meaning of 'non-on-site' immediately, rather than in the 'Especially, …' after (A.14)
45- Comment on whether the notion of quasi-locality in Definition A.2 is the same as that used in

E Ilievski, M Medenjak and T Prosen, 'Quasilocal conserved operators in the isotropic Heisenberg spin-1/2 chain', Phys Rev Lett 115 (2015) 120601, arXiv:1506.05049

and related papers in the context of the generalised Gibbs ensemble.
46- In Examples, use the standard upright font (rather than italics) to make them less tiring to read.
47- In Definition A.6, explicitly define the abbreviation 'LPA'
48- In Definition A.7 use the phrase 'near inclusion' to help finding the term when used on p41
49- p11, top: clarify 'one first needs' to prove? to understand?
50- p15: 'More formally' suggests that what follows is a rephrasing of what came before. This does not seem to be the case; reformulate.
51- Definition A.12 the term 'cyclic vector' means that all of \mathcal{H}_\psi can be constructed by acting with \mathcal{C} on \ket{\psi}, which is true for \ket{\psi} = [I] but not clear from the definition
52- Cor A.2, Def A.13, …: should \mathcal{A}^{gl} be \mathcal{C}? Check for consistency throughout.
53- Footnote 53: include a reference for details.
54- Remark A.3: reformulate the first sentence, which is unclear
55- eq (A.63), (A.68), …: include appropriate whitespace between different parts of the equation
56- Theorem A.1: clarify 'The converse is also true' - namely, do (A.64) + (A.63) imply independence of \psi_0, \psi_1 or does (A.64) imply both (A.63) and the independence?
57- Theorem A.2: use \emph{…} to clarify that 'independent decomposition' is defined here
58- before (A.77): does 'below' mean next (immediately after) or further below (where?)
59- I suppose that \pi_+ = \pi_{\psi_+} and \mathcal{H}_+ = \mathcal{H}_{\psi_+} ?
60- Is the discussion in A.5 still valid for arbitrary C*-algebra \mathcal{C} or special to \mathcal{A}^{gl}?
61- Example A.10: I find the statement at the end confusing. One can certainly get arbitrarily close to -1 as N increases, but the actual limit \to -1 seems to require N\to\infty ?
61- below Proposition A.3, 'The proof is based on von Neumann's double commutant theorem so we state it here without a proof' - why the 'so' ?? Moreover, the next page starts with a 'Proof to proposition A.3'?!
62- (A.95) explain the notation with the double arrow
63- Around Definition A.16 it seems to me that *any* semi-infinite \Gamma (not necessarily at the origin) would work equally well: certainly for the definition; and presumably with equivalent result for translationally invariant systems. Clarify and comment on this.
64- above (A.104): should this 'hence' be 'because'
65- Definition A.21, Corollary A.4: is this 'the' or 'a' ground state?
66- line below (A.128): \dag missing in 0 = \psi(A \delta_H(A))
67- Definition A.23: is Z \subset \Lambda ?
68- line below (A.131): 'cannot' (ie impossible) or 'might not' ?
69- Lemma A.6: replace 'above definition' by (A.131)
70- Proposition A.4: a strongly continuous …what?
71- above (A.138): use \emph{…} for 'reproducing'
72- Corollary A.5: clarify 'Typically'
73- Corollary A.6: clarify (A.157) - does there exist C>0 such that for all u,v, |(u,v)| \leqslant C ||u||\cdot ||v|| ? Moreover, include a 'Then' in the appropriate place.
74- 'Proof to theorem A.5': change to e.g. 'Sketch of proof to theorem A.5', similarly for all other proofs that are heuristic rather than rigorous.
75- B.I: 'assumed to be unitary' is this about the group?? or its representation.
76- above (B.11) explain that G^n = G \times \dots \times G (n times)
77- Example B.4: mention that additive notation is used for the group A
78- Example B.5: again, mention additive notation. change \mathcal{A} -> A.
79- on p36, superscript * should be a \bullet (cf notation elsewhere)
80- Comment on choice to cut at 0, give reference to earlier appendix where this appeared, and comment on whether the results are independent of this choice for translationally invariant systems
81- recall that \mathcal{G}^{lp} was defined in Definition A.6
82- Definition C.1: clarify whether is U a single, any single, or all quasi-local unitaries; and include missing superscript 'gl' in part 3
83- (C.3) clarify that \alpha(g) = Ad_U
84- Lemma C.1: recall what 'LPA' stands for and refer to Definition A.6
85- (C.8) and elsewhere: improve spacing to make the formula easier to read, and make notation consistent - \alpha_{\geqslant 0}(g) vs \alpha(g)_{\geqslant 0} etc
86- above (C.13): define 'generalized translations'
87- (C.14) define \omega_\beta
88- Proposition C.1: state the conditions! does this depend on whether the system is translationally invariant (or even has an action of the translation group at all)?
89- Corollary C.1, proof: the proof only deals with a single boundary, yielding a semi-infinite lattice. what about the case of two boundaries? this seems to be contained in the statement of the Corollary.
90- above Remark C.1: last sentence of proof should not be in the proof. define 'SPT'
91- section E.1: define 'having a size L'
92- below (E.2): if there exist multiple such convergent subsequences with different limits then the thermodynamic limit does not exist. comment on the physical meaning on this.
93- Theorem E.3: define 'in the same phase'

Recommendation

Ask for major revision

---

## Editorial Decision

awaiting_resubmission